# GENERALIZATION FOR LEAST SQUARES REGRESSION WITH SIMPLE SPIKED COVARIANCES

## ABSTRACT

Random matrix theory has proven to be a valuable tool in analyzing the generalization of linear models. However, the generalization properties of even two-layer neural networks trained by gradient descent remain poorly understood. To understand the generalization performance of such networks, it is crucial to characterize the spectrum of the feature matrix at the hidden layer. Recent work has made progress in this direction by describing the spectrum after a single gradient step, revealing a spiked covariance structure. Yet, the generalization error for linear models with spiked covariances has not been previously determined. This paper addresses this gap by examining two simple models exhibiting spiked covariances. We derive their generalization error in the asymptotic proportional regime. Our analysis demonstrates that the eigenvector and eigenvalue corresponding to the spike significantly influence the generalization error.

## 1 INTRODUCTION

Significant theoretical work has been dedicated to understanding generalization in linear regression models (Dobriban & Wager, 2018; Advani et al., 2020; Mel & Ganguli, 2021; Derezinski et al., 2020; Hastie et al., 2022; Kausik et al., 2024; Wang et al., 2024a). In an effort to extend this understanding to two-layer neural networks, researchers have explored various approximations, including the random features model (Mei et al., 2022; Mei & Montanari, 2021; Jacot et al., 2020), the mean-field limit of two-layer networks (Mei et al., 2018), the neural tangent kernel (Jacot et al., 2018; Adlam & Pennington, 2020), and kernelized ridge regression (Barzilai & Shamir, 2024; Liang et al., 2020; Xiao et al., 2022; Hu et al., 2024).

For the random features approximation, the first layer of the neural network is considered fixed, and only the outer layer is trained. Concretely, consider a two-layer neural network

$$f(x) = \sum_{i=1}^{m} \zeta_i \sigma(w_i^T x) = \zeta^T \sigma(W^T x),$$

where $x \in \mathbb{R}^d$ is a data point, $[\zeta_1, \ldots, \zeta_m]^T = \zeta \in \mathbb{R}^m$ are the outer layer weights, and $w_i \in \mathbb{R}^d$ for $i = 1, \ldots, m$ ($W \in \mathbb{R}^{d \times m}$) are the inner layer weights. Let us define $F = \sigma(XW)$ as the feature matrix, where $X \in \mathbb{R}^{n \times d}$ is the data matrix. It has been shown that to understand the generalization, we need to analyze the distribution of singular values of $F$. Works such as Pennington & Worah (2017); Adlam et al. (2019); Benigni & Péché (2021); Fan & Wang (2020); Wang & Zhu (2024); Péché (2019); Piccolo & Schröder (2021) have studied the spectrum of $F$ in the asymptotic limit, enabling us to understand the generalization. However, random feature models do not leverage the feature learning capabilities of neural networks. To gain further insights into the performance of two-layer neural networks and their feature learning capabilities, we need to train the inner layer.

Recent studies such as Ba et al. (2022); Moniri et al. (2023) have examined the effects on $F$ of taking one gradient step for the inner layer. Specifically, Ba et al. (2022) showed that with a sufficiently large step size $\eta$, two-layer models can already outperform random feature models after just one step. Moniri et al. (2023) extended this work to study many different scales for the step size. Concretely, let $(x_1, y_1), \ldots, (x_n, y_n)$ be $n$ data points and let $\eta \approx n^\alpha$ be the step size with $\alpha \in \left( \frac{\ell-1}{2\ell}, \frac{\ell}{2\ell+2} \right)$ for $\ell \in Z_{\geq 0}$. Perform one gradient step with the following loss function

$$\mathcal{L}_{tr} = \frac{1}{n} \sum_{i=1}^{n} \left( y_i - \zeta^T \sigma(W^T x_i) \right)^2$$

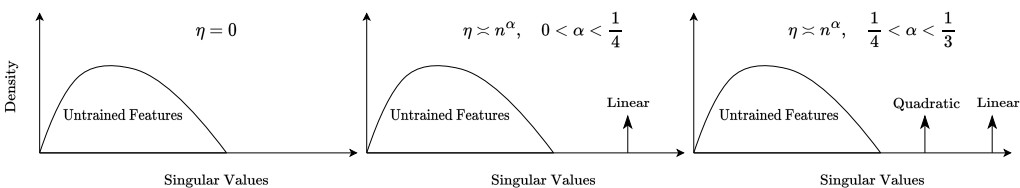

**Figure 1.1:** Figure from Moniri et al. (2023) showing the singular values of $F_0 + P$. The bulk corresponds to $F_0$, while the spikes represent the effect of $P$.

to get $W_1$. Then, they showed that there exists a rank $\ell$ matrix $P$ such that

$$\sigma(XW_1) =: F_1 = F_0 + P + o(\sqrt{n}),$$

where $F_0$ represents the features at initialization and $X \in R^{n \times d}$ is the data matrix. As shown in Figure 1.1, the singular values of $F_0 + P$ consist of two parts: the *bulk* corresponds primarily to the singular values of $F_0$, and the *spikes* correspond to the isolated singular values due to $P$. Hence, the sample covariance $\frac{1}{n} F_1^T F_1$ has a spiked structure.

Furthermore, they demonstrated that if

$$\mathcal{L}_{te}(\zeta) = \mathbb{E}_{x,y} \| y - \zeta^T \sigma(W_1^T x) \|^2$$

is the expected mean squared generalization error, then for some regimes, $\mathcal{L}_{te}(\zeta^*(F_1))$ is asymptotically equal to $\mathcal{L}_{te}(\zeta^*(F_0 + P))$, where $\zeta^*(F)$ is the minimum norm solution to the ridge regularized problem with features $F$,

$$\zeta^*(F) := \arg\min_{\zeta} \frac{1}{n} \| y - F\zeta \|^2 + \lambda \| \zeta \|^2.$$

*However, they did not quantify $\mathcal{L}_{te}(\zeta^*(F_0 + P))$.* The challenge with quantifying the effect of the spike on the generalization error is that since we have a fixed number of spikes, *the asymptotic spectrum does not see the spike.* This paper takes a step towards quantifying such errors.

**Contributions** This paper considers linear regression where the data $x$ has a spiked covariance model. The main contributions of the paper are as follows.

(i) We consider two linear regression problems that capture some of the challenges present in the spiked covariance model from Moniri et al. (2023). These regression problems extend prior models from Hastie et al. (2022) and Li & Sonthalia (2024). In particular, we introduce two regression targets, providing a more comprehensive understanding of how the learned features interact with the target function.

(ii) We derive closed-form expressions for the generalization error for both models (Theorem 3 and Theorem 4), offering precise quantification of the generalization error.

(iii) We show that the risk can be decomposed into the asymptotic risk for the unspiked case plus a correction term that depends on the eigenvector and eigenvalue corresponding to the spike. We show that for finite matrices, if the variance for the distribution of the bulk eigenvalues is small enough, then the correction is significant.

**Other Related Works** Spiked covariance models have gotten significant attention. Prior works such as Benaych-Georges & Nadakuditi (2012); Baik & Silverstein (2006) have examined the largest eigenvalue and its corresponding eigenvector in the asymptotic limit. Spiked covariances can be seen when denoising low rank signals (Nadakuditi, 2014; Sonthalia & Nadakuditi, 2023; Kausik et al., 2024). Additionally, recent work Li & Sonthalia (2024) also considers two particular spiked covariance models and shows interesting double descent phenomena. See Couillet & Liao (2022) for more applications where spiked covariance models appear.

Recent work has also sought to understand the features of the spectrum beyond a single step (Wang et al., 2024b) and for three layer neural networks (Wang et al., 2024c; Nichani et al., 2023).

**Paper structure** The rest of the paper is organized as follows. Section 2 provides a brief introduction to random matrix theory and how it can be used to understand the generalization error of the models. This section also highlights the difficulty in understanding the generalization error for spiked covariance models. Section 3 sets up the problem formulation we analyze, and Section 4 presents our theoretical results. Finally, Section 5 represents some limitations of our work and avenues for future work.

## 2 CHALLENGES WITH SPIKED COVARIANCES

In this section, we identify and elaborate on the specific challenges the spiked covariances from Moniri et al. (2023) introduce in analyzing generalization errors.

### 2.1 RANDOM MATRIX THEORY BACKGROUND

We need to define a few important objects for this discussion and, more broadly, for the paper. Let $\mathcal{D}$ be a distribution on $\mathbb{R}^d$ with uncentered covariance $\Sigma = \mathbb{E}_{x \sim \mathcal{D}}\left[xx^T\right]$ and let $X = [x_1, \ldots, x_n]^T$ be I.I.D. samples from $\mathcal{D}$. Let $\hat{\Sigma} = \frac{1}{n}X^T X$ be the sample covariance matrix.

**Definition 1** (Empirical Spectral Distribution (e.s.d.)). *Let $\lambda_1, \ldots, \lambda_n$ be the eigenvalues of a matrix $\Sigma$ and $\delta(x)$ be the Dirac delta function. Then the empirical spectral distribution of $\Sigma$ is*

$$\nu_\Sigma(\lambda) := \frac{1}{d}\sum_{i=1}^{d}\delta_{\lambda_i}(\lambda).$$

One of the most common assumptions made in this field is that as $d \to \infty$, $\nu_\Sigma$ converges almost surely to a deterministic measure $\nu_H$ at every point of continuity of $\nu_H$[1]. Once we know the e.s.d. for the population covariance, we can express the limiting risk for ride regression as a function of the deterministic quantity $\nu_H$ (Dobriban & Wager, 2018; Hastie et al., 2022). One of the most common ways of describing $\nu_H$ is via its Stieltjes transform.

**Definition 2** (Stieltjes Transform). *Given a measure $\nu$ on $\mathbb{R}$ or its corresponding density function $f_\nu$, the Stieltjes transform $m_\nu : \mathbb{C} \setminus \text{supp}(\nu) \to \mathbb{C}$ of $\nu$ is defined by*

$$m_\nu(z) := \int \frac{1}{\lambda - z}d\nu(\lambda) = \int \frac{1}{\lambda - z}f_\nu(\lambda)d\lambda.$$

For the sample covariance, we see that

$$m_{\nu_{\hat{\Sigma}}}(z) = \frac{1}{d}\sum_{i=1}^{d}\frac{1}{\lambda_i(\hat{\Sigma}) - z} = \frac{1}{d}\text{Tr}\left[\left(\hat{\Sigma} - zI\right)^{-1}\right].$$

One of the seminal results in random matrix theory develops a connection between the limiting e.s.d. for the *population* covariance matrix and the limiting e.s.d. for the *sample* covariance matrix. Marchenko & Pastur (1967) showed that under some mild assumptions, the following theorem holds:

**Theorem 1** (Marchenko & Pastur (1967)). *Let $\{(n_k, d_k)\}_{k \in \mathbb{N}}$ be a sequence of pairs of integers such that $d_k/n_k \to c$ as $k \to \infty$. Suppose $\Sigma(d_k)$ and $X_k \in \mathbb{R}^{n_k \times d_k}$ has $n_k$ I.I.D. samples from $\mathcal{N}(0, \Sigma(d_k))$. If $\nu_\Sigma$ converges almost surely to $\nu_H$, then there exists a deterministic $\nu_F$ such that the e.s.d. of the sample covariance matrix $\nu_{\hat{\Sigma}}$ converges almost surely to $\nu_F$ at all points of continuity of $\nu_F$ and for all $z \in \mathbb{C}^+$, we have that $m_{\nu_{\hat{\Sigma}}}(z) \to m_{\nu_F}(z)$, where*

$$m_{\nu_F}(z) = \int \frac{1}{t\left(1 - c - czm_{\nu_F}(z)\right) - z}d\nu_H(t).$$

The result is more general, but we provide a simplified version here.

**Example 1.** *Suppose $\Sigma(d) = I$. Then the e.s.d. of $\Sigma$ is a Dirac delta measure at 1, so its limiting e.s.d. $\nu_H$ is $\delta_1$. Hence, in this case, if we apply Theorem 1, we have that the Stieltjes transform of the limiting e.s.d. $\nu_F$ for the sample covariance satisfies the following*

$$m_{\nu_F}(z) = \frac{1}{(1 - c - czm_{\nu_F}(z)) - z}.$$

---

[1] Note a measure is continuous at $x$ if and only if $\nu(\{x\}) = 0$.

*Such distributions $\nu_F$ are called the Marchenko-Pastur distribution with shape c.*

Results such as the above theorem from Marchenko & Pastur (1967) and Theorem 1.1 Bai & Zhou (2008) are qualitative results about the limit. Prior work such as Dobriban & Wager (2018); Wu & Xu (2020); Advani et al. (2020); Xiao et al. (2022) use these results to understand the limiting generalization error. However, there are also quantitative versions. Specifically, works such as Hastie et al. (2022) use results from Knowles & Yin (2017) to provide more nuanced conclusions.

## 2.2 CHALLENGES WITH SPIKED COVARIANCE

Let us recall the setup from Moniri et al. (2023). Specifically, they assume that the outer layer weight $\zeta \sim \mathcal{N}(0, \frac{1}{m}I)$ and inner layer weights $w_i \sim \mathrm{Unif}(\mathbb{S}^{d-1})$ with $W_0 = [w_1 \ldots w_m]^T \in \mathbb{R}^{m \times d}$. Additionally, they assume that the training data $(x_1, y_1), \ldots, (x_n, y_n)$ is of the following form:

$$x_i \sim \mathcal{N}(0, I) \text{ and } y_i = \sigma_*(w^T x) + \varepsilon_i.$$

Here $\varepsilon \sim \mathcal{N}(0, 1)$, $w \sim \mathcal{N}(0, \frac{1}{d}I)$, and $\sigma_*$ is $\Theta(1)$-Lipschitz. Then $W_1$ is obtained after taking one gradient step. Let $\tilde{X}$ and $\tilde{y}$ be new independent data and let $F_1 = \sigma(\tilde{X} W_1)$, and $F_0 = \sigma(\tilde{X} W_0)$. In the proportional asymptotic regime, with some additional technical assumptions on the student networks activation $\sigma$, Moniri et al. (2023) shows that

$$F_1 = F_0 + C(\sigma, \eta)(\tilde{X}w) \cdot \zeta^T + o(\sqrt{n}),$$

where $C(\sigma, \eta)$ is a constant. After taking one gradient step for the inner layer, we train the outer layer using least squares ridge regression. Hence, to understand the generalization performance of such networks, we need to understand the generalization error for the following problem.

$$\zeta^*(F) := \arg\min_{\zeta} \frac{1}{n} \|y - F\zeta\|^2 + \lambda \|\zeta\|^2.$$

The standard approach to do this is by understanding the spectrum of $F_1^T F_1$. We break $F_1^T F_1$ up into three terms – the diagonal terms $F_0^T F_0$ and $C(\sigma, \eta)^2 \|(\tilde{X}w)\|^2 \zeta \zeta^T$ and the cross term $F_0 \tilde{X}w \zeta^T$.

**Spectrum of** $C(\sigma, \eta)^2 \|(\tilde{X}w)\|^2 \zeta \zeta^T$**:** This term is important for understanding the feature learning capabilities of neural networks, as it is the new term that appears after taking one gradient step. *However, the issue is that the limiting e.s.d. for the population covariance does not see this spike.* For example, suppose the population covariance matrix is

$$\Sigma = I + \ell u u^T.$$

Then the e.s.d. for $\Sigma$ is given by $\frac{1}{d}\delta_{\ell+1} + \frac{d-1}{d}\delta_1$, which converges to $\delta_1$ once we send $d \to \infty$. This illustrates the case when the limiting spectrum does not "see" the spike. However, if we consider the value of the largest eigenvalue of the sample covariance, then it can converge to something outside of the support (Baik & Silverstein, 2006; Benaych-Georges & Nadakuditi, 2012; Couillet & Liao, 2022).

**Theorem 2** (Baik & Silverstein (2006) Theorem 1.1)**.** *Under the same setting as Theorem 1, and let $\Sigma = I + \ell u u^T$. Denoting $\hat{\lambda}_1$ the largest eigenvalue of $\frac{1}{n}X^T X$, as $n, d \to \infty$ with $d/n \to c \in (0, 1)$, we have that*

$$\hat{\lambda}_1 \to \begin{cases} \ell + c\frac{\ell}{\ell-1} & \ell > 1 + \sqrt{c} \\ (1 + \sqrt{c})^2 & \ell \le 1 + \sqrt{c} \end{cases}.$$

From Theorem 1, the limiting spectrum for the eigenvalues for the sample covariance matrix is the Marchenko-Pastur distribution, whose support is $[(1 - \sqrt{c})^2, (1 + \sqrt{c})^2]$. However, we see that if $\ell$ is big enough, then the largest eigenvalue escapes from the continuous bulk on $[(1 - \sqrt{c})^2, (1 + \sqrt{c})^2]$ and creates a spike. This spike is from a set of measure zero. Hence, its effect on the generalization error cannot be detected in the asymptotic limit. However, in the finite case, this spike affects the generalization error. In Section 4 we provide a concrete example for this.

**Spectrum of $F_0^T F_0$** For this, Theorem 1.4 from Péché (2019) establishes that the spectrum of $\frac{1}{n} F_0 F_0^T$ can be approximated by the spectrum of

$$\frac{1}{n} \left( \phi_2 \tilde{X} W_0 + \phi_1 \Xi \right)^T \left( \phi_2 \tilde{X} W_0 + \phi_1 \Xi \right),$$

where $\Xi$ has IID standard Gaussian entries and $\phi_1, \phi_2$ are constants that only depend on $\sigma$. Here $\tilde{X}$ and $W_0$ are freely independent, and the asymptotic spectrum of $\frac{1}{n} X X^T$ and $W_0^T W_0$ are described by appropriate Marchenko-Pastur distributions. Prior work such as Nadakuditi & Edelman (2008) can be used to analytically determine the spectrum of the product.

**Spectrum of Cross Term -** $F_0 \tilde{X} w \, \zeta^T$ This term is also particularly challenging as $F_0$ and $\tilde{X}$ are both functions of $\tilde{X}$ and hence are dependent. This paper shall not consider this dependence.

## 3 PROBLEM SETTING

Building upon the challenges identified in Section 2, we explore two spiked covariance settings. We aim to understand the generalization error of least squares regression in such data models.

### 3.1 DATA MODEL

We consider a data matrix $X \in \mathbb{R}^{n \times d}$, whose rows are the data points, that is generated as the sum of a rank-one signal component corresponding to the spike and a full rank component corresponding to the bulk. *Since we are interested in the effect of the spike on the risk, we call the spike component the **signal** and we shall refer to the bulk as the **noise**.* This is the signal plus noise spiked covariance model from Couillet & Liao (2022).

$$X = Z + A.$$

The *signal (spike) component* is represented by $Z$. Let $u \in \mathbb{R}^d$ be a fixed unit-norm vector representing the direction of the spike in the covariance matrix[2]. We generate $Z$ as:

$$Z = \theta v u^T,$$

where $\theta$ scales the norm of the matrix, and $v \in \mathbb{R}^n$ has unit norm.

The *noise (bulk) component* is represented by $A$. The noise matrix $A \in \mathbb{R}^{n \times d}$ has entries $A_{ij}$ that are independent and identically distributed (i.i.d.) with mean zero and variance $\tau_A^2/d$. Additionally:

- The entries of $A$ are uncorrelated.
- The distribution of $A$ is rotationally bi-invariant; it remains the same under orthogonal transformations from both the left and the right.
- $A$ is full rank with probability 1, and empirical spectral distribution of $\frac{1}{\tau_A^2} A A^T$ converges to the Marchenko-Pastur distribution as $n, d \to \infty$ with $n/d \to c$.

Note that the isotropic Gaussian satisfies all of the noise assumptions. For a larger family of distributions that satisfy the assumptions, see Sonthalia & Nadakuditi (2023).

**Connection to two-layer model** In the setting of Moniri et al. (2023) we can think of $A$ as the representing $F_0$[3]. We can also think of $u$ as being $Aw$ for some isotropic Gaussian vector $w$. In this situation, $A$ and $Z$ should be dependent. We shall not consider this dependence and assume that $Z, A$ are independent. This difference is significant. However, understanding the generalization error while ignoring this dependence is still an important step.

### 3.2 TARGET FUNCTIONS

We study two different scenarios for the target vector $y \in \mathbb{R}^n$, depending on whether the target depends solely on the signal or on both the signal and the noise:

---

[2]This is not the exact eigenvector for the spike, as we have perturbed it by the noise matrix

[3]Note that this is note exact as the limiting e.s.d. for $F_0$ is not necessarily the Marchenko-Pastur distribution, only true of $\phi_2 = 0$. This difference is not too important, as instead of using the Stieltjes transform for the Marchenko-Pastur distribution in our paper, we could use the result from Péché (2019); Piccolo & Schröder (2021) instead.

**Signal-Only Model:** The target depends only on the signal (spike) component $Z$:

$$y_i = z_i^T \beta_* + \varepsilon_i,$$

where $\beta_* \in \mathbb{R}^d$ is the true parameter vector we aim to estimate and $\varepsilon_i$ is the observation noise, independent of $Z$ and $A$, with $\mathbb{E}[\varepsilon_i] = 0$ and $\mathbb{E}[\varepsilon_i^2] = \tau_\varepsilon^2$. If we consider our analogy of $u = Aw$, then we see that $y_i$ is similar to a quadratic function of the data.

**Signal-plus-Noise Model:** The target depends on both the signal (spike) component $Z$ and the noise (bulk) component $A$:

$$y_i = (z_i + a_i)^T \beta_* + \varepsilon_i,$$

where $a_i \in \mathbb{R}^d$ is the $i$-th row of $A$.

### 3.3 LEAST SQUARES ESTIMATION AND RISK

We consider least squares regression to estimate the parameter vector $\beta \in \mathbb{R}^d$, with regularization parameter $\mu > 0$.

For the *signal-only problem*, we solve:

$$\beta_{so} = \arg\min_\beta \|y - X\beta\|_2^2 + \mu^2 \|\beta\|_2^2, \tag{3.1}$$

For the *signal-plus-noise problem*, we solve:

$$\beta_{spn} = \arg\min_\beta \|y - X\beta\|_2^2, \tag{3.2}$$

**Instance-Specific Risk:** We consider the *instance-specific risk*, the error obtained when evaluating performance on a specific testing dataset. We introduce testing data $X_{tst} = Z_{tst} + A_{tst}$, generated similarly to the training data but potentially with different variances.

To account for possible differences between training and testing data, we distinguish: $\theta_{trn}^2, \theta_{tst}^2$ as the strengths of the signal (spike) component and $\tau_{A_{trn}}^2, \tau_{A_{tst}}^2$ as the variances of the noise (bulk) component, each during training and testing. Additionally, $\tau_{\varepsilon_{trn}}^2$ will denote the variance in the observation noise during training.

The instance-specific risk for the *signal-only model* is:

$$\mathcal{R}_{so}(c; \mu, \tau, \theta) = \frac{1}{n_{tst}} \mathbb{E}\left[ \|Z_{tst}\beta_* - X_{tst}\beta_{so}\|_2^2 \right], \tag{3.3}$$

where the expectation is over the randomness in $A_{trn}, A_{tst}, \varepsilon_{trn}$. $\tau$ collectively represents all the variances involved and $\theta$ represents both $\theta_{trn}$ and $\theta_{tst}$.

The instance-specific risk for the *signal-plus-noise model* is:

$$\mathcal{R}_{spn}(c; \tau, \theta) = \frac{1}{n_{tst}} \mathbb{E}\left[ \|X_{tst}\beta_* - X_{tst}\beta_{spn}\|_2^2 \right]. \tag{3.4}$$

By analyzing these two settings, we aim to understand how the spike in the covariance matrix affects the generalization performance. The distinction between the signal-only and signal-plus-noise models allows us to explore how the inclusion of the noise (bulk) in the target function influences the estimator's ability to generalize.

## 4 GENERALIZATION ERROR

This section presents the generalization errors for the two models. The detailed proof can be found in Appendix A and B. We present a proof sketch at the end of the section. We begin by considering the signal-plus-noise model first. Here we use the Vinogradov notation where $f \ll g$ means $f = O(g)$.

**Theorem 3** (Risk for Signal Plus Noise Problem). *Let $\tau_{\varepsilon_{trn}} \asymp 1$, $d/n = c + o(1)$ and $d/n_{tst} = c + o(1)$. Then, for any data $X \in \mathbb{R}^{n \times d}, y \in \mathbb{R}^n$ from the signal-plus-noise model that satisfy: $1 \ll \tau_{A_{trn}}^2, \tau_{A_{tst}}^2 \ll d$, $\theta_{trn}^2/\tau_{A_{trn}}^2 \ll n$, $\theta_{tst}^2/\tau_{A_{tst}}^2 \ll n_{tst}$. Then for $c < 1$, the instance specific risk is given by*

$$\mathcal{R}_{spn}(c; \tau, \theta) = \left[ \frac{\theta_{tst}^2}{n_{tst}} \frac{1}{(\theta_{trn}^2 c + \tau_{A_{trn}}^2)} + \frac{\tau_{A_{tst}}^2}{\tau_{A_{trn}}^2} \left( 1 - \frac{\theta_{trn}^2 c}{d(\theta_{trn}^2 c + \tau_{A_{trn}}^2)} \right) \right] \frac{c\tau_{\varepsilon_{trn}}^2}{1 - c} + o\left( \frac{1}{d} \right).$$

*For $c > 1$, it is given by*

$$\mathcal{R}_{spn}(c; \tau, \theta) = \|\beta_*\|^2 \left(1 - \frac{1}{c}\right) \frac{\tau_{A_{tst}}^2}{d} + \frac{\tau_{A_{tst}}^2 \tau_{\varepsilon_{trn}}^2}{\tau_{A_{trn}}^2} \left(1 - \frac{\theta_{trn}^2 c}{d(\theta_{trn}^2 + \tau_{A_{trn}}^2)}\right) \frac{1}{c-1} + o\left(\frac{1}{d}\right)$$

$$+ \frac{\theta_{tst}^2 \tau_{A_{trn}}^4}{n_{tst}\left(\theta_{trn}^2 + \tau_{A_{trn}}^2\right)^2} \left[\left(1 - \frac{1}{c}\right)\left((\beta_*^T u)^2 + \|\beta_*\|^2 \frac{\theta_{trn}^2}{d\tau_{A_{trn}}^2}\right) + \frac{\tau_{\varepsilon_{trn}}^2}{\tau_{A_{trn}}^4}\left(\frac{\theta_{trn}^2 c + \tau_{A_{trn}}^2}{c-1}\right)\right].$$

Theorem 3 can be seen as an extension of Theorem 1 from Hastie et al. (2022). Specifically, if we set $\tau_{A_{trn}}^2 = \tau_{A_{tst}}^2 = d$, $\theta_{trn} = \tau_{A_{trn}}^2 n$, and $\theta_{tst} = \tau_{A_{tst}}^2 n_{tst}$, and send $d/n \to c$, then we obtain from Theorem 3:

$$\mathcal{R}_{spn} = \begin{cases} \tau_{\varepsilon_{trn}}^2 \frac{c}{1-c} & c < 1 \\ \|\beta_*\|^2 \left(1 - \frac{1}{c}\right) + \tau_{\varepsilon_{trn}}^2 \frac{1}{c-1} & c > 1 \end{cases}. \tag{4.1}$$

This is the risk from Hastie et al. (2022). Hence, we can see that Theorem 3 lets us interpolate between a prior model that does not have the spike and spiked models smoothly.

Theorem 3 shows that the presence of the spike affects the risk in the non-asymptotic case. To see this, let $\theta_{tst} = \tau_{A_{tst}}\sqrt{n_{tst}}$ and $\theta_{trn} = \tau_{A_{trn}}\sqrt{n}$. This results in the $Z$ and $A$ matrices having the same expected norm. Then, in the underparameterized ($c < 1$) case, this risk simplifies to

$$\frac{\tau_{A_{tst}}^2}{\tau_{A_{trn}}^2} \cdot \tau_{\varepsilon_{trn}}^2 \cdot \frac{c}{1-c} + o\left(\frac{1}{d}\right).$$

Here we see that spike does not effect the risk. Hence, not seeing the spike in the asymptotic limit is not an issue. On the other hand, for the overparameterized ($c > 1$) case, with the same simplification ($\theta_{tst} = \tau_{tst}\sqrt{n_{tst}}$ and $\theta_{trn} = \tau_{A_{trn}}\sqrt{n}$), the risk becomes

$$\|\beta_*\|^2 \left(1 - \frac{1}{c}\right) \frac{\tau_{A_{tst}}^2}{d} + \tau_{\varepsilon_{trn}}^2 \left[\frac{1}{c-1} + \frac{\theta_{trn}^2}{(\theta_{trn}^2 + \tau_{A_{trn}}^2)^2}\right] + o\left(\frac{1}{d}\right). \tag{4.2}$$

Comparing this to the unspiked case, we should observe a correction term $\frac{\tau_{\varepsilon_{trn}}^2 \theta_{trn}^2}{(\theta_{trn}^2 + \tau_{A_{trn}}^2)^2}$ that depends on the relative strength of the spike ($\theta_{trn}$) to the bulk ($\tau_{A_{trn}}^2$). If we assume large bulk strength, that is, $\tau_{A_{trn}} = \tau_{A_{tst}} = d$, then we see that this term is of order $O(1/d^2)$, which can be ignored. In other words, the spike does not affect the risk.

However, if $\tau_{A_{trn}} = \Theta(1)$, then the correction term is of order $\Theta(1/d)$. Hence, the spike does not have an effect in the asymptotic case but does in the finite case. We verify this empirically.

Figure 4.1 shows four lines. The blue line corresponds to the true risk computed by empirically training the model. The orange line is the risk predicted by Theorem 3 or, more specifically, Equation 4.2. The green line is the correction term $\frac{\tau_{\varepsilon_{trn}}^2 \theta_{trn}^2}{(\theta_{trn}^2 + \tau_{A_{trn}}^2)^2}$. Finally, the red line is the asymptotic risk which does not have the correction term.

We consider two settings. For the left hand side figure, we let $\tau_{\varepsilon_{trn}} = 5$, $\tau_{A_{trn}} = \tau_{A_{tst}} = 1$ and $d = 5000$. We then varied $n$ from 50 to 200. Here, we can see that the spike correction term is significant and affects the risk. For the second setting, we consider the case when $\tau_{A_{trn}} = \tau_{A_{tst}} = d = 500$ is large. In this case, the correction term has a small magnitude, and both the asymptotic risk formula and Equation 4.2 match the true empirical risk.

Hence, we observe that if the target vector $y$ has a smaller dependence on the noise (bulk) component $A$, then the spike affects the generalization error. To better understand this, we can consider the extreme case where the targets $y$ only depend on the signal (spike) component $Z$. This is exactly the signal-only model.

**Theorem 4** (Risk for Signal Only Problem). *Let $\mu \geq 0$ be fixed. Let $\tau_{\varepsilon_{trn}} \asymp 0$, $d/n = c + o(1)$ and $d/n_{tst} = c + o(1)$. Then, any for data $X \in \mathbb{R}^{n \times d}$, $y \in \mathbb{R}^n$ from the signal-only model that satisfy: $1 \ll \tau_{A_{trn}}^2, \tau_{A_{tst}}^2 \ll d$, $\theta_{trn}^2/\tau_{A_{trn}}^2 \ll n$, $\theta_{tst}^2/\tau_{A_{tst}}^2 \ll n_{tst}$. Then for $c < 1$, the instance specific risk is given by*

$$\mathcal{R}(c; \mu, \tau, \theta) = \mathbf{Bias} + \mathbf{Variance_{A_{trn}}} + \mathbf{Variance_{A_{trn}, \varepsilon_{trn}}} + o\left(\frac{1}{d}\right)$$

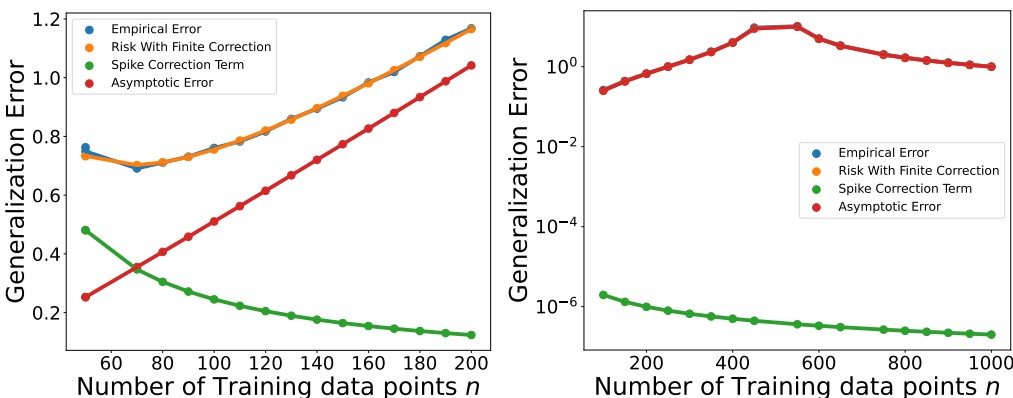

**Figure 4.1:** Figure showing the affect of the spike on the generalization error for finite matrices. Left: when the strength of the spike is large compared to the bulk, we see an effect that this is not detected by asymptotic risk. Right: the bulk and the spike have the same strength and we do not see any effects of the spike on the risk.

*with*

$$\mathbf{Bias} = \frac{\theta_{tst}^2}{n_{tst}} \frac{1}{\gamma^2} \left[ (\beta_*^T u)^2 + \frac{\tau_{\varepsilon_{trn}}^2}{2\tau_{A_{trn}}^4} \left( \theta_{trn}^2 c + \tau_{A_{trn}}^2 \right) (T_2 - 1) \right],$$

$$\mathbf{Variance_{A_{trn}}} = \frac{\theta_{trn}^2 \tau_{A_{tst}}^2}{d} \frac{1}{\gamma^2} (\beta_*^T u)^2 \left[ \frac{c \left( \theta_{trn}^2 + \tau_{A_{trn}}^2 \right)}{2\tau_{A_{trn}}^4} (T_2 - 1) \right],$$

$$\mathbf{Variance_{A_{trn}, \varepsilon_{trn}}} = \frac{\tau_{\varepsilon_{trn}}^2 \tau_{A_{tst}}^2}{2\tau_{A_{trn}}^2} \left[ 1 + \frac{c\theta_{trn}^2}{\tau_{A_{trn}}^2} \frac{T_2}{d\gamma^2} \left( \frac{(c+1)\theta_{trn}^2}{\tau_{A_{trn}}^2} + 1 \right) \right] (T_2 - 1)$$
$$- \frac{c^2(c+1)\theta_{trn}^4 \tau_{\varepsilon_{trn}}^2 \tau_{A_{tst}}^2}{d\tau_{A_{trn}}^2} \frac{1}{\gamma^2 T_1^2} - \frac{2c^2\theta_{trn}^2 \tau_{\varepsilon_{trn}}^2 \tau_{A_{tst}}^2}{d\gamma} \left( \frac{1}{T_1^2} - \frac{c\mu^2}{T_1^3} \right),$$

*where*

$$T_1 = \sqrt{\left( \tau_{A_{trn}}^2 + \mu^2 c - c\tau_{A_{trn}}^2 \right)^2 + 4\mu^2 c^2 \tau_{A_{trn}}^2}, \quad T_2 = \frac{\mu^2 c + \tau_{A_{trn}}^2 + c\tau_{A_{trn}}^2}{T_1},$$

$$and \ \gamma = 1 + \frac{\theta_{trn}^2}{2\tau_{A_{trn}}^4} \left( \tau_{A_{trn}}^2 + c\tau_{A_{trn}}^2 + \mu^2 c - T_1 \right).$$

*For $c > 1$, the same formula holds except $T_1 = \sqrt{\left( -\tau_{A_{trn}}^2 + \mu^2 c + c\tau_{A_{trn}}^2 \right)^2 + 4\mu^2 c\tau_{A_{trn}}^2}$.*

Theorem 4 breaks the risk into three terms – the bias, the variance due to bulk, and the variance due to the bulk and the observation noise. To further interpret the expression, we consider some simplifications. First, setting $\tau_\varepsilon$ to zero recovers Theorem 1 from Li & Sonthalia (2024). Second, let us consider the unregularized problem, that is $\mu = 0$.

**Corollary 1** (Non-Regularized Error). *For the same setting as Theorem 4, for $c < 1$, we have that*

$$\mathcal{R}_{so}(c; \mu = 0, \tau, \theta) = \mathbf{Bias} + \mathbf{Variance} + o\left( \frac{1}{d} \right),$$

$$\mathbf{Bias} = \frac{\theta_{tst}^2}{n_{tst} \left( \theta_{trn}^2 c + \tau_{A_{trn}}^2 \right)^2} \left( \tau_{A_{trn}}^4 (\beta_*^T u)^2 + \tau_{\varepsilon_{trn}}^2 \left( \frac{\theta_{trn}^2 c^2 + \tau_{A_{trn}}^2 c}{1 - c} \right) \right),$$

$$\mathbf{Variance} = \frac{\tau_{A_{tst}}^2 \tau_{\varepsilon_{trn}}^2 c}{\tau_{A_{trn}}^2 (1 - c)} + \left( (\beta_*^T u)^2 \frac{\theta_{trn}^2 + \tau_{A_{trn}}^2}{\theta_{trn}^2 c + \tau_{A_{trn}}^2} - \frac{\tau_{\varepsilon_{trn}}^2}{\tau_{A_{trn}}^2} \right) \frac{\theta_{trn}^2 \tau_{A_{tst}}^2}{d \left( \tau_{A_{trn}}^2 + \theta_{trn}^2 c \right)} \frac{c^2}{1 - c}.$$

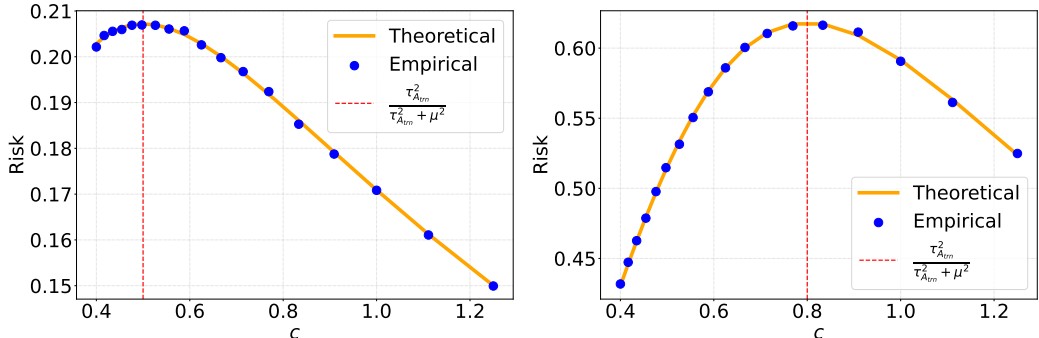

**Figure 4.2:** The peak for generalization error versus $c$ curve has a peak at $c = \frac{\tau^2_{A_{trn}}}{\tau^2_{A_{trn}}+\mu^2}$. For both figures $\mu = \tau_{\varepsilon_{trn}} = \theta_{trn} = \theta_{tst} = 1$ and $d = 1000$. Left: We set $\tau_{A_{trn}} = 1$, hence the peak should occur at $c = 1/2$. Right: We set $\tau_{A_{trn}} = 2$, hence the peak should occur at $c = 4/5$.

*For $c > 1$, the bias and variance become*

$$\textbf{Bias} = \frac{\theta^2_{tst}}{n_{tst}\left(\theta^2_{trn} + \tau^2_{A_{trn}}\right)^2}\left(\tau^4_{A_{trn}}(\beta^T_* u)^2 + \tau^2_{\varepsilon_{trn}}\left(\frac{\theta^2_{trn}c + \tau^2_{A_{trn}}}{c-1}\right)\right),$$

$$\textbf{Variance} = \frac{\tau^2_{A_{tst}}\tau^2_{\varepsilon_{trn}}}{\tau^2_{A_{trn}}(c-1)} + \left((\beta^T_* u)^2 - \frac{\tau^2_{\varepsilon_{trn}}}{\tau^2_{A_{trn}}}\right)\frac{\theta^2_{trn}\tau^2_{A_{tst}}}{d\left(\tau^2_{A_{trn}}+\theta^2_{trn}\right)}\frac{c}{c-1}.$$

Here, both the bias and variance terms have been simplified and become more interpretable. For example, the presence of $1-c$ and $c-1$ in the denominator shows that the error blows up as we approach the interpolation point ($c = 1$), leading to double descent. Suppose that $\theta_{trn} = \tau_{A_{trn}}\sqrt{n}$, $\tau^2_{A_{trn}} = d$, and $d, n \to \infty$. Then in the underparameterized case ($c < 1$), the asymptotic risk becomes

$$\tau^2_{\varepsilon_{trn}}\frac{c}{1-c} + (\beta^T_* u)\frac{1}{1-c}.$$

For the overparameterized case ($c > 1$), the asymptotic risk becomes

$$\tau^2_{\varepsilon_{trn}}\frac{1}{c-1} + (\beta^T_* u)\frac{c}{c-1}.$$

Again, we see that a correction term appears and that the asymptotic risk is dependent on the spike. Specifically, the correction term depends on the alignment between the eigenvector $u$ corresponding to the spike and the target function $\beta$. We note that this correction term also exhibits double descent. Finally, we do not get the $\|\beta_*\|^2(1 - 1/c)$ term present in Equation 4.1 as $\beta_*$ is independent of the noise.

Alignment terms such as $\beta^T_* u$ have been seen before. For example, Wei et al. (2022) considers estimating the generalization error for least squares regression for data with non-identity covariance. They show that the risk depends on the weighted alignment between the target $\beta_*$ and the eigenvectors of the covariance matrix.

**Double Descent Peak Location Depends on Variance of the Bulk:** While we obtained interpretable results in the unregularized case, we would also like to understand the regularized case. One common feature of generalization risks for least squares regression in the proportional regime is that the asymptotic risk exhibits double descent. As seen from Theorem 3 and Corollary 1, we have double descent, and the peak occurs at $c = 1$. However, looking at the formula in Theorem 4, it is unclear if the risk exhibits double descent. Empirically, examining the risk shows us that the model does exhibit double descent. However, the peak is no longer at $c = 1$ and occurs at

$$c = \frac{\tau^2_{A_{trn}}}{\tau^2_{A_{trn}} + \mu^2}.$$

Figure 4.2 empirically verify this in two cases.

**Proof Idea** In this section, we provide a brief discussion of the proofs of the two theorems, particularly focusing on how we handle the spike in the covariance matrix. The proof relies on the asymptotic limiting spectrum but only for the noise matrix $A$, not for $A + Z$. This is why we cannot let $\tau_{A_{trn}}, \tau_{A_{tst}}$ go to zero. The proof builds upon ideas from Sonthalia & Nadakuditi (2023); Li & Sonthalia (2024). The main idea is that the solution $\beta_{so}$ or $\beta_{spn}$ is of the form:

$$X^\dagger y = (Z + A)^\dagger y.$$

Here, instead of using the spectrum of $Z + A$ to quantify the error, we expand $(Z + A)^\dagger$ using the result from Meyer (1973) into sums of terms where we only invert $A$, not $A + Z$. Thus, we only care about the Stieltjes transform of $A$.

Let $\beta_0$ be the solution to the signal-only problem when $\tau^2_{\varepsilon_{trn}} = 0$. Then we see that

$$\beta_{so} = \beta_0 + (Z + A)^\dagger \varepsilon_{trn}, \quad \text{and} \quad \beta_{spn} = \beta_{so} + (Z + A)^\dagger A \beta_*.$$

We compute a bias-variance type decomposition for the risk. For example, for the signal-plus-noise problem, we decompose it as

$$\|Z_{tst}\beta_* - Z_{tst}\beta_{spn}\|^2_F + \|A_{tst}\beta_{spn}\|^2_F + \|A_{tst}\beta_*\|^2_F - 2\beta_*^T A_{tst}^T A_{tst}\beta_{spn}.$$

Then we show that each of these can be expressed as the product of *dependent* quadratic forms that are mostly of the form $\omega_1^T (A_{trn}^T A_{trn})^\dagger \omega_2$ or $\omega_1^T (A_{trn}^T A_{trn})^\dagger (A_{trn}^T A_{trn})^\dagger \omega_2$ for some vectors $\omega_1, \omega_2$. We use the almost sure weak convergence of the spectrum of $A$ to express these as $m(-\mu^2)$, i.e., the Stieltjes transform of the limiting e.s.d for $A$ at $-\mu^2$. Additionally, we show that these terms concentrate and bound the variance. See lemmas numbered 10 through 22 in the Appendix. As such, we can estimate the expectation of the product using the product of the expectations. We need to keep track of two forms of error: first, from the approximation of the finite expectation using the asymptotic version, and second, from using the product of expectations to approximate the expectation of the product. These result in the $o(1/d)$ error in the theorems.

## 5 LIMITATION AND FUTURE WORK

While this work takes an important step in understanding the generalization error for data with spiked covariances, significant work remains to be done.

**Discrepancies with Moniri et al. (2023)** There are a few discrepancies between the model considered here and the spiked covariance model from Moniri et al. (2023). Specifically:

(i) The distribution of the spectrum for $F_0$ versus that of $A$.

(ii) The dependency between $F_0$ (for us $A$) and $\tilde{X}w$ (for us $Z$).

We believe the distribution of the spectrum of $F_0$ is solvable using the techniques presented here. We need to use the appropriate Stieltjes transform, which has been studied in prior work (Péché, 2019). The dependency between $F_0$ and $\tilde{X}w$ also appears tractable but would introduce some additional quadratic forms that would need bounding. While these two problems are approachable, they require significant work and are avenues for future research.

**Multiple Spikes and Steps** This paper only considers the model where there is a singular spike. However, depending on the step size, we may see multiple spikes. We believe that the manner in which Kausik et al. (2024) generalizes Sonthalia & Nadakuditi (2023) from rank one to generic low rank could be adapted to study the problem with multiple spikes. Additionally, we only consider the case where we take one step.

## 6 CONCLUSION

The feature matrix of a two-layer neural network has been shown to have spiked covariance with finitely many spikes. However, these spikes cannot be detected when we look at the asymptotic proportional limit. Nevertheless, the spikes are crucial as they arise due to the feature learning capabilities of neural networks. This paper considers linear regression with data that has a simplified spiked covariance. We show that the models here are natural extensions of prior work and quantify the generalization error. We show that for the signal-plus-noise model, the spike has an effect for finite matrices, but this effect disappears in the asymptotic limit. For the signal-only problem, we show that the dependence on the spike appears even in the asymptotic limit.

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

## A   PROOF OF THEOREM 4 (SIGNAL ONLY)

In order to take advantage of previous results, we reformulate the problem in Equation 3.4 to make it align better with those settings. In particular, we consider

$$\beta_{so}^T = \arg\min_{\beta^T} \|\beta_*^T Z_{trn} + \varepsilon_{trn}^T - \beta^T (Z_{trn} + A_{trn})\|_F^2 + \mu^2 \|\beta\|_F^2, \tag{A.1}$$

$$\mathcal{R}_{so}(c; \mu, \tau, \theta) = \frac{1}{n_{tst}} \mathbb{E}_{A_{trn}, A_{tst}, \varepsilon_{trn}} \left[ \left\| \beta_*^T Z_{tst} - \beta_{so}^T (Z_{tst} + A_{tst}) \right\|_F^2 \right], \tag{A.2}$$

where $A_{trn} \in \mathbb{R}^{d \times n}$, $Z_{trn} = \theta_{trn} u v_{trn}^T \in \mathbb{R}^{d \times n}$, $\beta, \beta_*, \varepsilon_{trn} \in \mathbb{R}^d$. Here we simply transpose everything and adjust the matrix dimensions accordingly. We also change the dimensions of the test data in the same way. This is equivalent to Equation 3.4 but allows us to match previous settings, from which we derive important results. Now we present the full proof in five steps.

### A.1   STEP 1: DECOMPOSE THE ERROR TERM INTO BIAS AND VARIANCE

This step is foundational and relies on Lemma 1. The key idea here is to separate the error into two components: bias and variance. This decomposition is crucial because it allows us to analyze these two sources of error independently. The first term represents the bias (error due to the model's systematic deviation from the true function), and the second term represents the variance (error due to the model's sensitivity to fluctuations in the training data).

**Lemma 1.** *Suppose entries of $A_{tst} \in \mathbb{R}^{d \times n_{tst}}$ have mean 0 and variance $\tau_{A_{tst}}^2/d$. Then*

$$\mathbb{E}_{A_{trn}, A_{tst}, \varepsilon_{trn}} \left[ \|\beta_*^T Z_{tst} - \beta_{so}^T (Z_{tst} + A_{tst})\|_F^2 \right]$$

$$= \underbrace{\mathbb{E}_{A_{trn}, A_{tst}, \varepsilon_{trn}} \left[ \|\beta_*^T Z_{tst} - \beta_{so}^T Z_{tst}\|_F^2 \right]}_{Bias} + \underbrace{\mathbb{E}_{A_{trn}, A_{tst}, \varepsilon_{trn}} \left[ \|\beta_{so}^T A_{tst}\|_F^2 \right]}_{Variance}.$$

*Proof.* This lemma is a direct extension of Lemma 1 in Sonthalia & Nadakuditi (2023). It follows from the fact that the cross term is zero in expectation because the entries of $A_{tst}$ are zero in expectation. $\square$

### A.2   STEP 2: OBTAIN PRELIMINARY EXPANSIONS FOR BIAS AND VARIANCE

This step involves deriving expressions for $\beta_{so}$ and then using these to expand the bias and variance terms.

We start by reformulating the ridge-regularized regression problem:

$$\beta_{so}^T = \arg\min_{\beta^T} \|\beta_*^T Z_{trn} + \varepsilon_{trn}^T - \beta^T (Z_{trn} + A_{trn})\|_F^2 + \mu^2 \|\beta\|_F^2. \tag{A.3}$$

This can be rewritten using augmented matrices:

$$\beta_{so}^T = \arg\min_{\beta^T} \|\beta_*^T Z_{trn} + \varepsilon_{trn}^T - \beta^T (Z_{trn} + A_{trn})\|_F^2 + \mu^2 \|\beta\|_F^2$$

$$= \arg\min_{\beta^T} \|\beta_*^T \hat{Z}_{trn} + \hat{\varepsilon}_{trn}^T - \beta^T (\hat{Z}_{trn} + \hat{A}_{trn})\|_F^2,$$

where $\hat{A}_{trn} = [A_{trn} \quad \mu I]$, $\hat{Z}_{trn} = [Z_{trn} \quad 0]$, $\hat{\varepsilon}_{trn}^T = [\varepsilon_{trn}^T \quad 0]$.

The solution to this problem is given by:

$$\beta_{so}^T = (\beta_*^T \hat{Z}_{trn} + \hat{\varepsilon}_{trn}^T)(\hat{Z}_{trn} + \hat{A}_{trn})^\dagger, \tag{A.4}$$

where $\dagger$ denotes the Moore-Penrose pseudoinverse.

Let $\hat{u} = u$, $\hat{v}_{trn} = [v_{trn} \quad 0]$, $\hat{v}_{tst} = [v_{tst} \quad 0]$ such that $\hat{Z}_{trn} = \theta_{trn} u \hat{v}_{trn}^T$ and $\hat{Z}_{tst} = \theta_{tst} u \hat{v}_{tst}^T$. We then define several helper variables ($\hat{h}$, $\hat{k}$, $\hat{s}$, $\hat{t}$, $\hat{\xi}$, $\gamma$, $\hat{p}$, $\hat{q}$) to simplify our expressions. These

variables capture different aspects of the data and the solution, such as projections onto the signal and noise spaces.

$$\hat{h} = \hat{v}_{trn}^T \hat{A}_{trn}^\dagger, \qquad\qquad \hat{k} = \hat{A}_{trn}^\dagger u,$$
$$\hat{s} = (I - \hat{A}_{trn}\hat{A}_{trn}^\dagger)u, \qquad \hat{t} = \hat{v}_{trn}^T(I - \hat{A}_{trn}^\dagger \hat{A}_{trn}),$$
$$\hat{\xi} = 1 + \theta_{trn}\hat{v}_{trn}^T \hat{A}_{trn}^\dagger u, \qquad \gamma = \theta_{trn}^2 \|\hat{t}\|^2 \|\hat{k}\|^2 + \hat{\xi}^2,$$
$$\hat{p} = -\frac{\theta_{trn}^2 \|\hat{k}\|^2}{\hat{\xi}} \hat{t}^T - \theta_{trn}\hat{k},$$
$$\hat{q}^T = -\frac{\theta_{trn}\|\hat{t}\|^2}{\hat{\xi}} \hat{k}^T \hat{A}_{trn}^\dagger - \hat{h}.$$

Our main objective is to compute the expectations in Lemma 1 in terms of the regularization constant $\mu$, the asymptotic ratio or Marchenko-Pastur shape $c$, the data parameters ($\theta_{trn}$, $\theta_{tst}$, $d$, $n_{tst}$), the noise parameters ($\tau_{A_{trn}}, \tau_{A_{tst}}, \tau_{\varepsilon_{trn}}, \tau_{\varepsilon_{tst}}$), and the ground-truth parameters, in particular $\beta_*^T u$.

Note that in practice, we only assume access to $Z_{trn} + A_{trn}$, $\beta_*^T Z_{trn} + \varepsilon_{trn}$, and noise distributions during training.

**Lemma 2.** *Suppose $\gamma \neq 0$. Under our assumptions,*

$$\beta_{so}^T = \frac{\theta_{trn}\hat{\xi}}{\gamma}\beta_*^T u\hat{h} + \frac{\theta_{trn}^2\|\hat{t}\|^2}{\gamma}\beta_*^T u\hat{k}^T \hat{A}_{trn}^\dagger + \hat{\varepsilon}_{trn}^T\left(\hat{A}_{trn}^\dagger + \frac{\theta_{trn}}{\hat{\xi}}\hat{t}^T\hat{k}^T\hat{A}_{trn}^\dagger - \frac{\hat{\xi}}{\gamma}\hat{p}\hat{q}^T\right).$$

*Proof.* From our optimization setting, it is clear that the optimal solution is given by

$$\beta_{so}^T = (\beta_*^T \hat{Z}_{trn} + \hat{\varepsilon}_{trn}^T)(\hat{Z}_{trn} + \hat{A}_{trn})^\dagger$$
$$= (\theta_{trn}\beta_*^T u\hat{v}_{trn}^T + \hat{\varepsilon}_{trn}^T)(\theta_{trn}u\hat{v}_{trn} + \hat{A}_{trn})^\dagger$$
$$= \theta_{trn}\beta_*^T u\hat{v}_{trn}^T(\theta_{trn}u\hat{v}_{trn} + \hat{A}_{trn})^\dagger + \hat{\varepsilon}_{trn}^T(\theta_{trn}u\hat{v}_{trn}^T + \hat{A}_{trn})^\dagger.$$

By Theorem 3 in Meyer (1973), the pseudoinverse is

$$(\hat{A}_{trn} + \theta_{trn}u\hat{v}_{trn}^T)^\dagger = \hat{A}_{trn}^\dagger + \frac{\theta_{trn}}{\hat{\xi}}\hat{t}^T\hat{k}^T\hat{A}_{trn}^\dagger - \frac{\hat{\xi}}{\gamma}\hat{p}\hat{q}^T. \qquad (A.5)$$

By Lemma 2 in Li & Sonthalia (2024), the first term is

$$\theta_{trn}\beta_*^T u\hat{v}_{trn}^T\left(\hat{A}_{trn}^\dagger + \frac{\theta_{trn}}{\hat{\xi}}\hat{t}^T\hat{k}^T\hat{A}_{trn}^\dagger - \frac{\hat{\xi}}{\gamma}\hat{p}\hat{q}^T\right) = \frac{\theta_{trn}\hat{\xi}}{\gamma}\beta_*^T u\hat{h} + \frac{\theta_{trn}^2\|\hat{t}\|^2}{\gamma}\beta_*^T u\hat{k}^T\hat{A}_{trn}^\dagger.$$

We then combine these results. $\qquad \square$

**Lemma 3.** *Suppose $\gamma \neq 0$. Under our assumptions,*

$$y - \beta_{so}^T Z_{tst} = \beta_*^T Z_{tst} - \beta_{so}^T Z_{tst} = \frac{\hat{\xi}}{\gamma}\beta_*^T Z_{tst} + \frac{\theta_{tst}\hat{\xi}}{\theta_{trn}\gamma}\hat{\varepsilon}_{trn}^T\hat{p}v_{tst}^T.$$

*Proof.* From Lemma 2, we know that

$$\beta_*^T Z_{tst} - \beta_{so}^T Z_{tst} = \beta_*^T Z_{tst} - \left(\frac{\theta_{trn}\hat{\xi}}{\gamma}\beta_*^T u\hat{h} + \frac{\theta_{trn}^2\|\hat{t}\|^2}{\gamma}\beta_*^T u\hat{k}^T\hat{A}_{trn}^\dagger\right)Z_{tst}$$

$$- \hat{\varepsilon}_{trn}^T\left(\hat{A}_{trn}^\dagger + \frac{\theta_{trn}}{\hat{\xi}}\hat{t}^T\hat{k}^T\hat{A}_{trn}^\dagger - \frac{\hat{\xi}}{\gamma}\hat{p}\hat{q}^T\right)Z_{tst}.$$

Substitute $Z_{tst} = \theta_{tst} u v_{tst}^T$. The first two terms can be rewritten as

$$\theta_{tst}\beta_*^T \left( uv_{tst}^T - \frac{\theta_{trn}\hat{\xi}}{\gamma}u\hat{h}uv_{tst}^T + \frac{\theta_{trn}^2\|\hat{t}\|^2}{\gamma}u\hat{k}^T\hat{A}_{trn}^\dagger uv_{tst}^T \right)$$

$$= \theta_{tst}\beta_*^T \left( uv_{tst}^T - \frac{\theta_{trn}\hat{\xi}}{\gamma}u\hat{v}_{trn}^T\hat{A}_{trn}^\dagger uv_{tst}^T + \frac{\theta_{trn}^2\|\hat{t}\|^2}{\gamma}u\hat{k}^T\hat{A}_{trn}^\dagger uv_{tst}^T \right).$$

Note $\hat{\xi} - 1 = \theta_{trn}\hat{v}_{trn}^T\hat{A}_{trn}^\dagger u$, $\hat{k}^T\hat{A}_{trn}^\dagger u = \hat{k}^T\hat{k} = \|\hat{k}\|^2$. The above equation becomes

$$\theta_{tst}\beta_*^T \left( uv_{tst}^T - \frac{\hat{\xi}(\hat{\xi}-1)}{\gamma}uv_{tst}^T - \frac{\theta_{trn}^2\|\hat{t}\|^2\|\hat{k}\|^2}{\gamma}uv_{tst}^T \right).$$

Using $\gamma = \theta_{trn}^2\|\hat{t}\|^2\|\hat{k}\|^2 + \hat{\xi}^2$ to combine the coefficients, we have that

$$1 - \frac{\hat{\xi}(\hat{\xi}-1)}{\gamma} - \frac{\theta_{trn}^2\|\hat{t}\|^2\|\hat{k}\|^2}{\gamma} = \frac{\gamma + \hat{\xi} - \hat{\xi}^2 - \theta_{trn}^2\|\hat{t}\|^2\|\hat{k}\|^2}{\gamma} = \frac{\hat{\xi}}{\gamma}.$$

Finally, the first two terms are nothing but

$$\frac{\hat{\xi}}{\gamma}\beta_*^T\theta_{tst}uv_{tst}^T = \frac{\hat{\xi}}{\gamma}\beta_*^T Z_{tst}.$$

Additionally, after substitutions, the last term can be simplified as

$$\hat{\varepsilon}_{trn}^T \left( \hat{A}_{trn}^\dagger + \frac{\theta_{trn}}{\hat{\xi}}\hat{t}^T\hat{k}^T\hat{A}_{trn}^\dagger - \frac{\hat{\xi}}{\gamma}\hat{p}\left( -\frac{\theta_{trn}\|\hat{t}\|^2}{\hat{\xi}}\hat{h}^T\hat{A}_{trn}^\dagger - \hat{h} \right) \right) Z_{tst} \qquad (\star)$$

$$= \theta_{tst}\hat{\varepsilon}_{trn}^T \left( \hat{A}_{trn}^\dagger uv_{tst}^T + \frac{\theta_{trn}}{\hat{\xi}}\hat{t}^T\hat{k}^T\hat{A}_{trn}^\dagger uv_{tst}^T + \frac{\hat{\xi}}{\gamma}\hat{p}\left( \frac{\theta_{trn}\|\hat{t}\|^2}{\hat{\xi}}\hat{k}^T\hat{A}_{trn}^\dagger u + \hat{h}u \right) v_{tst}^T \right).$$

Since $\hat{k} = \hat{A}_{trn}^\dagger u$ and $\hat{h}u = \hat{v}_{trn}^T\hat{A}_{trn}^\dagger u = \frac{\hat{\xi}-1}{\theta_{trn}}$, we then have that

$$(\star) = \theta_{tst}\hat{\varepsilon}_{trn}^T \left( \hat{k}v_{tst}^T + \frac{\theta_{trn}\|\hat{k}\|^2}{\hat{\xi}}\hat{t}^Tv_{tst}^T + \frac{\hat{\xi}}{\gamma}\hat{p}\left( \frac{\theta_{trn}\|\hat{t}\|^2\|\hat{k}\|^2}{\hat{\xi}} + \frac{\hat{\xi}-1}{\theta_{trn}} \right) v_{tst}^T \right)$$

$$= \theta_{tst}\hat{\varepsilon}_{trn}^T \left( \hat{k}v_{tst}^T + \frac{\theta_{trn}\|\hat{k}\|^2}{\hat{\xi}}\hat{t}^Tv_{tst}^T + \frac{\hat{\xi}}{\gamma}\hat{p}\left( \frac{\theta_{trn}^2\|\hat{t}\|^2\|\hat{k}\|^2 + \hat{\xi}^2 - \hat{\xi}}{\hat{\xi}\theta_{trn}} \right) v_{tst}^T \right)$$

$$= \theta_{tst}\hat{\varepsilon}_{trn}^T \left( \hat{k}v_{tst}^T + \frac{\theta_{trn}\|\hat{k}\|^2}{\hat{\xi}}\hat{t}^Tv_{tst}^T + \frac{1}{\gamma}\hat{p}\left( \frac{\gamma - \hat{\xi}}{\theta_{trn}} \right) v_{tst}^T \right)$$

$$= \theta_{tst}\hat{\varepsilon}_{trn}^T \left( \frac{1}{\theta_{trn}}\left( \frac{\theta_{trn}^2\|\hat{k}\|^2}{\hat{\xi}}\hat{t}^T + \theta_{trn}\hat{k} \right) v_{tst}^T + \frac{1}{\theta_{trn}}\hat{p}v_{tst}^T - \frac{\hat{\xi}}{\theta_{trn}\gamma}\hat{p}v_{tst}^T \right)$$

$$= \hat{\varepsilon}_{trn}^T \left( -\frac{\theta_{tst}}{\theta_{trn}}\hat{p}v_{tst}^T + \frac{\theta_{tst}}{\theta_{trn}}\hat{p}v_{tst}^T - \frac{\theta_{tst}\hat{\xi}}{\theta_{trn}\gamma}\hat{p}v_{tst}^T \right) = -\frac{\theta_{tst}\hat{\xi}}{\theta_{trn}\gamma}\hat{\varepsilon}_{trn}^T\hat{p}v_{tst}^T,$$

where we recall the expression of $\hat{p}$ for the second to last equality. We then obtain the result. $\qquad\square$

**Lemma 4.** *If $A_{tst}$ has independent entries of mean 0 and variance $\tau_{A_{tst}}^2/d$, then* $\mathbb{E}_{A_{tst}}[\|\beta_{so}^T A_{tst}\|_F^2] = \frac{\tau_{A_{tst}}^2 n_{tst}}{d}\|\beta_{so}\|_F^2.$

*Proof.* Consider $\tilde{A}_{tst} = \frac{1}{\tau_{A_{tst}}}A_{tst}$, which has entries with variance $1/d$. We have that

$$\mathbb{E}_{A_{tst}}[\|\beta_{so}^T A_{tst}\|^2] = \tau_{A_{tst}}^2 \mathbb{E}_{A_{tst}}[\|\beta_{so}^T\tilde{A}_{tst}\|^2] = \frac{\tau_{A_{tst}}^2 n_{tst}}{d}\|\beta_{so}\|_F^2.$$

The last equality directly follows from Lemma 3 in Sonthalia & Nadakuditi (2023). $\qquad\square$

**Lemma 5.** *In the above setting,*

$$\|\beta_{so}^T\|_F^2 = (\beta_*^T u)^2 \|\tilde{\beta}\|_F^2 + 2\beta_*^T \tilde{W}_{opt}^T (\hat{Z}_{trn} + \hat{A}_{trn})^{\dagger T} \hat{\varepsilon}_{trn}$$
$$+ \hat{\varepsilon}_{trn}^T (\hat{Z}_{trn} + \hat{A}_{trn})^{\dagger} (\hat{Z}_{trn} + \hat{A}_{trn})^{\dagger T} \hat{\varepsilon}_{trn},$$

*where* $\tilde{\beta}^T = \hat{Z}_{trn}(\hat{Z}_{trn} + \hat{A}_{trn})^{\dagger}$, *the optimal solution to the rank 1 denoising problem* $\arg\min_{\beta_*^T} \|\hat{Z}_{trn} - \beta_*^T (\hat{Z}_{trn} + \hat{A}_{trn})\|_F^2$.

*Proof.* A direct expansion of $\|\beta_{so}\|_F^2$ yields

$$\|\beta_{so}^T\|_F^2 = (\beta_*^T \hat{Z}_{trn} + \hat{\varepsilon}_{trn}^T)(\hat{Z}_{trn} + \hat{A}_{trn})^{\dagger} (\hat{Z}_{trn} + \hat{A}_{trn})^{\dagger T} (\beta_*^T \hat{Z}_{trn} + \hat{\varepsilon}_{trn}^T)^T$$
$$= \beta_*^T \hat{Z}_{trn} (\hat{Z}_{trn} + \hat{A}_{trn})^{\dagger} (\hat{Z}_{trn} + \hat{A}_{trn})^{\dagger T} \hat{Z}_{trn}^T \beta_*$$
$$+ 2\beta_*^T \hat{Z}_{trn} (\hat{Z}_{trn} + \hat{A}_{trn})^{\dagger} (\hat{Z}_{trn} + \hat{A}_{trn})^{\dagger T} \hat{\varepsilon}_{trn}$$
$$+ \hat{\varepsilon}_{trn}^T (\hat{Z}_{trn} + \hat{A}_{trn})^{\dagger} (\hat{Z}_{trn} + \hat{A}_{trn})^{\dagger T} \hat{\varepsilon}_{trn}.$$

Using $\tilde{\beta}^T = \hat{Z}_{trn}(\hat{Z}_{trn} + \hat{A}_{trn})^{\dagger}$ and $\hat{Z}_{trn} = \theta_{trn} u \hat{v}_{trn}$, we have that

$$\beta_*^T \hat{Z}_{trn}(\hat{Z}_{trn} + \hat{A}_{trn})^{\dagger} (\hat{Z}_{trn} + \hat{A}_{trn})^{\dagger T} \hat{Z}_{trn}^T W$$

$$= \beta_*^T u \operatorname{Tr}\left( \theta_{trn}^2 \hat{v}_{trn}^T (\hat{Z}_{trn} + \hat{A}_{trn})^{\dagger} (\hat{Z}_{trn} + \hat{A}_{trn})^{\dagger T} \hat{v}_{trn} \right) u^T W$$

$$= \beta_*^T u \operatorname{Tr}\left( \underbrace{\theta_{trn} u \hat{v}_{trn}^T}_{\hat{Z}_{trn}} (\hat{Z}_{trn} + \hat{A}_{trn})^{\dagger} (\hat{Z}_{trn} + \hat{A}_{trn})^{\dagger T} \underbrace{\theta_{trn} \hat{v}_{trn} u^T}_{\hat{Z}_{trn}} \right) u^T W$$

$$= (\beta_*^T u)^2 \|\tilde{\beta}\|_F^2,$$

where the second to last equality is since $u$ is a unit vector, and inserting it on both sides of the trace does not change the value. The other two terms follow. $\qquad\square$

Note that these expressions in Lemma 4, 5 can be expanded even further. We will come back to them once we have the necessary expectations in the next step.

### A.3   STEP 3: COMPUTE EXPECTATIONS OF IMPORTANT TERMS

Now we leverage techniques from random matrix theory to establish the following lemmas.

**Lemma 6** (Li & Sonthalia (2024)). *Let* $A \in \mathbb{R}^{d \times n}$ *and* $\hat{A} = [A_{trn} \quad \mu I] \in \mathbb{R}^{d \times (n+d)}$. *Suppose* $A = U\Sigma V^T$ *and* $\hat{A} = \hat{U}\hat{\Sigma}\hat{V}^T$ *are the respective singular value decompositions, then* $\hat{U} = U$, *and*

*(a) If* $d < n$ *(underparameterized regime),*

$$\hat{\Sigma} = \begin{bmatrix} \sqrt{\sigma_1(A)^2 + \mu^2} & 0 & \cdots & 0 \\ 0 & \sqrt{\sigma_2(A)^2 + \mu^2} & & 0 \\ \vdots & & \ddots & \vdots \\ 0 & 0 & \cdots & \sqrt{\sigma_d(A)^2 + \mu^2} \end{bmatrix} \in \mathbb{R}^{d \times d},$$

*and*

$$\hat{V} = \begin{bmatrix} V_{1:d} \Sigma \hat{\Sigma}^{-1} \\ \mu U \hat{\Sigma}^{-1} \end{bmatrix} \in \mathbb{R}^{(d+n) \times n}.$$

*(b) If* $d > n$ *(overparameterized regime),*

$$\hat{\Sigma} = \begin{bmatrix} \sqrt{\sigma_1(A)^2 + \mu^2} & 0 & \cdots & 0 & & \cdots & 0 \\ 0 & \sqrt{\sigma_2(A)^2 + \mu^2} & & 0 & & & \\ \vdots & & \ddots & \vdots & & & \vdots \\ 0 & 0 & \cdots & \sqrt{\sigma_n(A)^2 + \mu^2} & & & 0 \\ & & & & \mu & & \\ \vdots & & & & & \ddots & 0 \\ 0 & 0 & \cdots & 0 & \cdots & 0 & \mu \end{bmatrix} \in \mathbb{R}^{d \times d},$$

*and*

$$\hat{V} = \begin{bmatrix} V\Sigma_{1:n,1:n}^T C^{-1} & 0 \\ \mu U_{1:n} C^{-1} & U_{(n+1):d} \end{bmatrix} \in \mathbb{R}^{(d+n)\times d},$$

*where $C$ is the upper left $n \times n$ submatrix of $\hat{\Sigma}$.*

The following Lemmas are in Li & Sonthalia (2024) for the case when $\tau^2 = 1$. We need the lemmas for general $\tau^2$ and so we present them here. The proofs are very similar to the proof in Li & Sonthalia (2024) with the appropriate rescaling.

**Lemma 7.** *Suppose $A \in \mathbb{R}^{d\times n}$ such that $d < n$, where the entries of $A$ are independent and have mean 0, variance $\tau^2/d$, and bounded fourth moment. Let $c = d/n$, $\hat{A} = [A \quad \mu I] \in \mathbb{R}^{d\times(n+d)}$, $W_d = \hat{A}\hat{A}^T$, and $W_n = \hat{A}^T\hat{A}$. Suppose $\lambda_d$ is a random non-zero eigenvalue of $W_d$, and $\lambda_n$ is a random non-zero eigenvalue of the largest $n$ eigenvalues of $W_n$. Then*

*(i)* $\mathbb{E}\left[\frac{1}{\lambda_d}\right] = \mathbb{E}\left[\frac{1}{\lambda_n}\right] = \frac{\sqrt{(\tau^2+\mu^2 c - c\tau^2)^2 + 4\mu^2 c^2\tau^2} - \tau^2 - \mu^2 c + c\tau^2}{2\mu^2\tau^2 c} + o(1/\tau^2).$

*(ii)* $\mathbb{E}\left[\frac{1}{\lambda_d^2}\right] = \mathbb{E}\left[\frac{1}{\lambda_n^2}\right] = \frac{\mu^2 c^2 + \mu^2 c + (c-1)^2\tau^2}{2\mu^4 c\sqrt{(\tau^2+\mu^2 c - c\tau^2)^2 + 4\mu^2 c^2\tau^2}} + \frac{1}{2\mu^4}\left(1 - \frac{1}{c}\right) + o(1/\tau^2).$

*(iii)* $\mathbb{E}\left[\frac{1}{\lambda_d^3}\right] = \mathbb{E}\left[\frac{1}{\lambda_n^3}\right] = \frac{c^3}{2\tau^6}m_c''(-\frac{c\mu^2}{\tau^2}) + o(1/\tau^2),$

*where $m_c(z) = -\frac{1-z-c-\sqrt{(1-z-c)^2-4cz}}{-2zc}$ is the Stieltjes transform.*

*Proof.* First, it is trivial to see that $1/\lambda_d = 1/\lambda_n$ in expectation since $W_d$ and $W_n$ share the same set of eigenvalues from which we sample. Here we consider $W_d$ and define $\tilde{\mu} = \mu/\tau$, $\tilde{A} = A/\tau$. Note $\tilde{A}$ then has entries with mean 0 and variance $1/d$.

By the definition of $W_d$, $\frac{c}{\tau^2}W_d$ is the correct normalization to turn it into a Wishart matrix. Also, by assumptions on $A$, the eigenvalues of $\frac{c}{\tau^2}AA^T = c\tilde{A}\tilde{A}^T$ converge to the Marchenko-Pastur distribution with shape c. With these results, we have that

$$cW_d = c\begin{bmatrix} A & \mu I \end{bmatrix}\begin{bmatrix} A^T \\ \mu I \end{bmatrix} = cAA^T + c\mu^2 I$$

$$\rightarrow (c\lambda_d)_i = c\sigma_i(A)^2 + c\mu^2$$

$$\rightarrow (c\lambda_d)_i = c\tau^2\sigma_i(\tilde{A})^2 + c\tau^2\tilde{\mu}^2$$

$$\rightarrow \left(\frac{c\lambda_d}{\tau^2}\right)_i = c\sigma_i(\tilde{A})^2 + c\tilde{\mu}^2.$$

The rest of the proof follows the same fashion as in Li & Sonthalia (2024), with additional care on the general variances. We provide a sketch here: we consider the Stieltjes transform for computing the expectation of inversed eigenvalues, which is given by

$$m_c(z) = \mathbb{E}_\lambda\left[\frac{1}{\lambda - z}\right] = -\frac{1-z-c-\sqrt{(1-z-c)^2-4cz}}{-2zc}.$$

We plug in $z = -c\tilde{\mu}^2$ to obtain the needed result,

$$\mathbb{E}\left[\frac{\tau^2}{c\lambda_d}\right] = m_c(-c\tilde{\mu}^2) \longrightarrow \mathbb{E}\left[\frac{1}{\lambda_d}\right] = \frac{c}{\tau^2}m_c(-c\tilde{\mu}^2).$$

Simplifying and plugging in $\tilde{\mu} = \mu/\tau$, we have that

$$\mathbb{E}\left[\frac{1}{\lambda_d}\right] = \mathbb{E}\left[\frac{1}{\lambda + \mu^2}\right] = \frac{\sqrt{(\tau^2 + \mu^2 c - c\tau^2)^2 + 4\mu^2 c^2\tau^2} - \tau^2 - \mu^2 c + c\tau^2}{2\mu^2\tau^2 c}.$$

To get expectations for the squared and cubed inverse, we need to compute the derivatives of $m_c(z)$:

$$m_c'(z) = \mathbb{E}_\lambda\left[\frac{1}{(\lambda - z)^2}\right] = \frac{(c - z + \sqrt{-4cz + (1-c-z)^2} - 1)(c + z + \sqrt{-4cz + (1-c-z)^2} - 1)}{4cz^2\sqrt{-4cz + (1-c-z)^2}}.$$

$$m_c''(z) = \mathbb{E}_\lambda \left[ \frac{2}{(\lambda - z)^3} \right] = \frac{z(c+1)(z^2 + 3(c-1)^2) - 3z^2(c^2+1) - (c-1)^4}{cz^3(-4cz + (1-c-z)^2)^{3/2}}$$
$$+ \frac{(c-1)(2z(c+1) - z^2 - (c-1)^2)}{cz^3(-4cz + (1-c-z)^2)}.$$

Then we have

$$\mathbb{E} \left[ \frac{\tau^4}{c^2 \lambda_d^2} \right] = m_c'(-c\tilde{\mu}^2) \longrightarrow \mathbb{E} \left[ \frac{1}{\lambda_d^2} \right] = \frac{c^2}{\tau^4} m_c' \left( -\frac{c\mu^2}{\tau^2} \right),$$
$$\mathbb{E} \left[ \frac{2\tau^6}{c^3 \lambda_d^3} \right] = m_c''(-c\tilde{\mu}^2) \longrightarrow \mathbb{E} \left[ \frac{1}{\lambda_d^3} \right] = \frac{c^3}{2\tau^6} m_c'' \left( -\frac{c\mu^2}{\tau^2} \right).$$

Similarly, we simplify these results to get the conclusion. Note for $\mathbb{E} \left[ \frac{1}{\lambda_d^3} \right]$, the formula becomes extremely complicated. Hence, we only provide a heuristic formula in the lemma statement and use Sympy to simplify further computations when needed. $\qquad \square$

Next we have similar Lemma for $c > 1$.

**Lemma 8.** *Suppose $A \in \mathbb{R}^{d \times n}$ such that $d > n$, where the entries of $A$ are independent and have mean 0, variance $\tau^2/d$, and bounded fourth moment. Let $c = d/n$, $\hat{A} = [A \quad \mu I] \in \mathbb{R}^{d \times (d+n)}$, $W_d = \hat{A}\hat{A}^T$, and $W_n = \hat{A}^T \hat{A}$. Suppose $\lambda_n$ is a random non-zero eigenvalue of $W_n$, and $\lambda_d$ is a random non-zero eigenvalue of the largest $d$ eigenvalues of $W_d$. Then*

*(i)* $\mathbb{E} \left[ \frac{1}{\lambda_d} \right] = \mathbb{E} \left[ \frac{1}{\lambda_n} \right] = \frac{\sqrt{(-\tau^2 + \mu^2 c + c\tau^2)^2 + 4\mu^2 c\tau^2} - \tau^2 - \mu^2 c + c\tau^2}{2\mu^2 \tau^2} + o(1/\tau^2).$

*(ii)* $\mathbb{E} \left[ \frac{1}{\lambda_d^2} \right] = \mathbb{E} \left[ \frac{1}{\lambda_n^2} \right] = \frac{\mu^2 c^2 + \mu^2 c + (c-1)^2 \tau^2}{2\mu^4 \sqrt{(-\tau^2 + \mu^2 c + c\tau^2)^2 + 4\mu^2 c\tau^2}} + \frac{1}{2\mu^4}(1-c) + o(1/\tau^2).$

*(iii)* $\mathbb{E} \left[ \frac{1}{\lambda_d^3} \right] = \mathbb{E} \left[ \frac{1}{\lambda_n^3} \right] = \frac{1}{2\tau^6} m_{1/c}''(-\frac{u^2}{\tau^2}) + o(1/\tau^2),$

*where $m_{1/c}(z) = -\frac{1-z-1/c-\sqrt{(1-z-1/c)^2 - 4z/c}}{-2z/c}$ is the Stieltjes transform.*

*Proof.* The proof is analogous to the $c < 1$ case. We consider $W_n$ and define $\tilde{\mu} = \mu/\tau$, $\tilde{A} = A/\tau$. By assumptions on $A$, the eigenvalues of $\frac{1}{\tau^2} A^T A = \tilde{A}^T \tilde{A}$ converge to the Marchenko-Pastur distribution with shape $1/c$, and

$$(\lambda_n)_i = \sigma_i(A)^2 + \mu^2$$
$$\rightarrow (\lambda_n)_i = \tau^2 \sigma_i(\tilde{A})^2 + \tau^2 \tilde{\mu}^2$$
$$\rightarrow \left( \frac{\lambda_n}{\tau^2} \right)_i = \sigma_i(\tilde{A})^2 + \tilde{\mu}^2.$$

The Stieltjes transform becomes

$$m_{1/c}(z) = -\frac{1 - z - 1/c - \sqrt{(1-z-1/c)^2 - 4z/c}}{-2z/c}. \qquad (A.6)$$

Similar to Lemma 7, we need to plug in $z = -\tilde{\mu}^2$ here and compute necessary derivatives:

$$\mathbb{E} \left[ \frac{\tau^2}{\lambda_n} \right] = m_{1/c}(-\tilde{\mu}^2) \longrightarrow \mathbb{E} \left[ \frac{1}{\lambda_n} \right] = \frac{1}{\tau^2} m_{1/c}(-\tilde{\mu}^2) = \frac{1}{\tau^2} m_{1/c} \left( -\frac{\mu^2}{\tau^2} \right).$$

$$\mathbb{E} \left[ \frac{\tau^4}{\lambda_n^2} \right] = m_{1/c}'(-\tilde{\mu}^2) \longrightarrow \mathbb{E} \left[ \frac{1}{\lambda_n^2} \right] = \frac{1}{\tau^4} m_{1/c}' \left( -\frac{\mu^2}{\tau^2} \right).$$

$$\mathbb{E} \left[ \frac{2\tau^6}{\lambda_n^3} \right] = m_{1/c}''(-\tilde{\mu}^2) \longrightarrow \mathbb{E} \left[ \frac{1}{\lambda_n^3} \right] = \frac{1}{2\tau^6} m_{1/c}'' \left( -\frac{\mu^2}{\tau^2} \right).$$

We simplify these terms to get the results. Again we skip the full formula for the cubed inverse. $\square$

Finally, we shall need the following estimates as well.

**Lemma 9.** *Suppose $A \in \mathbb{R}^{d \times n}$, where the entries of $A$ are independent and have mean 0, variance $\tau^2/d$, and bounded fourth moment. Let $c = d/n$. Suppose $\lambda$ is a random eigenvalue of $A$. Then*

*(i) If $d > n$, $\mathbb{E}\left[\frac{\lambda}{\lambda+\mu^2}\right] = c\left(\frac{1}{2} + \frac{\tau^2+\mu^2 c - \sqrt{(-\tau^2+\mu^2 c+c\tau^2)^2+4\mu^2 c\tau^2}}{2c\tau^2}\right) + o(1/\tau^2).$*

*(ii) If $d < n$, $\mathbb{E}\left[\frac{\lambda}{\lambda+\mu^2}\right] = \frac{1}{2} + \frac{\tau^2+\mu^2 c - \sqrt{(\tau^2+\mu^2 c - c\tau^2)^2+4\mu^2 c^2\tau^2}}{2c\tau^2} + o(1/\tau^2).$*

*(iii) If $d > n$, $\mathbb{E}\left[\frac{\lambda}{(\lambda+\mu^2)^2}\right] = c\left(\frac{\tau^2+c\tau^2+\mu^2 c}{2\tau^2\sqrt{(-\tau^2+\mu^2 c+c\tau^2)^2+4\mu^2 c\tau^2}} - \frac{1}{2\tau^2}\right) + o(1/\tau^2).$*

*(iv) If $d < n$, $\mathbb{E}\left[\frac{\lambda}{(\lambda+\mu^2)^2}\right] = \frac{\tau^2+c\tau^2+\mu^2 c}{2\tau^2\sqrt{(\tau^2+\mu^2 c - c\tau^2)^2+4\mu^2 c^2\tau^2}} - \frac{1}{2\tau^2} + o(1/\tau^2).$*

*(v) If $d > n$, $\mathbb{E}\left[\frac{\lambda^2}{(\lambda+\mu^2)^2}\right] = \frac{c\tau^2+\tau^2+2\mu^2 c}{2\tau^2} - \frac{c^2\tau^4+3c^2\tau^2\mu^2-2c^2\mu^4+2c\tau^4+3c\tau^2\mu^2-\tau^4}{2\tau^2\sqrt{(-\tau^2+\mu^2 c+c\tau^2)^2+4\mu^2 c\tau^2}} + o(1/\tau^2).$*

*(vi) If $d < n$, $\mathbb{E}\left[\frac{\lambda^2}{(\lambda+\mu^2)^2}\right] = \frac{c\tau^2+\tau^2+2\mu^2 c}{2c\tau^2} - \frac{c^2\tau^4+3c^2\tau^2\mu^2-2c^2\mu^4+2c\tau^4+3c\tau^2\mu^2-\tau^4}{2c\tau^2\sqrt{(\tau^2+\mu^2 c - c\tau^2)^2+4\mu^2 c^2\tau^2}} + o(1/\tau^2).$*

*Proof.* The results immediately follow from Lemmas 7, 8 by

$$\mathbb{E}\left[\frac{\lambda}{\lambda+\mu^2}\right] = 1 - \mu^2 \mathbb{E}\left[\frac{1}{\lambda+\mu^2}\right] = 1 - \mu^2 \mathbb{E}\left[\frac{1}{\lambda_d}\right],$$

$$\mathbb{E}\left[\frac{\lambda}{(\lambda+\mu^2)^2}\right] = \mathbb{E}\left[\frac{1}{\lambda+\mu^2}\right] - \mu^2 \mathbb{E}\left[\frac{1}{(\lambda+\mu^2)^2}\right],$$

$$\mathbb{E}\left[\frac{\lambda^2}{(\lambda+\mu^2)^2}\right] = \mathbb{E}\left[\frac{\lambda}{\lambda+\mu^2}\right] - \mu^2 \mathbb{E}\left[\frac{\lambda}{(\lambda+\mu^2)^2}\right].$$

$\square$

**Remark 1.** *We can also evaluate the following expectations:*

$$\mathbb{E}\left[\frac{\lambda}{(\lambda+\mu^2)^3}\right] = \mathbb{E}\left[\frac{1}{(\lambda+\mu^2)^2}\right] - \mu^2 \mathbb{E}\left[\frac{1}{(\lambda+\mu^2)^3}\right],$$

$$\mathbb{E}\left[\frac{\lambda^2}{(\lambda+\mu^2)^3}\right] = \mathbb{E}\left[\frac{\lambda}{(\lambda+\mu^2)^2}\right] - \mu^2 \mathbb{E}\left[\frac{\lambda}{(\lambda+\mu^2)^3}\right].$$

*However, they are too complicated to be presented here and are not always useful. We will use Sympy when these terms show up.*

**A note on bounded variances:** Previous works in Li & Sonthalia (2024), Sonthalia & Nadakuditi (2023), and Kausik et al. (2024) have established proofs that bound variances of terms present in our $\beta_{so}^T$ formula, which implies that their variances asymptotically decay to 0. In our setting, since the variance parameters $\tau_A$ is at most $O(n)$ and we normalize by $\tau_A$ to get the appropriate limits. However this means that when $\tau_A$ grows we actually get faster convergence. $\tau_\varepsilon$ has finite value. Hence they only induce a multiplicative change in the total variance of terms and do not affect the asymptotic decaying phenomena. In other words, these terms are still highly concentrated, and we can treat them as almost independent when $d, n_{trn} \to \infty$. A direct consequence of this is that we can compute the expectation of a product as the product of its individual expectations.

### A.4 Step 4: Estimate Quantities Using Random Matrix Estimates

The following lemmas compute the mean and variance of terms in the $\beta_{so}^T$ formula. The proofs are similar to Lemmas 13-18 in Li & Sonthalia (2024); we repeat it for Lemma 10 and provide a sketch for the rest.

**Lemma 10.** *Under our assumptions, we have that*

$$
\mathbb{E}_{A_{trn}}\left[\|\hat{h}\|^2\right] = \begin{cases} c\left(\dfrac{\tau_{A_{trn}}^2 + c\tau_{A_{trn}}^2 + \mu^2 c}{2\tau_{A_{trn}}^2\sqrt{\left(\tau_{A_{trn}}^2 + \mu^2 c - c\tau_{A_{trn}}^2\right)^2 + 4\mu^2 c^2 \tau_{A_{trn}}^2}} - \dfrac{1}{2\tau_{A_{trn}}^2}\right) + o(1/\tau_{A_{trn}}^2) & c < 1 \\[4ex] c\left(\dfrac{\tau_{A_{trn}}^2 + c\tau_{A_{trn}}^2 + \mu^2 c}{2\tau_{A_{trn}}^2\sqrt{\left(-\tau_{A_{trn}}^2 + \mu^2 c + c\tau_{A_{trn}}^2\right)^2 + 4\mu^2 c \tau_{A_{trn}}^2}} - \dfrac{1}{2\tau_{A_{trn}}^2}\right) + o(1/\tau_{A_{trn}}^2) & c > 1 \end{cases}
$$

*and* $Var(\|\hat{h}\|^2) = o(1/\tau_{A_{trn}}^2)$.

*Proof.* Recall that $\hat{h} = \hat{v}_{trn}^T \hat{A}_{trn}^\dagger$, where $\hat{v}_{trn} = [v_{trn}\ \mathbf{0}_d] \in \mathbb{R}^{n_{trn}+d}$, with $v_{trn} \in \mathbb{R}^{n_{trn}}$ being a unit vector. We aim to compute $\mathbb{E}_{A_{trn}}[\|\hat{h}\|^2]$. First, consider the singular value decomposition (SVD) of $\hat{A}_{trn}$:

$$\hat{A}_{trn} = U\hat{\Sigma}\hat{V}^T,$$

where $U \in \mathbb{R}^{d \times d}$ is orthogonal, $\hat{\Sigma} \in \mathbb{R}^{d \times d}$ is diagonal with non-negative entries, and $\hat{V} \in \mathbb{R}^{(n_{trn}+d) \times d}$ has orthonormal columns. Then, the pseudoinverse of $\hat{A}_{trn}$ is given by

$$\hat{A}_{trn}^\dagger = \hat{V}\hat{\Sigma}^{-1}U^T$$

and

$$\hat{h} = \hat{v}_{trn}^T \hat{A}_{trn}^\dagger.$$

Therefore, we have

$$\|\hat{h}\|^2 = \hat{h}\hat{h}^T = \hat{v}_{trn}^T \hat{A}_{trn}^\dagger \hat{A}_{trn}^{\dagger T}\hat{v}_{trn} = \hat{v}_{trn}^T \hat{V}\hat{\Sigma}^{-2}\hat{V}^T \hat{v}_{trn}.$$

Assume $c < 1$. We can partition $\hat{V}$ and $\hat{v}_{trn}$ to reflect the structure of $\hat{A}_{trn}$. Using Lemma 6 let us write $\hat{V}$ as

$$\hat{V} = \begin{bmatrix} V_{1:d}\Sigma\hat{\Sigma}^{-1} \\ \mu U\hat{\Sigma}^{-1} \end{bmatrix}.$$

Since the last $d$ elements of $\hat{v}_{trn}$ are 0, we get that

$$\hat{v}_{trn}^T \hat{V} = v_{trn}^T V_{1:d}\Sigma\hat{\Sigma}^{-1}.$$

Thus, we see that

$$
\begin{aligned}
\mathbb{E}\|\hat{h}\|^2 &= v_{trn}^T V_{1:d}\Sigma\hat{\Sigma}^{-4}\Sigma V_{1:d}^T v_{trn} \\
&= \sum_{i=1}^{d}(v_{trn}^T V_{1:d})_i^2 \frac{\lambda_i}{(\lambda_i + \mu^2)^2}.
\end{aligned}
$$

Note $v_{trn}^T V_{1:d}$ is a uniformly random unit vector in $\mathbb{R}^{n_{trn}}$ by the rotational bi-invariance assumption on $A_{trn}$. Thus, when we take expectations, this becomes $1/n_{trn}$. We then see that

$$\mathbb{E}\left[\|\hat{h}\|^2\right] = \mathbb{E}\left[\sum_{i=1}^{d}\frac{1}{n_{trn}}\frac{\lambda_i}{(\lambda_i + \mu^2)^2}\right]$$

The term inside the expectation is another expectation and we can use weak convergence. Thus, in expectation, this term by Lemma 7 becomes

$$\mathbb{E}_{A_{trn}}\left[\frac{\lambda}{(\lambda + \mu^2)^2}\right] = c\left(\frac{\tau_{A_{trn}}^2 + c\tau_{A_{trn}}^2 + \mu^2 c}{2\tau_{A_{trn}}^2\sqrt{\left(\tau_{A_{trn}}^2 + \mu^2 c - c\tau_{A_{trn}}^2\right)^2 + 4\mu^2 c^2 \tau_{A_{trn}}^2}} - \frac{1}{2\tau_{A_{trn}}^2}\right) + o(1/\tau_{A_{trn}}^2),$$

where the additional factor of $c$ comes from projecting $d$ entries onto the $n_{trn}$ coordinates of the randomly uniform vector.

For $c > 1$, we use the corresponding SVD in Lemma 6 and the expectation in Lemma 8. We get

$$\mathbb{E}_{A_{trn}}\left[\frac{\lambda}{(\lambda+\mu^2)^2}\right] = c\left(\frac{\tau_{A_{trn}}^2 + c\tau_{A_{trn}}^2 + \mu^2 c}{2\tau_{A_{trn}}^2\sqrt{\left(-\tau_{A_{trn}}^2 + \mu^2 c + c\tau_{A_{trn}}^2\right)^2 + 4\mu^2 c\tau_{A_{trn}}^2}} - \frac{1}{2\tau_{A_{trn}}^2}\right) + o(1/\tau_{A_{trn}}^2).$$

$\square$

**Lemma 11.** *Under our assumptions, we have that*

$$\mathbb{E}_{A_{trn}}\left[\|\hat{k}\|^2\right] = \begin{cases} \frac{\sqrt{\left(\tau_{A_{trn}}^2 + \mu^2 c - c\tau_{A_{trn}}^2\right)^2 + 4\mu^2 c^2\tau_{A_{trn}}^2} - \tau_{A_{trn}}^2 - \mu^2 c + c\tau_{A_{trn}}^2}{2\mu^2\tau_{A_{trn}}^2} + o(1/\tau_{A_{trn}}^2) & c < 1 \\ \frac{\sqrt{\left(-\tau_{A_{trn}}^2 + \mu^2 c + c\tau_{A_{trn}}^2\right)^2 + 4\mu^2 c\tau_{A_{trn}}^2} - \tau_{A_{trn}}^2 - \mu^2 c + c\tau^2}{2\mu^2\tau_{A_{trn}}^2} + o(1/\tau_{A_{trn}}^2) & c > 1 \end{cases}$$

*and* $Var(\|\hat{k}\|^2) = o(1/\tau_{A_{trn}}^2)$.

*Proof.* (Sketch) Recall that $k = \hat{A}_{trn}^\dagger u$. Using SVD of $\hat{A}_{trn}$ and a similar argument, we have

$$\mathbb{E}_{A_{trn}}\left[u^T\hat{A}_{trn}^{\dagger T}\hat{A}_{trn}^\dagger u\right] = \begin{cases} \mathbb{E}_{A_{trn}}\left[\frac{1}{\lambda+\mu^2}\right] & c < 1 \\ \frac{1}{c}\mathbb{E}_{A_{trn}}\left[\frac{1}{\lambda+\mu^2}\right] + \left(1 - \frac{1}{c}\right)\frac{1}{\mu^2} & c > 1 \end{cases}$$

where for $c > 1$, $1/c$ of the eigenvalues follow the expectation and the rest equals $1/\mu^2$. $\square$

**Lemma 12.** *Under our assumptions, we have that*

$$\mathbb{E}_{A_{trn}}\left[\|\hat{t}\|^2\right] = \begin{cases} \frac{1}{2\tau_{A_{trn}}^2}\left(\tau_{A_{trn}}^2 - c\tau_{A_{trn}}^2 - \mu^2 c + \sqrt{\left(\tau_{A_{trn}}^2 - c\tau_{A_{trn}}^2 + \mu^2 c\right)^2 + 4c^2\mu^2\tau_{A_{trn}}^2}\right) + o(1/\tau_{A_{trn}}^2) & c < 1 \\ \frac{1}{2\tau_{A_{trn}}^2}\left(-\tau_{A_{trn}}^2 + c\tau_{A_{trn}}^2 - \mu^2 c + \sqrt{\left(-\tau_{A_{trn}}^2 + c\tau_{A_{trn}}^2 + \mu^2 c\right)^2 + 4c\mu^2\tau_{A_{trn}}^2}\right) + o(1/\tau_{A_{trn}}^2) & c > 1 \end{cases}$$

*and* $Var(\|\hat{t}\|^2) = o(1/\tau_{A_{trn}}^2)$.

*Proof.* (Sketch) Recall that $\hat{t} = \hat{v}_{trn}^T(I - \hat{A}_{trn}^\dagger\hat{A}_{trn})$. Since $(I - \hat{A}_{trn}^\dagger\hat{A}_{trn})$ is a projection matrix, we have $\|\hat{t}\|^2 = 1 - \hat{v}_{trn}^T\hat{A}_{trn}^\dagger\hat{A}_{trn}\hat{v}_{trn}$, and with SVD of $\hat{A}_{trn}$,

$$1 - \mathbb{E}_{A_{trn}}\left[\hat{v}_{trn}^T\hat{V}\hat{\Sigma}^{-2}\hat{V}^T\hat{v}_{trn}\right] = \begin{cases} 1 - c\mathbb{E}_{A_{trn}}\left[\frac{\lambda}{\lambda+\mu^2}\right] & c < 1 \\ 1 - \mathbb{E}_{A_{trn}}\left[\frac{\lambda}{\lambda+\mu^2}\right] & c > 1 \end{cases}.$$

$\square$

**Lemma 13.** *Under our assumptions, we have that* $\mathbb{E}_{A_{trn}}\left[\hat{\xi}\right] = 1$ *and* $Var(\hat{\xi}) = O(\theta_{trn}^2/(d\tau_{A_{trn}}^2))$.

*Proof.* Recall that $\hat{\xi} = 1 + \theta_{trn}\hat{v}_{trn}^T\hat{A}_{trn}^\dagger u$. Using the SVD of $\hat{A}_{trn}$, we have that

$$\mathbb{E}_{A_{trn}}\left[\hat{\xi}\right] = 1 + \mathbb{E}_{A_{trn}}\left[\theta_{trn}\hat{v}_{trn}^T\hat{V}\hat{\Sigma}U^T u\right] = 1.$$

because $U$ is a uniformly random orthogonal matrix, which makes $U^T u$ a uniformly random vector that is independent of $\hat{V}$ and $\hat{\Sigma}$. We similarly compute $\mathbb{E}_{A_{trn}}\left[\hat{\xi}^2\right]$ for the variance. $\square$

**Lemma 14.** *Under our assumptions, we have that*

$$\mathbb{E}_{A_{trn}}[\gamma] = \begin{cases} 1 + \frac{\theta_{trn}^2}{2\tau_{A_{trn}}^4}\left(\tau_{A_{trn}}^2 + c\tau_{A_{trn}}^2 + \mu^2 c - \sqrt{\left(\tau_{A_{trn}}^2 - c\tau_{A_{trn}}^2 + \mu^2 c\right)^2 + 4c^2\mu^2\tau_{A_{trn}}^2}\right) + o(1/\tau_{A_{trn}}^2) & c < 1 \\ 1 + \frac{\theta_{trn}^2}{2\tau_{A_{trn}}^4}\left(\tau_{A_{trn}}^2 + c\tau_{A_{trn}}^2 + \mu^2 c - \sqrt{\left(-\tau_{A_{trn}}^2 + c\tau_{A_{trn}}^2 + \mu^2 c\right)^2 + 4c\mu^2\tau_{A_{trn}}^2}\right) + o(1/\tau_{A_{trn}}^2) & c > 1 \end{cases}$$

*with* $Var(\gamma/\theta_{trn}^2) = o(1/(\tau_{A_{trn}}^2))$.

*Proof.* Recall that $\gamma = \theta_{trn}^2 \|\hat{t}\|^2 \|\hat{k}\|^2 + \hat{\xi}^2$. The expectation of the individual terms were computed in Lemmas 11, 12, and 13.

The difference between product of the expectations and the expectation of the product can be bounded by the square root of the product of the variances. Hence in the this case, we see that the product of the expectations has an error term of $o(1/\tau_{A_{trn}}^2)$ and the square root of the product of the variances is also $o(1/\tau_{A_{trn}}^2)$. Hence, we see that the error in the expectation is $o(1/\tau_{A_{trn}}^2)$.

We also need to compute the variance. The variance of the product of two dependent random variables $\mathcal{X}$ and $\mathcal{Y}$ is given by

$$Cov(\mathcal{X}^2, \mathcal{Y}^2) + \left[Var(\mathcal{X}) + \mathbb{E}[\mathcal{X}]^2\right]\left[Var(\mathcal{Y}) + \mathbb{E}[\mathcal{Y}]^2\right] - \left[Cov(\mathcal{X}, \mathcal{Y}) + \mathbb{E}[\mathcal{X}]\mathbb{E}[\mathcal{Y}]\right]^2.$$

Note that for both variances decay at order $o(1/\tau_{A_{trn}}^2)$ additionally, we see that the constant order term represented by $(\mathbb{E}[\mathcal{X}]\mathbb{E}[\mathcal{Y}])^2$ cancels out. Hence we see that

$$Var(\gamma) = \theta_{trn}^2 \left[Cov(\|\hat{t}\|^4, \|\hat{k}\|^4) + o(1/\tau_{A_{trn}}^2)\right].$$

Similar to before, we can compute and check that $Cov(\|\hat{t}\|^4, \|\hat{k}\|^4)$ is $o(1/\tau_{A_{trn}}^2)$. $\qquad\square$

**Lemma 15.** *Under our assumptions, we have that*

$$\mathbb{E}_{A_{trn}}\left[\hat{k}^T \hat{A}_{trn}^\dagger \hat{A}_{trn}^{\dagger T} \hat{k}\right] = \begin{cases} \frac{\mu^2 c^2 + \mu^2 c + (c-1)^2 \tau_{A_{trn}}^2}{2\mu^4 c \sqrt{\left(\tau_{A_{trn}}^2 + \mu^2 c - c\tau_{A_{trn}}^2\right)^2 + 4\mu^2 c^2 \tau_{A_{trn}}^2}} + \frac{1}{2\mu^4}\left(1 - \frac{1}{c}\right) + o(1/\tau_{A_{trn}}^2) & c < 1 \\[4mm] \frac{\mu^2 c^2 + \mu^2 c + (c-1)^2 \tau_{A_{trn}}^2}{2\mu^4 c \sqrt{\left(-\tau_{A_{trn}}^2 + \mu^2 c + c\tau_{A_{trn}}^2\right)^2 + 4\mu^2 c \tau_{A_{trn}}^2}} + \frac{1}{2\mu^4}\left(1 - \frac{1}{c}\right) + o(1/\tau_{A_{trn}}^2) & c > 1 \end{cases}$$

*and that* $Var(\hat{k}^T \hat{A}_{trn}^\dagger \hat{A}_{trn}^{\dagger T} \hat{k}) = o(1/\tau_{A_{trn}}^2)$.

*Proof.* (Sketch) Using $\hat{k} = \hat{A}_{trn}^\dagger u$ and the SVD, we have

$$\mathbb{E}_{A_{trn}}\left[u^T U \hat{\Sigma}^{-4} U^T u\right] = \begin{cases} \mathbb{E}_{A_{trn}}\left[\frac{1}{(\lambda+\mu^2)^2}\right] & c < 1 \\[3mm] \frac{1}{c}\mathbb{E}_{A_{trn}}\left[\frac{1}{(\lambda+\mu^2)^2}\right] + \left(1 - \frac{1}{c}\right)\frac{1}{\mu^4} & c > 1 \end{cases}$$

where for $c > 1$, $1/c$ of the eigenvalues follow the expectation and the rest equals $1/\mu^4$. $\qquad\square$

Details of the above expectations have been discussed in Li & Sonthalia (2024). The following lemmas establish expectations unique to this setting.

**Lemma 16.** *Suppose* $\varepsilon \in \mathbb{R}^n$ *whose entries have mean 0, variance* $\tau_\varepsilon$, *and follow our noise assumptions. Then for any random matrix* $Q \in \mathbb{R}^{n \times n}$ *independent, we have*

$$\mathbb{E}_{\varepsilon,Q}\left[\varepsilon^T Q \varepsilon\right] = \tau_\varepsilon^2 \mathbb{E}\left[\text{Tr}(Q)\right].$$

*Proof.* We have that

$$\varepsilon^T Q \varepsilon = \sum_{i=1}^n \sum_{j=1}^n \varepsilon_i \varepsilon_j q_{ij}.$$

We take the expectation of this sum. By the independence assumption and assumption $\mathbb{E}[\varepsilon_i \varepsilon_j] = 0$ when $i \neq j$, we then have

$$\mathbb{E}_{\varepsilon,Q}\left[\varepsilon^T Q \varepsilon\right] = \sum_{i=1}^n \mathbb{E}\left[\varepsilon_i^2\right] \mathbb{E}\left[q_{ij}\right] = \tau_\varepsilon^2 \mathbb{E}\left[\sum_{i=1}^n q_{ij}\right] = \tau_\varepsilon^2 \mathbb{E}\left[\text{Tr}(Q)\right].$$

$\qquad\square$

For the following Lemmas 17, 18, 19, 20, 21, 22 we need that variance with respect to $A_{trn}$ is bounded. We do not need it to decay. All of the expressions can be expressed as bounded functions of the non-zero eigenvalues of $A_{trn}$. Hence, due to weak convergence, they converge to some random variable on a compact measure space (the measure is the Marchenko-Pastur measure). Hence, these random variables have finite moments. Some of the variances do actually decay, but it is not too important.

**Lemma 17.** *Under our assumptions, the following terms have zero expectation w.r.t. $A_{trn}$ and $\varepsilon_{trn}$*
*$\forall c \in (0, \infty)$*

  *(i)* $\mathbb{E}_{A_{trn}} \left[ \hat{k}^T \hat{A}_{trn}^\dagger \hat{h}^T \right] = 0.$

  *(ii)* $\mathbb{E}_{A_{trn}} \left[ \hat{\varepsilon}_{trn}^T \hat{k} \hat{t} \hat{\varepsilon}_{trn} \right] = 0.$

  *(iii)* $\mathbb{E}_{A_{trn}} \left[ \hat{\varepsilon}_{trn}^T \hat{A}_{trm}^\dagger \hat{A}^{\dagger T} \hat{k} \hat{t} \hat{\varepsilon}_{trn} \right] = 0.$

  *(iv)* $\mathbb{E}_{A_{trn}} \left[ \hat{\varepsilon}_{trn}^T \hat{A}_{trn}^\dagger \hat{h}^T \hat{k} \hat{\varepsilon}_{trn} \right] = 0.$

*Proof.* The heuristics for this proof will be the following: if the term contains an odd number of a uniformly random vector centered around 0 (call it $a \in \mathbb{R}^N$) that is independent of the rest, then by matrix multiplication, the expectation can be written as

$$\sum_{i=1}^N \mathbb{E}\left[a_i^{2k+1}\right] \mathbb{E}\left[\text{other terms}\right] \text{ for some } k \in \mathbb{N}.$$

This becomes 0 since the expectation of an odd moment is 0 for a centered uniform distribution.

We use this idea to expand these 4 terms for $c < 1$:

  (i) The term $\hat{k}^T \hat{A}_{trn}^\dagger \hat{h}^T$ follows directly from Lemma 18 in Li & Sonthalia (2024).

  (ii) $\hat{\varepsilon}_{trn}^T \hat{k} \hat{t} \hat{\varepsilon}_{trn} = \hat{\varepsilon}_{trn}^T \hat{A}_{trn}^\dagger u \hat{v}_{trn}^T (I - \hat{A}_{trn}^\dagger \hat{A}_{trn}) \hat{\varepsilon}_{trn}.$

   Using the SVD $\hat{A} = U \hat{\Sigma} \hat{V}^T$ and the fact that the last d entries of $\hat{v}_{trn}, \hat{\varepsilon}_{trn}$ are 0, we have

   $\hat{\varepsilon}_{trn}^T \hat{V} \hat{\Sigma}^{-1} U^T u \hat{v}_{trn}^T (I - \hat{V} \hat{V}^T) \hat{\varepsilon}_{trn}$

   $= \begin{bmatrix} \varepsilon_{trn}^T & 0_d^T \end{bmatrix} \begin{bmatrix} V_{1:d} \Sigma \hat{\Sigma}^{-1} \\ \mu U \hat{\Sigma}^{-1} \end{bmatrix} \hat{\Sigma}^{-1} U^T u \begin{bmatrix} v_{trn}^T & 0_d^T \end{bmatrix} \left( I - \begin{bmatrix} V_{1:d} \Sigma \hat{\Sigma}^{-1} \\ \mu U \hat{\Sigma}^{-1} \end{bmatrix} \begin{bmatrix} \hat{\Sigma}^{-1} \Sigma V_{1:d}^T & \mu \hat{\Sigma}^{-1} U^T \end{bmatrix} \right) \begin{bmatrix} \varepsilon_{trn} \\ 0_d \end{bmatrix}$

   $= \varepsilon_{trn}^T V_{1:d} \Sigma \hat{\Sigma}^{-2} \underbrace{U^T u}\, v_{trn}^T (I - V_{1:d} \Sigma \hat{\Sigma}^{-2} \Sigma V_{1:d}^T) \varepsilon_{trn}.$

   We notice the vector $U^T u$ is uniformly random and centered by the rotational bi-invariance assumption. Hence, the expectation equals 0.

  (iii) $\hat{\varepsilon}_{trn}^T \hat{A}^\dagger \hat{A}^{\dagger T} \hat{k} \hat{t} \hat{\varepsilon}_{trn} = \hat{\varepsilon}_{trn}^T \hat{A}_{trn}^\dagger \hat{A}_{trn}^{\dagger T} \hat{A}_{trn}^\dagger u \hat{v}_{trn}^T (I - \hat{A}_{trn}^\dagger \hat{A}_{trn}) \hat{\varepsilon}_{trn}.$

   Similarly, with SVD this is just

   $\hat{\varepsilon}_{trn}^T \hat{V} \hat{\Sigma}^{-2} \hat{V}^T \hat{V} \hat{\Sigma}^{-1} U^T u \hat{v}_{trn}^T (I - \hat{V} \hat{V}^T) \hat{\varepsilon}_{trn}$

   $= \varepsilon_{trn}^T V_{1:d} \Sigma \hat{\Sigma}^{-3} \left( \hat{\Sigma}^{-1} \Sigma^2 \hat{\Sigma}^{-1} + \mu^2 \hat{\Sigma}^{-2} \right) \hat{\Sigma}^{-1} \underbrace{U^T u}\, v_{trn}^T (I - V_{1:d} \Sigma \hat{\Sigma}^{-2} \Sigma V_{1:d}^T) \varepsilon_{trn}.$

   Again we use the explicit form of $\hat{V}$ and note $\hat{V}^T \hat{V} = \hat{\Sigma}^{-1} \Sigma^2 \hat{\Sigma}^{-1} + \mu^2 \hat{\Sigma}^{-2}$. The vector $U^T u$ is uniformly random, so we have zero expectation.

  (iv) $\hat{\varepsilon}_{trn}^T \hat{A}^\dagger \hat{h}^T \hat{k}^T \hat{\varepsilon}_{trn} = \hat{\varepsilon}_{trn}^T \hat{A}_{trn}^\dagger \hat{A}_{trn}^{\dagger T} \hat{v}_{trn} u^T \hat{A}_{trn}^{\dagger T} \hat{\varepsilon}_{trn}.$

   We then have

   $\hat{\varepsilon}_{trn}^T \hat{V} \hat{\Sigma}^{-2} \hat{V}^T \hat{v}_{trn} u^T U \hat{\Sigma}^{-1} \hat{V}^T \hat{\varepsilon}_{trn} = \varepsilon_{trn}^T V_{1:d} \Sigma \hat{\Sigma}^{-4} \Sigma V_{1:d}^T v_{trn} \underbrace{u^T U}\, \hat{\Sigma}^{-1} \Sigma V_{1:d}^T \varepsilon_{trn}.$

   The vector $u^T U$ is uniformly random, so we have zero expectation.

Similarly, for the $c > 1$ case, we can prove it using the corresponding SVD, and the same results hold. $\qquad \square$

It is important to note that Lemma 16 does not directly apply due to the zeros in $\hat{\varepsilon}$. For the following Lemmas we only need that the variance is bounded. We do not need the variance to decay.

**Lemma 18.** *Under our assumptions, we have that*

$$
\mathbb{E}_{\varepsilon_{trn}, A_{trn}} \left[ \hat{\varepsilon}_{trn}^T \hat{k} \hat{k}^T \hat{\varepsilon}_{trn} \right] =
\begin{cases}
\tau_{\varepsilon_{trn}}^2 \left( \frac{\tau_{A_{trn}}^2 + c\tau_{A_{trn}}^2 + \mu^2 c}{2\tau_{A_{trn}}^2 \sqrt{\left(\tau_{A_{trn}}^2 + \mu^2 c - c\tau_{A_{trn}}^2\right)^2 + 4\mu^2 c^2 \tau_{A_{trn}}^2}} - \frac{1}{2\tau_{A_{trn}}^2} \right) + o(1/\tau_{A_{trn}}^2) & c < 1 \\
\tau_{\varepsilon_{trn}}^2 \left( \frac{\tau_{A_{trn}}^2 + c\tau_{A_{trn}}^2 + \mu^2 c}{2\tau_{A_{trn}}^2 \sqrt{\left(-\tau_{A_{trn}}^2 + \mu^2 c + c\tau_{A_{trn}}^2\right)^2 + 4\mu^2 c \tau_{A_{trn}}^2}} - \frac{1}{2\tau_{A_{trn}}^2} \right) + o(1/\tau_{A_{trn}}^2) & c > 1
\end{cases}.
$$

*Proof.* Suppose $c < 1$. We expand this term using SVD and have

$$
\begin{aligned}
\hat{\varepsilon}_{trn}^T \hat{k} \hat{k}^T \hat{\varepsilon}_{trn} &= \hat{\varepsilon}_{trn}^T \hat{A}_{trn}^{\dagger} uu^T \hat{A}_{trn}^{\dagger T} \hat{\varepsilon}_{trn} \\
&= \hat{\varepsilon}_{trn}^T \begin{bmatrix} V_{1:d} \Sigma \hat{\Sigma}^{-1} \\ \mu U \hat{\Sigma}^{-1} \end{bmatrix} \hat{\Sigma}^{-1} U^T uu^T U \hat{\Sigma}^{-1} \begin{bmatrix} \hat{\Sigma}^{-1} \Sigma V_{1:d}^T & \mu \hat{\Sigma}^{-1} U^T \end{bmatrix} \hat{\varepsilon}_{trn} \\
&= \varepsilon^T V_{1:d} \Sigma \hat{\Sigma}^{-2} U^T uu^T U \hat{\Sigma}^{-2} \Sigma V_{1:d}^T \varepsilon.
\end{aligned}
$$

We take its expectation and by Lemma 16, we have

$$
\begin{aligned}
\mathbb{E}_{\varepsilon_{trn}, A_{trn}} \left[ \hat{\varepsilon}_{trn}^T \hat{k} \hat{k}^T \hat{\varepsilon}_{trn} \right] &= \tau_{\varepsilon_{trn}}^2 \mathbb{E}_{A_{trn}} \left[ \text{Tr} \left( V_{1:d} \Sigma \hat{\Sigma}^{-2} U^T uu^T U \hat{\Sigma}^{-2} \Sigma V_{1:d}^T \right) \right] \\
&= \tau_{\varepsilon_{trn}}^2 \mathbb{E}_{A_{trn}} \left[ u^T U \hat{\Sigma}^{-2} \Sigma V_{1:d}^T V_{1:d} \Sigma \hat{\Sigma}^{-2} U^T u \right] \\
&= \tau_{\varepsilon_{trn}}^2 \mathbb{E}_{A_{trn}} \left[ \frac{\lambda}{(\lambda + \mu^2)^2} \right].
\end{aligned}
$$

The rest follows from Lemma 9. For $c > 1$, we use the same approach and get

$$
\mathbb{E}_{\varepsilon_{trn}, A_{trn}} \left[ \hat{\varepsilon}_{trn}^T \hat{k} \hat{k}^T \hat{\varepsilon}_{trn} \right] = \frac{\tau_{\varepsilon_{trn}}^2}{c} \mathbb{E}_{A_{trn}} \left[ \frac{\lambda}{(\lambda + \mu^2)^2} \right].
$$

where the additional factor of $1/c$ comes from projecting $n_{trn}$ entries onto the $d$ coordinates of the randomly uniform vector. $\qquad \square$

**Lemma 19.** *Under our assumptions, we have that*

$$
\mathbb{E}_{\varepsilon_{trn}, A_{trn}} \left[ \hat{\varepsilon}_{trn}^T \hat{t}^T \hat{t} \hat{\varepsilon}_{trn} \right] =
\begin{cases}
\tau_{\varepsilon_{trn}}^2 \left( \frac{\mu^2 c^2 + \mu^2 c + (c-1)^2 \tau_{A_{trn}}^2}{2\sqrt{(\tau_{A_{trn}}^2 + \mu^2 c - c\tau_{A_{trn}}^2)^2 + 4\mu^2 c^2 \tau_{A_{trn}}^2}} + \frac{1}{2}(1-c) \right) + o(1/\tau_{A_{trn}}^2) & c < 1 \\
\tau_{\varepsilon_{trn}}^2 \left( \frac{\mu^2 c^2 + \mu^2 c + (c-1)^2 \tau_{A_{trn}}^2}{2\sqrt{(-\tau_{A_{trn}}^2 + \mu^2 c + c\tau_{A_{trn}}^2)^2 + 4\mu^2 c \tau_{A_{trn}}^2}} + \frac{1}{2}(1-c) \right) + o(1/\tau_{A_{trn}}^2) & c > 1
\end{cases}.
$$

*Proof.* Suppose $c < 1$. We expand this term using SVD and have

$$
\begin{aligned}
\hat{\varepsilon}_{trn}^T \hat{t}^T \hat{t} \hat{\varepsilon}_{trn} &= \hat{\varepsilon}_{trn}^T (I - \hat{A}_{trn}^T \hat{A}_{trn}^{\dagger T}) \hat{v}_{trn} \hat{v}_{trn}^T (I - \hat{A}_{trn}^{\dagger} \hat{A}_{trn}) \hat{\varepsilon}_{trn} \\
&= \hat{\varepsilon}_{trn}^T (I - \hat{V} \hat{V}^T) \hat{v}_{trn} \hat{v}_{trn}^T (I - \hat{V} \hat{V}^T) \hat{\varepsilon}_{trn} \\
&= \varepsilon_{trn}^T (I - V_{1:d} \Sigma \hat{\Sigma}^{-2} \Sigma V_{1:d}^T) v_{trn} v_{trn}^T (I - V_{1:d} \Sigma \hat{\Sigma}^{-2} \Sigma V_{1:d}^T) \varepsilon_{trn}.
\end{aligned}
$$

We take its expectation and by Lemma 16, we have

$$
\begin{aligned}
\mathbb{E}_{\substack{\varepsilon_{trn}, \\ A_{trn}}} \left[ \hat{\varepsilon}_{trn}^T \hat{t}^T \hat{t} \hat{\varepsilon}_{trn} \right] &= \tau_{\varepsilon_{trn}}^2 \mathbb{E}_{A_{trn}} \left[ \text{Tr} \left( (I - V_{1:d} \Sigma \hat{\Sigma}^{-2} \Sigma V_{1:d}^T) v_{trn} v_{trn}^T (I - V_{1:d} \Sigma \hat{\Sigma}^{-2} \Sigma V_{1:d}^T) \right) \right] \\
&= \tau_{\varepsilon_{trn}}^2 \mathbb{E}_{A_{trn}} \left[ v_{trn}^T (I - V_{1:d} \Sigma \hat{\Sigma}^{-2} \Sigma V_{1:d}^T)^2 v_{trn} \right] \\
&= \tau_{\varepsilon_{trn}}^2 \mathbb{E}_{A_{trn}} \left[ v_{trn}^T v_{trn} - 2 v_{trn}^T V_{1:d} \Sigma \hat{\Sigma}^{-2} \Sigma V_{1:d}^T v_{trn} \right. \\
&\qquad\qquad \left. + v_{trn}^T V_{1:d} \Sigma \hat{\Sigma}^{-2} \Sigma^2 \hat{\Sigma}^{-2} \Sigma V_{1:d}^T v_{trn} \right] \\
&= \tau_{\varepsilon_{trn}}^2 \left( 1 + c \mathbb{E}_{A_{trn}} \left[ \frac{\lambda^2}{(\lambda + \mu^2)^2} \right] - 2c \mathbb{E}_{A_{trn}} \left[ \frac{\lambda}{\lambda + \mu^2} \right] \right).
\end{aligned}
$$

The factor of $c$ comes from projecting $d$ entries onto the $n_{trn}$ coordinates of the uniformly random vector.

The rest follows from Lemma 9. For $c > 1$, we use the same approach and get

$$\mathbb{E}_{\substack{\varepsilon_{trn}, \\ A_{trn}}} \left[ \hat{\varepsilon}_{trn}^T \hat{t}^T \hat{t} \hat{\varepsilon}_{trn} \right] = \tau_{\varepsilon_{trn}}^2 \left( 1 + \mathbb{E}_{A_{trn}} \left[ \frac{\lambda^2}{(\lambda + \mu^2)^2} \right] - 2\mathbb{E}_{A_{trn}} \left[ \frac{\lambda}{\lambda + \mu^2} \right] \right).$$

The variance directly follows from concentration. $\qquad\square$

**Lemma 20.** *Under our assumptions, we have that*

$$\mathbb{E}_{\substack{\varepsilon_{trn}, \\ A_{trn}}} \left[ \hat{\varepsilon}_{trn}^T \hat{A}_{trn}^\dagger \hat{A}_{trn}^{\dagger T} \hat{\varepsilon}_{trn} \right] = \begin{cases} \tau_{\varepsilon_{trn}}^2 d \left( \frac{\tau_{A_{trn}}^2 + c\tau_{A_{trn}}^2 + \mu^2 c}{2\tau_{A_{trn}}^2 \sqrt{(\tau_{A_{trn}}^2 + \mu^2 c - c\tau_{A_{trn}}^2)^2 + 4\mu^2 c^2 \tau_{A_{trn}}^2}} - \frac{1}{2\tau_{A_{trn}}^2} \right) + o(1/\tau_{A_{trn}}^2) & c < 1 \\[4mm] \tau_{\varepsilon_{trn}}^2 d \left( \frac{\tau_{A_{trn}}^2 + c\tau_{A_{trn}}^2 + \mu^2 c}{2\tau_{A_{trn}}^2 \sqrt{(-\tau_{A_{trn}}^2 + \mu^2 c + c\tau_{A_{trn}}^2)^2 + 4\mu^2 c\tau_{A_{trn}}^2}} - \frac{1}{2\tau_{A_{trn}}^2} \right) + o(1/\tau_{A_{trn}}^2) & c > 1 \end{cases}.$$

*Proof.* Suppose $c < 1$. We expand this term using SVD and have

$$\hat{\varepsilon}_{trn}^T \hat{A}_{trn}^\dagger \hat{A}_{trn}^{\dagger T} \hat{\varepsilon}_{trn} = \hat{\varepsilon}_{trn}^T \hat{V} \hat{\Sigma}^{-2} \hat{V}^T \hat{\varepsilon}_{trn} = \varepsilon_{trn}^T V_{1:d} \Sigma \hat{\Sigma}^{-4} \Sigma V_{1:d}^T \varepsilon_{trn}.$$

Again taking the expectation, we have

$$\mathbb{E}_{\varepsilon_{trn}, A_{trn}} \left[ \hat{\varepsilon}_{trn}^T \hat{A}_{trn}^\dagger \hat{A}_{trn}^{\dagger T} \hat{\varepsilon}_{trn} \right] = \tau_{\varepsilon_{trn}}^2 \mathbb{E}_{A_{trn}} \left[ \mathrm{Tr} \left( V_{1:d} \Sigma \hat{\Sigma}^{-4} \Sigma V_{1:d}^T \right) \right]$$

$$= \tau_{\varepsilon_{trn}}^2 \mathbb{E}_{A_{trn}} \left[ \mathrm{Tr} \left( \Sigma \hat{\Sigma}^{-4} \Sigma \right) \right]$$

$$= \tau_{\varepsilon_{trn}}^2 d \mathbb{E}_{A_{trn}} \left[ \frac{\lambda}{(\lambda + \mu^2)^2} \right].$$

where the factor of $d$ comes from summing up the $d$ diagonal elements.

The rest follows from Lemma 9. For $c > 1$, we use the same approach and get

$$\mathbb{E}_{\varepsilon_{trn}, A_{trn}} \left[ \hat{\varepsilon}_{trn}^T \hat{A}_{trn}^\dagger \hat{A}_{trn}^{\dagger T} \hat{\varepsilon}_{trn} \right] = \tau_{\varepsilon_{trn}}^2 N_{trn} \mathbb{E}_{A_{trn}} \left[ \frac{\lambda}{(\lambda + \mu^2)^2} \right] = \tau_{\varepsilon_{trn}}^2 \frac{M}{c} \mathbb{E}_{A_{trn}} \left[ \frac{\lambda}{(\lambda + \mu^2)^2} \right].$$

$\qquad\square$

**Lemma 21.** *Under our assumptions, we have that*

$$\mathbb{E}_{\varepsilon_{trn}, A_{trn}} \left[ \hat{\varepsilon}_{trn}^T \hat{A}_{trn}^\dagger \hat{h}^T \hat{t} \hat{\varepsilon}_{trn} \right] = \begin{cases} \frac{\tau_{\varepsilon_{trn}}^2 c^3 \mu^2 \tau_{A_{trn}}^2}{\left( (\tau_{A_{trn}}^2 + \mu^2 c - c\tau_{A_{trn}}^2)^2 + 4\mu^2 c^2 \tau_{A_{trn}}^2 \right)^{3/2}} + o(1/\tau_{A_{trn}}^2) & c < 1 \\[4mm] \frac{\tau_{\varepsilon_{trn}}^2 c^3 \mu^2 \tau_{A_{trn}}^2}{\left( (-\tau_{A_{trn}}^2 + \mu^2 c + c\tau_{A_{trn}}^2)^2 + 4\mu^2 c\tau_{A_{trn}}^2 \right)^{3/2}} + o(1/\tau_{A_{trn}}^2) & c > 1 \end{cases}.$$

*Proof.* Suppose $c < 1$. We expand this term using SVD and have

$$\hat{\varepsilon}_{trn}^T \hat{A}_{trn}^\dagger \hat{h}^T \hat{t} \hat{\varepsilon}_{trn} = \hat{\varepsilon}_{trn}^T \hat{A}_{trn}^\dagger \hat{A}_{trn}^{\dagger T} \hat{v}_{trn} \hat{v}_{trn}^T (I - \hat{A}_{trn}^\dagger \hat{A}_{trn}) \hat{\varepsilon}_{trn}$$

$$= \hat{\varepsilon}_{trn}^T \hat{V} \hat{\Sigma}^{-2} \hat{V}^T \hat{v}_{trn} \hat{v}_{trn}^T (I - \hat{V} \hat{V}^T) \hat{\varepsilon}_{trn}$$

$$= \varepsilon_{trn}^T V_{1:d} \Sigma \hat{\Sigma}^{-4} \Sigma V_{1:d}^T v_{trn} v_{trn}^T (I - V_{1:d} \Sigma \hat{\Sigma}^{-2} \Sigma V_{1:d}^T) \varepsilon_{trn}.$$

We take its expectation and by Lemma 16, we have

$$\mathbb{E}_{\substack{\varepsilon_{trn}, \\ A_{trn}}} \left[ \hat{\varepsilon}_{trn}^T \hat{A}_{trn}^\dagger \hat{h}^T \hat{t} \hat{\varepsilon}_{trn} \right] = \tau_{\varepsilon_{trn}}^2 \mathbb{E}_{A_{trn}} \left[ \mathrm{Tr} \left( v_{trn}^T (I - V_{1:d} \Sigma \hat{\Sigma}^{-2} \Sigma V_{1:d}^T) V_{1:d} \Sigma \hat{\Sigma}^{-4} \Sigma V_{1:d}^T v_{trn} \right) \right]$$

$$= \tau_{\varepsilon_{trn}}^2 \mathbb{E}_{A_{trn}} \left[ v_{trn}^T V_{1:d} \Sigma \hat{\Sigma}^{-4} \Sigma V_{1:d}^T v_{trn} - v_{trn}^T V_{1:d} \Sigma \hat{\Sigma}^{-2} \Sigma^2 \hat{\Sigma}^{-4} \Sigma V_{1:d}^T v_{trn} \right]$$

$$= \tau_{\varepsilon_{trn}}^2 \left( c\mathbb{E}_{A_{trn}} \left[ \frac{\lambda}{(\lambda + \mu^2)^2} \right] - c\mathbb{E}_{A_{trn}} \left[ \frac{\lambda^2}{(\lambda + \mu^2)^3} \right] \right)$$

$$= \tau_{\varepsilon_{trn}}^2 \left( c\mu^2 \mathbb{E}_{A_{trn}} \left[ \frac{1}{(\lambda + \mu^2)^2} \right] - c\mu^4 \mathbb{E}_{A_{trn}} \left[ \frac{1}{(\lambda + \mu^2)^3} \right] \right).$$

The factor of $c$ comes from projecting $d$ entries onto the $n_{trn}$ coordinates of the uniformly random vector. The rest follows from Lemma 7. For $c > 1$, we use the same approach and get

$$\mathbb{E}_{\substack{\varepsilon_{trn}, \\ A_{trn}}} \left[ \hat{\varepsilon}_{trn}^T \hat{A}_{trn}^\dagger \hat{h}^T \hat{t} \hat{\varepsilon}_{trn} \right] = \tau_{\varepsilon_{trn}}^2 \left( \mu^2 \mathbb{E}_{A_{trn}} \left[ \frac{1}{(\lambda + \mu^2)^2} \right] - \mu^4 \mathbb{E}_{A_{trn}} \left[ \frac{1}{(\lambda + \mu^2)^3} \right] \right).$$

$\square$

**Lemma 22.** *Under our assumptions, we have that*

$$\mathbb{E}_{\varepsilon_{trn}, A_{trn}} \left[ \hat{\varepsilon}_{trn}^T \hat{A}_{trn}^\dagger \hat{A}_{trn}^{\dagger T} \hat{k} \hat{k}^T \hat{\varepsilon}_{trn} \right] = \begin{cases} \frac{\tau_{\varepsilon_{trn}}^2 c^2 \tau_{A_{trn}}^2}{\left( (\tau_{A_{trn}}^2 + \mu^2 c - c\tau_{A_{trn}}^2)^2 + 4\mu^2 c^2 \tau_{A_{trn}}^2 \right)^{3/2}} + o(1) & c < 1 \\ \frac{\tau_{\varepsilon_{trn}}^2 c^2 \tau_{A_{trn}}^2}{\left( (-\tau_{A_{trn}}^2 + \mu^2 c + c\tau_{A_{trn}}^2)^2 + 4\mu^2 c^2 \tau_{A_{trn}}^2 \right)^{3/2}} + o(1) & c > 1 \end{cases}$$

*and that* $Var(\hat{\varepsilon}_{trn}^T \hat{A}_{trn}^\dagger \hat{A}_{trn}^{\dagger T} \hat{k} \hat{k}^T \hat{\varepsilon}_{trn}) = o(1).$

*Proof.* Suppose $c < 1$. We expand this term using SVD and have

$$\hat{\varepsilon}_{trn}^T \hat{A}_{trn}^\dagger \hat{A}_{trn}^{\dagger T} \hat{k} \hat{k}^T \hat{\varepsilon}_{trn} = \hat{\varepsilon}_{trn}^T \hat{A}_{trn}^\dagger \hat{A}_{trn}^{\dagger T} \hat{A}_{trn}^\dagger u u^T \hat{A}_{trn}^{\dagger T} \hat{\varepsilon}_{trn}$$

$$= \hat{\varepsilon}_{trn}^T \hat{V} \hat{\Sigma}^{-2} \hat{V}^T \hat{V} \hat{\Sigma}^{-1} U^T u u^T U \hat{\Sigma}^{-1} \hat{V}^T \hat{\varepsilon}_{trn}$$

$$= \varepsilon_{trn}^T V_{1:d} \Sigma \hat{\Sigma}^{-3} \left( \hat{\Sigma}^{-1} \Sigma^2 \hat{\Sigma}^{-1} + \mu^2 \hat{\Sigma}^{-2} \right) \hat{\Sigma}^{-1} U^T u u^T U \hat{\Sigma}^{-2} \Sigma V_{1:d}^T \varepsilon_{trn}.$$

We take its expectation and by Lemma 16, we have

$$\mathbb{E}_{\substack{\varepsilon_{trn}, \\ A_{trn}}} \left[ \hat{\varepsilon}_{trn}^T \hat{A}_{trn}^\dagger \hat{A}_{trn}^{\dagger T} \hat{k} \hat{k}^T \hat{\varepsilon}_{trn} \right] = \tau_{\varepsilon_{trn}}^2 \mathbb{E}_{A_{trn}} \left[ u^T U \hat{\Sigma}^{-2} \Sigma V_{1:d}^T V_{1:d} \Sigma \hat{\Sigma}^{-3} \left( \hat{\Sigma}^{-1} \Sigma^2 \hat{\Sigma}^{-1} + \mu^2 \hat{\Sigma}^{-2} \right) \hat{\Sigma}^{-1} U^T u \right]$$

$$= \tau_{\varepsilon_{trn}}^2 \mathbb{E}_{A_{trn}} \left[ u^T U \hat{\Sigma}^{-2} \Sigma^2 \hat{\Sigma}^{-4} \Sigma^2 \hat{\Sigma}^{-2} U^T u - \mu^2 u^T U \hat{\Sigma}^{-2} \Sigma^2 \hat{\Sigma}^{-6} U^T u \right]$$

$$= \tau_{\varepsilon_{trn}}^2 \left( \mathbb{E}_{A_{trn}} \left[ \frac{\lambda^2}{(\lambda + \mu^2)^4} \right] - \mathbb{E}_{A_{trn}} \left[ \mu^2 \frac{\lambda}{(\lambda + \mu^2)^4} \right] \right)$$

$$= \tau_{\varepsilon_{trn}}^2 \left( \mathbb{E}_{A_{trn}} \left[ \frac{\lambda}{(\lambda + \mu^2)^3} \right] \right)$$

$$= \tau_{\varepsilon_{trn}}^2 \left( \mathbb{E}_{A_{trn}} \left[ \frac{1}{(\lambda + \mu^2)^2} \right] - \mu^2 \mathbb{E}_{A_{trn}} \left[ \frac{1}{(\lambda + \mu^2)^3} \right] \right).$$

The rest follows from Lemma 7. For $c > 1$, we use the same approach and get

$$\mathbb{E}_{\substack{\varepsilon_{trn}, \\ A_{trn}}} \left[ \hat{\varepsilon}_{trn}^T \hat{A}_{trn}^\dagger \hat{h}^T \hat{t} \hat{\varepsilon}_{trn} \right] = \tau_{\varepsilon_{trn}}^2 \left( \frac{1}{c} \mathbb{E}_{A_{trn}} \left[ \frac{1}{(\lambda + \mu^2)^2} \right] - \frac{\mu^2}{c} \mathbb{E}_{A_{trn}} \left[ \frac{1}{(\lambda + \mu^2)^3} \right] \right).$$

$\square$

These are all the expectations we need for the final derivation of the error formula. We present the full results here, but readers might have noticed a substantial similarity in each pair of cases: the two formulas only differ in the radical. This observation will allow us to present the final formula in a more concise way.

## A.5 STEP 5: PUT THINGS TOGETHER

**Proposition 1.** *Under our assumptions, we have that for the bias term, if $c < 1$,*

$$\mathbb{E}_{\varepsilon_{trn}, A_{trn}} \left\| \beta_*^T Z_{tst} - \beta_{so}^T Z_{tst} \right\|_F^2 = \frac{\theta_{tst}^2}{\gamma^2} \left[ (\beta_*^T u)^2 + \frac{\tau_{\varepsilon_{trn}}^2}{2\tau_{A_{trn}}^4} \left( \theta_{trn}^2 c + \tau_{A_{trn}}^2 \right) (T_2 - 1) \right] + o \left( \frac{\theta_{tst}^2}{\theta_{trn}^2} \right)$$

*where*

$$T_1 = \sqrt{\left( \tau_{A_{trn}}^2 + \mu^2 c - c\tau_{A_{trn}}^2 \right)^2 + 4\mu^2 c^2 \tau_{A_{trn}}^2}, \quad T_2 = \frac{\mu^2 c + \tau_{A_{trn}}^2 + c\tau_{A_{trn}}^2}{T_1},$$

$$\text{and } \gamma = 1 + \frac{\theta_{trn}^2}{2\tau_{A_{trn}}^4} \left( \tau_{A_{trn}}^2 + c\tau_{A_{trn}}^2 + \mu^2 c - T_1 \right).$$

*For $c > 1$, the same formula holds except*

$$T_1 = \sqrt{\left( -\tau_{A_{trn}}^2 + \mu^2 c + c\tau_{A_{trn}}^2 \right)^2 + 4\mu^2 c\tau_{A_{trn}}^2}.$$

*Proof.* By Lemma 3, we can rewrite the bias term as

$$\left\| \beta_*^T Z_{tst} - \beta_{so}^T Z_{tst} \right\|_F^2 = \left\| \frac{\hat{\xi}}{\gamma} \beta_*^T Z_{tst} + \frac{\theta_{tst}\hat{\xi}}{\theta_{trn}\gamma} \hat{\varepsilon}_{trn}^T \hat{p} v_{tst}^T \right\|_F^2$$

$$= \left\| \frac{\hat{\xi}}{\gamma} \beta_*^T Z_{tst} \right\|_F^2 + \left\| \frac{\theta_{tst}\hat{\xi}}{\theta_{trn}\gamma} \hat{\varepsilon}_{trn}^T \hat{p} v_{tst}^T \right\|_F^2 + 2\operatorname{Tr}\left( \frac{\theta_{tst}\hat{\xi}^2}{\theta_{trn}\gamma^2} Z_{tst}^T W \hat{\varepsilon}_{trn}^T \hat{p} v_{tst}^T \right).$$

Note the last term is zero in expectation since $\hat{\varepsilon}_{trn}$ has mean 0 entries. We go ahead and expand the other two terms.

Using $Z_{tst} = \theta_{tst} u v_{tst}^T$, we first have

$$\left\| \frac{\hat{\xi}}{\gamma} \beta_*^T Z_{tst} \right\|_F^2 = \frac{\theta_{tst}^2 \hat{\xi}^2}{\gamma^2} \beta_*^T u v_{tst}^T v_{tst} u^T \beta_* = \frac{\theta_{tst}^2 \hat{\xi}^2}{\gamma^2} (\beta_*^T u)^2$$

since $v_{tst}^T v_{tst} = \|v_{tst}\|^2 = 1$. We also have

$$\left\| \frac{\theta_{tst}\hat{\xi}}{\theta_{trn}\gamma} \hat{\varepsilon}_{trn}^T \hat{p} v_{tst}^T \right\|_F^2 = \frac{\theta_{tst}^2 \hat{\xi}^2}{\theta_{trn}^2 \gamma^2} \operatorname{Tr}\left( \hat{\varepsilon}_{trn}^T \hat{p} v_{tst}^T v_{tst} \hat{p}^T \hat{\varepsilon}_{trn} \right) = \frac{\theta_{tst}^2 \hat{\xi}^2}{\theta_{trn}^2 \gamma^2} \hat{\varepsilon}_{trn}^T \hat{p} \hat{p}^T \hat{\varepsilon}_{trn}.$$

We plug in $\hat{p} = -\frac{\theta_{trn}^2 \|\hat{k}\|^2}{\hat{\xi}} \hat{t}^T - \theta_{trn}\hat{k}$ and expand. We get

$$\frac{\theta_{tst}^2 \hat{\xi}^2}{\theta_{trn}^2 \gamma^2} \left( \frac{\theta_{trn}^4 \|\hat{k}\|^4}{\hat{\xi}^2} \hat{\varepsilon}_{trn}^T \hat{t}^T \hat{t} \hat{\varepsilon}_{trn} + \frac{2\theta_{trn}^3 \|\hat{k}\|^2}{\hat{\xi}} \underbrace{\hat{\varepsilon}_{trn}^T \hat{k} \hat{t} \hat{\varepsilon}_{trn}}_{0} + \theta_{trn}^2 \hat{\varepsilon}_{trn}^T \hat{k} \hat{k}^T \hat{\varepsilon}_{trn} \right).$$

Since the term has mean zero and $\theta_{trn}$ is a constant it doesn't effect the mean. Hence can divide and multiply by $\theta_{trn}^2$ so that we get a factor of $\theta_{trn}^2/\gamma^2$ which then has the appropriate variance. Hence, the second term equals $0 + o(1)$ in expectation. We then have that w.r.t. $A_{trn}$ and $\varepsilon_{trn}$,

$$\mathbb{E}\left\| \beta_*^T Z_{tst} - \beta_{so}^T Z_{tst} \right\|_F^2 = \theta_{tst}^2 (\beta_*^T u)^2 \mathbb{E}\left[ \frac{\hat{\xi}^2}{\gamma^2} \right] + \theta_{tst}^2 \theta_{trn}^2 \mathbb{E}\left[ \frac{\|\hat{k}\|^4}{\gamma^2} \hat{\varepsilon}_{trn}^T \hat{t}^T \hat{t} \hat{\varepsilon}_{trn} \right] + \theta_{tst}^2 \mathbb{E}\left[ \frac{1}{\gamma^2} \hat{\varepsilon}_{trn}^T \hat{k} \hat{k}^T \hat{\varepsilon}_{trn} \right].$$

By concentration and Lemmas 11, 13, 14, 18, 19, we use SymPy to directly multiply individual expectations and get the results. $\qquad\square$

**Proposition 2.** *Under our assumptions, with $\tilde{\beta}^T = \hat{Z}_{trn}(\hat{Z}_{trn} + \hat{A}_{trn})^{\dagger}$, we have that if $c < 1$,*

$$\mathbb{E}_{A_{trn}}\left[ (\beta_*^T u)^2 \|\tilde{\beta}\|^2 \right] = \frac{\theta_{trn}^2}{\gamma^2} (\beta_*^T u)^2 \left[ \frac{c\left( \theta_{trn}^2 + \tau_{A_{trn}}^2 \right)}{2\tau_{A_{trn}}^4} (T_2 - 1) \right] + o(1),$$

*where*

$$T_1 = \sqrt{\left( \tau_{A_{trn}}^2 + \mu^2 c - c\tau_{A_{trn}}^2 \right)^2 + 4\mu^2 c^2 \tau_{A_{trn}}^2}, \quad T_2 = \frac{\mu^2 c + \tau_{A_{trn}}^2 + c\tau_{A_{trn}}^2}{T_1},$$

$$\text{and } \gamma = 1 + \frac{\theta_{trn}^2}{2\tau_{A_{trn}}^4} \left( \tau_{A_{trn}}^2 + c\tau_{A_{trn}}^2 + \mu^2 c - T_1 \right).$$

*For $c > 1$, the same formula holds except*

$$T_1 = \sqrt{\left( -\tau_{A_{trn}}^2 + \mu^2 c + c\tau_{A_{trn}}^2 \right)^2 + 4\mu^2 c\tau_{A_{trn}}^2}.$$

*Proof.* Li & Sonthalia (2024) studies the ridge-regularized denoising setting. By its Lemma 4, with our notations,

$$\|\tilde{W}_{opt}\|^2 = \frac{\theta_{trn}^2 \hat{\xi}^2}{\gamma^2} \|\hat{h}\|^2 + 2\frac{\theta_{trn}^3 \|\hat{t}\|^2 \hat{\xi}}{\gamma^2} \hat{k}^T \hat{A}_{trn}^{\dagger} \hat{h}^T + \frac{\theta_{trn}^4 \|\hat{t}\|^4}{\gamma^2} \hat{k}^T \hat{A}_{trn}^{\dagger} \hat{A}_{trn}^{\dagger T} \hat{k}.$$

The rest follows from concentration and our Lemmas 10, 12, 13, 14, 15, 17.

**Proposition 3.** *Under our assumptions, we have that if $c < 1$,*

$$\mathbb{E}_{\varepsilon_{trn}, A_{trn}} \left[ \hat{\varepsilon}_{trn}^T (\hat{Z}_{trn} + \hat{A}_{trn})^\dagger (\hat{Z}_{trn} + \hat{A}_{trn})^{\dagger T} \hat{\varepsilon}_{trn} \right]$$

$$= \frac{\tau_{\varepsilon_{trn}}^2}{2\tau_{A_{trn}}^2} \left[ d + \frac{c\theta_{trn}^2}{\tau_{A_{trn}}^2} \frac{T_2}{\gamma^2} \left( \frac{(c+1)\theta_{trn}^2}{\tau_{A_{trn}}^2} + 1 \right) \right] (T_2 - 1)$$

$$- \frac{c^2(c+1)\theta_{trn}^4 \tau_{\varepsilon_{trn}}^2}{\tau_{A_{trn}}^2} \frac{1}{\gamma^2 T_1^2} - \frac{2\theta_{trn}^2 c^2 \tau_{\varepsilon_{trn}}^2}{\gamma} \left( \frac{1}{T_1^2} - \frac{c\mu^2}{T_1^3} \right),$$

*where*

$$T_1 = \sqrt{\left( \tau_{A_{trn}}^2 + \mu^2 c - c\tau_{A_{trn}}^2 \right)^2 + 4\mu^2 c^2 \tau_{A_{trn}}^2}, \; T_2 = \frac{\mu^2 c + \tau_{A_{trn}}^2 + c\tau_{A_{trn}}^2}{T_1},$$

$$\text{and } \gamma = 1 + \frac{\theta_{trn}^2}{2\tau_{A_{trn}}^4} \left( \tau_{A_{trn}}^2 + c\tau_{A_{trn}}^2 + \mu^2 c - T_1 \right).$$

*For $c > 1$, the same formula holds except*

$$T_1 = \sqrt{\left( -\tau_{A_{trn}}^2 + \mu^2 c + c\tau_{A_{trn}}^2 \right)^2 + 4\mu^2 c\tau_{A_{trn}}^2}.$$

**Remark:** This term corresponds to the part of variance further induced by $\varepsilon_{trn}$.

*Proof.* We first expand this term and cross out individual terms with zero expectation, denoting them accordingly.

$$\hat{\varepsilon}_{trn}^T (\hat{Z}_{trn} + \hat{A}_{trn})^\dagger (\hat{Z}_{trn} + \hat{A}_{trn})^{\dagger T} \hat{\varepsilon}_{trn}$$

$$= \hat{\varepsilon}_{trn}^T \left( \hat{A}_{trn}^\dagger + \frac{\theta_{trn}}{\hat{\xi}} \hat{t}^T \hat{k}^T \hat{A}_{trn}^\dagger - \frac{\hat{\xi}}{\gamma} \hat{p}\hat{q}^T \right) \left( \hat{A}_{trn}^\dagger + \frac{\theta_{trn}}{\hat{\xi}} \hat{t}^T \hat{k}^T \hat{A}_{trn}^\dagger - \frac{\hat{\xi}}{\gamma} \hat{p}\hat{q}^T \right)^T \hat{\varepsilon}_{trn}$$

$$= \hat{\varepsilon}_{trn}^T \hat{A}_{trn}^\dagger \hat{A}_{trn}^{\dagger T} \hat{\varepsilon}_{trn} + \frac{2\theta_{trn}}{\hat{\xi}} \underbrace{\hat{\varepsilon}_{trn}^T \hat{A}_{trn}^\dagger \hat{A}_{trn}^{\dagger T} \hat{k}\hat{t}\hat{\varepsilon}_{trn}}_{0} - \frac{2\hat{\xi}}{\gamma} \hat{\varepsilon}_{trn}^T \hat{A}_{trn}^\dagger \hat{q}\hat{p}^T \hat{\varepsilon}_{trn}$$

$$+ \frac{\theta_{trn}^2}{\hat{\xi}^2} \left( \hat{k}^T \hat{A}_{trn}^\dagger \hat{A}_{trn}^{\dagger T} \hat{k} \right) \hat{\varepsilon}_{trn}^T \hat{t}^T \hat{t}\hat{\varepsilon}_{trn} - \frac{2\theta_{trn}}{\gamma} \hat{\varepsilon}_{trn}^T \hat{t}^T \hat{k}^T \hat{A}_{trn}^\dagger \hat{q}\hat{p}^T \hat{\varepsilon}_{trn} + \frac{\hat{\xi}^2}{\gamma^2} \hat{\varepsilon}_{trn}^T \hat{p}\hat{q}^T \hat{q}\hat{p}^T \hat{\varepsilon}_{trn}$$

We know the expectation of the first term from Lemma. Here the second term equals 0 in expectation by Lemma 17. We expand the other terms one by one, marking those with zero expectations:

$$- \frac{2\hat{\xi}}{\gamma} \hat{\varepsilon}_{trn}^T \hat{A}_{trn}^\dagger \hat{q}\hat{p}^T \hat{\varepsilon}_{trn} = - \frac{2\hat{\xi}}{\gamma} \hat{\varepsilon}_{trn}^T \hat{A}_{trn}^\dagger \left( -\frac{\theta_{trn}\|\hat{t}\|^2}{\hat{\xi}} \hat{A}_{trn}^{\dagger T} \hat{k} - \hat{h}^T \right) \left( -\frac{\theta_{trn}^2 \|\hat{k}\|^2}{\hat{\xi}} \hat{t} - \theta_{trn}\hat{k}^T \right) \hat{\varepsilon}_{trn}$$

$$= - \frac{2\theta_{trn}^3 \|\hat{t}\|^2 \|\hat{k}\|^2}{\gamma\hat{\xi}} \underbrace{\hat{\varepsilon}_{trn}^T \hat{A}_{trn}^\dagger \hat{A}_{trn}^{\dagger T} \hat{k}\hat{t}\hat{\varepsilon}_{trn}}_{0} - \frac{2\theta_{trn}^2 \|\hat{t}\|^2}{\gamma} \hat{\varepsilon}_{trn}^T \hat{A}_{trn}^\dagger \hat{A}_{trn}^{\dagger T} \hat{k}\hat{k}^T \hat{\varepsilon}_{trn}$$

$$- \frac{2\theta_{trn}^2 \|\hat{k}\|^2}{\gamma} \hat{\varepsilon}_{trn}^T \hat{A}_{trn}^\dagger \hat{h}^T \hat{t}\hat{\varepsilon}_{trn} - \frac{2\theta_{trn}}{\gamma} \underbrace{\hat{\varepsilon}_{trn}^T \hat{A}_{trn}^\dagger \hat{h}^T \hat{k}^T \hat{\varepsilon}_{trn}}_{0}.$$

$$- \frac{2\theta_{trn}}{\gamma} \hat{\varepsilon}_{trn}^T \hat{t}^T \hat{k}^T \hat{A}_{trn}^\dagger \hat{q}\hat{p}^T \hat{\varepsilon}_{trn} = - \frac{2\theta_{trn}}{\gamma} \hat{\varepsilon}_{trn}^T \hat{t}^T \hat{k}^T \hat{A}_{trn}^\dagger \left( -\frac{\theta_{trn}\|\hat{t}\|^2}{\hat{\xi}} \hat{A}_{trn}^{\dagger T} \hat{k} - \hat{h}^T \right) \left( -\frac{\theta_{trn}^2 \|\hat{k}\|^2}{\hat{\xi}} \hat{t} - \theta_{trn}\hat{k}^T \right) \hat{\varepsilon}_{trn}$$

$$= - \frac{2\theta_{trn}^4 \|\hat{t}\|^2 \|\hat{k}\|^2}{\gamma\hat{\xi}^2} \left( \hat{k}^T \hat{A}_{trn}^\dagger \hat{A}_{trn}^{\dagger T} \hat{k} \right) \hat{\varepsilon}_{trn}^T \hat{t}^T \hat{t}\hat{\varepsilon}_{trn}$$

$$- \frac{2\theta_{trn}^3 \|\hat{t}\|^2}{\gamma\hat{\xi}} \left( \hat{k}^T \hat{A}_{trn}^\dagger \hat{A}_{trn}^{\dagger T} \hat{k} \right) \underbrace{\hat{\varepsilon}_{trn}^T \hat{t}^T \hat{k}^T \hat{\varepsilon}_{trn}}_{0}$$

$$- \frac{2\theta_{trn}^3 \|\hat{k}\|^2}{\gamma\hat{\xi}} \hat{\varepsilon}_{trn}^T \hat{t}^T \underbrace{\hat{k}^T \hat{A}_{trn}^\dagger \hat{h}^T}_{0} \hat{t}\hat{\varepsilon}_{trn} - \frac{2\theta_{trn}^2}{\gamma} \hat{\varepsilon}_{trn}^T \hat{t}^T \underbrace{\hat{k}^T \hat{A}_{trn}^\dagger \hat{h}^T}_{0} \hat{k}^T \hat{\varepsilon}_{trn}.$$

We further denote

$$Q = \hat{p}\hat{q}^T = \frac{\theta_{trn}^3 \|\hat{k}\|^2 \|\hat{t}\|^2}{\hat{\xi}^2} \hat{t}^T \hat{k}^T \hat{A}_{trn}^\dagger + \frac{\theta_{trn}^2 \|\hat{k}\|^2}{\hat{\xi}} \hat{t}^T \hat{h} + \frac{\theta_{trn}^2 \|\hat{t}\|^2}{\hat{\xi}} \hat{k}\hat{k}^T \hat{A}_{trn}^\dagger + \theta_{trn}\hat{k}\hat{h} \quad \text{(A.7)}$$

Then using this result, we expand the last term

$$\frac{\hat{\xi}^2}{\gamma^2} \hat{\varepsilon}_{trn}^T \hat{p}\hat{q}^T \hat{q}\hat{p}^T \hat{\varepsilon}_{trn} = \frac{\hat{\xi}^2}{\gamma^2} \hat{\varepsilon}_{trn}^T Q Q^T \hat{\varepsilon}_{trn}$$

$$= \frac{\theta_{trn}^6 \|\hat{t}\|^4 \|\hat{k}\|^4}{\gamma^2 \hat{\xi}^2} \left( \hat{k}^T \hat{A}_{trn}^\dagger \hat{A}_{trn}^{\dagger T} \hat{k} \right) \hat{\varepsilon}_{trn}^T \hat{t}^T \hat{t}\hat{\varepsilon}_{trn} + \frac{2\theta_{trn}^5 \|\hat{k}\|^4 \|\hat{t}\|^2}{\gamma^2 \hat{\xi}} \hat{\varepsilon}_{trn}^T \hat{t}^T \underbrace{\hat{k}^T \hat{A}_{trn}^\dagger \hat{h}^T}_{0} \hat{t}\hat{\varepsilon}_{trn}$$

$$+ \frac{2\theta_{trn}^5 \|\hat{k}\|^2 \|\hat{t}\|^4}{\gamma^2 \hat{\xi}} \left( \hat{k}^T \hat{A}_{trn}^\dagger \hat{A}_{trn}^{\dagger T} \hat{k} \right) \underbrace{\hat{\varepsilon}_{trn}^T \hat{k}\hat{t}\hat{\varepsilon}_{trn}}_{0} + \frac{2\theta_{trn}^4 \|\hat{k}\|^2 \|\hat{t}\|^2}{\gamma^2} \hat{\varepsilon}_{trn}^T \hat{k} \underbrace{\hat{k}^T \hat{A}_{trn}^\dagger \hat{h}^T}_{0} \hat{t}\hat{\varepsilon}_{trn}$$

$$+ \frac{\theta_{trn}^4 \|\hat{k}\|^4 \|\hat{h}\|^2}{\gamma^2} \hat{\varepsilon}_{trn}^T \hat{t}^T \hat{t}\hat{\varepsilon}_{trn} + \frac{4\theta_{trn}^4 \|\hat{k}\|^2 \|\hat{t}\|^2}{\gamma^2} \hat{\varepsilon}_{trn}^T \hat{t}^T \underbrace{\hat{k}^T \hat{A}_{trn}^\dagger \hat{h}^T}_{0} \hat{t}\hat{\varepsilon}_{trn}$$

$$+ \frac{2\theta_{trn}^3 \|\hat{k}\|^2 \|\hat{h}\|^2 \hat{\xi}}{\gamma^2} \underbrace{\hat{\varepsilon}_{trn}^T \hat{k}\hat{t}\hat{\varepsilon}_{trn}}_{0} + \frac{\theta_{trn}^4 \|\hat{t}\|^4}{\gamma^2} \left( \hat{k}^T \hat{A}_{trn}^\dagger \hat{A}_{trn}^{\dagger T} \hat{k} \right) \hat{\varepsilon}_{trn}^T \hat{k}\hat{k}^T \hat{\varepsilon}_{trn}$$

$$+ \frac{2\theta_{trn}^3 \|\hat{t}\|^2 \hat{\xi}}{\gamma^2} \hat{\varepsilon}_{trn}^T \hat{k} \underbrace{\hat{k}^T \hat{A}_{trn}^\dagger \hat{h}^T}_{0} \hat{k}^T \hat{\varepsilon}_{trn} + \frac{\theta_{trn}^2 \|\hat{h}\|^2 \hat{\xi}^2}{\gamma^2} \hat{\varepsilon}_{trn}^T \hat{k}\hat{k}^T \hat{\varepsilon}_{trn}.$$

All the cross terms will have zero expectation here.

Now the only terms with nonzero expectation are

(i) $\hat{\varepsilon}_{trn}^T \hat{A}_{trn}^\dagger \hat{A}_{trn}^{\dagger T} \hat{\varepsilon}_{trn}$.

(ii) $-\dfrac{2\theta_{trn}^2}{\gamma} \left( \|\hat{t}\|^2 \hat{\varepsilon}_{trn}^T \hat{A}_{trn}^\dagger \hat{A}_{trn}^{\dagger T} \hat{k}\hat{k}^T \hat{\varepsilon}_{trn} + \|\hat{k}\|^2 \hat{\varepsilon}_{trn}^T \hat{A}_{trn}^\dagger \hat{h}^T \hat{t}\hat{\varepsilon}_{trn} \right)$.

(iii) $\dfrac{\theta_{trn}^4}{\gamma^2} \left( \|\hat{t}\|^4 \left( \hat{k}^T \hat{A}_{trn}^\dagger \hat{A}_{trn}^{\dagger T} \hat{k} \right) \hat{\varepsilon}_{trn}^T \hat{k}\hat{k}^T \hat{\varepsilon}_{trn} + \|\hat{k}\|^4 \|\hat{h}\|^2 \hat{\varepsilon}_{trn}^T \hat{t}^T \hat{t}\hat{\varepsilon}_{trn} \right)$.

(iv)

$$\left( \frac{\theta_{trn}^6 \|\hat{t}\|^4 \|\hat{k}\|^4}{\gamma^2 \hat{\xi}^2} - \frac{2\theta_{trn}^4 \|\hat{t}\|^2 \|\hat{k}\|^2}{\gamma \hat{\xi}^2} + \frac{\theta_{trn}^2}{\hat{\xi}^2} \right) \left( \hat{k}^T \hat{A}_{trn}^\dagger \hat{A}_{trn}^{\dagger T} \hat{k} \right) \hat{\varepsilon}_{trn}^T \hat{t}^T \hat{t}\hat{\varepsilon}_{trn} + \frac{\theta_{trn}^2 \|\hat{h}\|^2 \hat{\xi}^2}{\gamma^2} \hat{\varepsilon}_{trn}^T \hat{k}\hat{k}^T \hat{\varepsilon}_{trn}$$

$$= \frac{\left( \theta_{trn}^3 \|\hat{t}\|^2 \|\hat{k}\|^2 - \theta_{trn}\gamma \right)^2}{\gamma^2 \hat{\xi}^2} \left( \hat{k}^T \hat{A}_{trn}^\dagger \hat{A}_{trn}^{\dagger T} \hat{k} \right) \hat{\varepsilon}_{trn}^T \hat{t}^T \hat{t}\hat{\varepsilon}_{trn} + \frac{\theta_{trn}^2 \|\hat{h}\|^2 \hat{\xi}^2}{\gamma^2} \hat{\varepsilon}_{trn}^T \hat{k}\hat{k}^T \hat{\varepsilon}_{trn}$$

$$= \frac{\theta_{trn}^2}{\gamma^2} \left( \left( \hat{k}^T \hat{A}_{trn}^\dagger \hat{A}_{trn}^{\dagger T} \hat{k} \right) \hat{\varepsilon}_{trn}^T \hat{t}^T \hat{t}\hat{\varepsilon}_{trn} + \|\hat{h}\|^2 \hat{\xi}^2 \hat{\varepsilon}_{trn}^T \hat{k}\hat{k}^T \hat{\varepsilon}_{trn} \right).$$

In the last step, the cancellation follows from $\gamma = \theta_{trn}^2 \|\hat{t}\|^2 \|\hat{k}\|^2 + \hat{\xi}^2$. We use Lemmas 10, 11, 12, 13, 14, 15, 17, 18, 19, 20, 21, 22, to multiply these expectations with SymPy. In particular, with the numbering above, we get

$$\mathbb{E}_{\varepsilon_{trn}, A_{trn}}[(i)] = \frac{d\tau_{\varepsilon_{trn}}^2}{2\tau_{A_{trn}}^2}(T_2 - 1), \quad \mathbb{E}_{\varepsilon_{trn}, A_{trn}}[(ii)] = -\frac{2\theta_{trn}^2 c^2 \tau_{\varepsilon_{trn}}^2}{\gamma} \left( \frac{T_1 - c\mu^2}{T_1^3} \right),$$

$$\mathbb{E}_{\varepsilon_{trn}, A_{trn}}[(iii)] = \frac{c(c+1)\theta_{trn}^4 \tau_{\varepsilon_{trn}}^2}{2\tau_{A_{trn}}^6 \gamma^2} \left( T_2^2 - T_2 - \frac{2c\tau_{A_{trn}}^4}{T_1^2} \right),$$

$$\mathbb{E}_{\varepsilon_{trn}, A_{trn}}[(iv)] = \frac{c\theta_{trn}^2 \tau_{\varepsilon_{trn}}^2}{2\tau_{A_{trn}}^4 \gamma^2} T_2(T_2 - 1).$$

We combine these terms to get the results. The variance follows from concentration. $\qquad\square$

**Theorem 4** (Risk for Signal Only Problem). *Let $\mu \geq 0$ be fixed. Let $\tau_{\varepsilon_{trn}} \asymp 0$, $d/n = c + o(1)$ and $d/n_{tst} = c + o(1)$. Then, any for data $X \in \mathbb{R}^{n \times d}$, $y \in \mathbb{R}^n$ from the signal-only model that satisfy: $1 \ll \tau_{A_{trn}}^2, \tau_{A_{tst}}^2 \ll d$, $\theta_{trn}^2/\tau_{A_{trn}}^2 \ll n$, $\theta_{tst}^2/\tau_{A_{tst}}^2 \ll n_{tst}$. Then for $c < 1$, the instance specific risk is given by*

$$\mathcal{R}(c; \mu, \tau, \theta) = \mathbf{Bias} + \mathbf{Variance_{A_{trn}}} + \mathbf{Variance_{A_{trn}, \varepsilon_{trn}}} + o\left(\frac{1}{d}\right)$$

*with*

$$\mathbf{Bias} = \frac{\theta_{tst}^2}{n_{tst}} \frac{1}{\gamma^2} \left[ (\beta_*^T u)^2 + \frac{\tau_{\varepsilon_{trn}}^2}{2\tau_{A_{trn}}^4} \left( \theta_{trn}^2 c + \tau_{A_{trn}}^2 \right) (T_2 - 1) \right],$$

$$\mathbf{Variance_{A_{trn}}} = \frac{\theta_{trn}^2 \tau_{A_{tst}}^2}{d} \frac{1}{\gamma^2} (\beta_*^T u)^2 \left[ \frac{c \left( \theta_{trn}^2 + \tau_{A_{trn}}^2 \right)}{2\tau_{A_{trn}}^4} (T_2 - 1) \right],$$

$$\mathbf{Variance_{A_{trn}, \varepsilon_{trn}}} = \frac{\tau_{\varepsilon_{trn}}^2 \tau_{A_{tst}}^2}{2\tau_{A_{trn}}^2} \left[ 1 + \frac{c\theta_{trn}^2}{\tau_{A_{trn}}^2} \frac{T_2}{d\gamma^2} \left( \frac{(c+1)\theta_{trn}^2}{\tau_{A_{trn}}^2} + 1 \right) \right] (T_2 - 1)$$

$$- \frac{c^2(c+1)\theta_{trn}^4 \tau_{\varepsilon_{trn}}^2 \tau_{A_{tst}}^2}{d\tau_{A_{trn}}^2} \frac{1}{\gamma^2 T_1^2} - \frac{2c^2 \theta_{trn}^2 \tau_{\varepsilon_{trn}}^2 \tau_{A_{tst}}^2}{d\gamma} \left( \frac{1}{T_1^2} - \frac{c\mu^2}{T_1^3} \right),$$

*where*

$$T_1 = \sqrt{\left( \tau_{A_{trn}}^2 + \mu^2 c - c\tau_{A_{trn}}^2 \right)^2 + 4\mu^2 c^2 \tau_{A_{trn}}^2}, \quad T_2 = \frac{\mu^2 c + \tau_{A_{trn}}^2 + c\tau_{A_{trn}}^2}{T_1},$$

$$\text{and } \gamma = 1 + \frac{\theta_{trn}^2}{2\tau_{A_{trn}}^4} \left( \tau_{A_{trn}}^2 + c\tau_{A_{trn}}^2 + \mu^2 c - T_1 \right).$$

*For $c > 1$, the same formula holds except $T_1 = \sqrt{\left( -\tau_{A_{trn}}^2 + \mu^2 c + c\tau_{A_{trn}}^2 \right)^2 + 4\mu^2 c\tau_{A_{trn}}^2}$.*

*Proof.* The proof follows from the decomposition in Lemma 1.

$$\frac{1}{n_{tst}} \mathbb{E}_{\varepsilon_{trn}, A_{trn}} \left\| \beta_*^T Z_{tst} - \beta_{so}^T Z_{tst} \right\|_F^2 \text{ gives the bias in Proposition1.}$$

Furthermore,

$$\frac{1}{n_{tst}} \frac{\tau_{A_{tst}}^2 n_{tst}}{M} \mathbb{E}_{\varepsilon_{trn}, A_{trn}} \| \beta_{so} \|_F^2 \text{ gives the variance,}$$

where by Lemma 5,

$$\| \beta_{so} \|_F^2 = (\beta_*^T u)^2 \| \tilde{W}_{opt} \|_F^2 + 2\beta_*^T \tilde{W}_{opt}^T (\hat{Z}_{trn} + \hat{A}_{trn})^{\dagger T} \hat{\varepsilon}_{trn} + \hat{\varepsilon}_{trn}^T (\hat{Z}_{trn} + \hat{A}_{trn})^{\dagger} (\hat{Z}_{trn} + \hat{A}_{trn})^{\dagger T} \hat{\varepsilon}_{trn},$$

The second term equals 0 in expectation due to entries of $\varepsilon_{trn}$ having mean 0. The other two terms have expectations given in Propositions 2, 3. $\qquad\square$

## B  PROOF OF THEOREM 3 (SIGNAL PLUS NOISE)

Now with a similar reformulation as A, we can rewrite the signal plus noise problem (no regularization) as follows:

$$\beta_{spn}^T = \arg\min_{\beta^T} \| \beta_*^T (Z_{trn} + A_{trn}) + \varepsilon_{trn}^T - \beta^T (Z_{trn} + A_{trn}) \|_F^2$$

We are interested in the error:

$$\mathcal{R}_{spn}(c; \tau, \theta) = \frac{1}{n_{tst}} \mathbb{E}_{A_{trn}, A_{tst}, \varepsilon_{trn}} \left[ \| \beta_*^T (Z_{tst} + A_{tst}) - \beta_{spn}^T (Z_{tst} + A_{tst}) \|_F^2 \right].$$

**Theorem 3** (Risk for Signal Plus Noise Problem). *Let $\tau_{\varepsilon_{trn}} \asymp 1$, $d/n = c + o(1)$ and $d/n_{tst} = c + o(1)$. Then, for any data $X \in \mathbb{R}^{n \times d}, y \in \mathbb{R}^n$ from the signal-plus-noise model that satisfy: $1 \ll \tau^2_{A_{trn}}, \tau^2_{A_{tst}} \ll d$, $\theta^2_{trn}/\tau^2_{A_{trn}} \ll n$, $\theta^2_{tst}/\tau^2_{A_{tst}} \ll n_{tst}$. Then for $c < 1$, the instance specific risk is given by*

$$
\mathcal{R}_{spn}(c; \tau, \theta) = \left[ \frac{\theta^2_{tst}}{n_{tst}} \frac{1}{(\theta^2_{trn}c + \tau^2_{A_{trn}})} + \frac{\tau^2_{A_{tst}}}{\tau^2_{A_{trn}}} \left( 1 - \frac{\theta^2_{trn}c}{d(\theta^2_{trn}c + \tau^2_{A_{trn}})} \right) \right] \frac{c\tau^2_{\varepsilon_{trn}}}{1 - c} + o\left(\frac{1}{d}\right).
$$

*For $c > 1$, it is given by*

$$
\mathcal{R}_{spn}(c; \tau, \theta) = \|\beta_*\|^2 \left( 1 - \frac{1}{c} \right) \frac{\tau^2_{A_{tst}}}{d} + \frac{\tau^2_{A_{tst}} \tau^2_{\varepsilon_{trn}}}{\tau^2_{A_{trn}}} \left( 1 - \frac{\theta^2_{trn}c}{d(\theta^2_{trn} + \tau^2_{A_{trn}})} \right) \frac{1}{c - 1} + o\left(\frac{1}{d}\right)
$$

$$
+ \frac{\theta^2_{tst} \tau^4_{A_{trn}}}{n_{tst} \left( \theta^2_{trn} + \tau^2_{A_{trn}} \right)^2} \left[ \left( 1 - \frac{1}{c} \right) \left( (\beta^T_* u)^2 + \|\beta_*\|^2 \frac{\theta^2_{trn}}{d\tau^2_{A_{trn}}} \right) + \frac{\tau^2_{\varepsilon_{trn}}}{\tau^4_{A_{trn}}} \left( \frac{\theta^2_{trn}c + \tau^2_{A_{trn}}}{c - 1} \right) \right].
$$

*Proof.* The proof techniques will be similar to A. For simplicity, here we say $A \overset{\mathrm{E}}{=} B$ when their expectations with respect to the random variables are equal. We also suppress the error terms for brevity.

$$
\left\| \beta^T_* (Z_{tst} + A_{tst}) - \beta^T_{spn}(Z_{tst} + A_{tst}) \right\|^2_F
$$

$$
\overset{\mathrm{E}}{=} \left\| \beta^T_* Z_{tst} - \beta^T_{spn} Z_{tst} \right\|^2_F + \left\| \beta^T_* A_{tst} - \beta^T_{spn} A_{tst} \right\|^2_F - 0
$$

$$
\overset{\mathrm{E}}{=} \underbrace{\left\| \beta^T_* Z_{tst} - \beta^T_{spn} Z_{tst} \right\|^2_F}_{bias} + \underbrace{\left\| \beta^T_{spn} A_{tst} \right\|^2_F}_{variance} + \underbrace{\left\| \beta^T_* A_{tst} \right\|^2_F - 2\beta^T_* A_{tst} A^T_{tst} \beta_{spn}}_{adjustment}
$$

In the first equality, the cross term equals 0 in expectation since $A_{tst}$ has mean 0 entries. There is an extra term due to the existence of noise in the signal.

In this setting, the optimal set of parameters is now given by

$$
\beta^T_{spn} = (\beta^T_*(Z_{trn} + A_{trn}) + \varepsilon^T_{trn})(Z_{trn} + A_{trn})^\dagger = \beta^T_{so} + \beta^T_* A_{trn}(Z_{trn} + A_{trn})^\dagger.
$$

With this in mind, we revisit the expectations of terms in the decomposition separately.

**Signal-plus-noise Bias:** For $c < 1$, we adopt similar notations from A (since when we consider $\hat{A}_{trn} \in \mathbb{R}^{n \times (d+n)}$, naturally $n < d + n$ and we are in this case). We define $h = v^T_{trn} A^\dagger_{trm}$, $k = A^\dagger_{trn} u$, $t = v^T_{trn}(I - A^\dagger_{trn} A_{trn})$, $\xi = 1 + \theta_{trn} v^T_{trn} A^\dagger_{trn} u$, $\gamma_1 = \theta^2_{trn} \|t\|^2 \|k\|^2 + \xi^2$, and

$$
p_1 = -\frac{\theta^2_{trn} \|k\|^2}{\xi} t^T - \theta_{trn} k, \ q_1 = -\frac{\theta_{trn} \|t\|^2}{\xi} k^T A^\dagger_{trn} - h.
$$

The same results hold:

$$
\beta^T_* Z_{tst} - \beta^T_{spn} Z_{tst} = \beta^T_* Z_{tst} - \beta^T_{so} Z_{tst} - \beta^T_* A_{trn}(Z_{trn} + A_{trn})^\dagger Z_{tst}
$$

$$
= \frac{\xi}{\gamma_1} \beta^T_* Z_{tst} + \frac{\theta_{tst} \xi}{\theta_{trn} \gamma_1} \varepsilon^T_{trn} p_1 v^T_{tst} - \beta^T_* A_{trn}(Z_{trn} + A_{trn})^\dagger Z_{tst}
$$

by Lemma 3. We then look at the third term. Combining the pseudo-inverse formula A.5, the expansion of $\hat{p}\hat{q}^T$ A.7, and $\hat{A}_{trn}$, we can use a similar approach as in 3 and have some nice cancellations:

$$
-\beta^T_* A_{trn}(Z_{trn} + A_{trn})^\dagger Z_{tst} = -\beta^T_* A_{trn} \left( A^\dagger_{trn} + \frac{\theta_{trn}}{\xi} t^T k^T A^\dagger_{trn} - \frac{\xi}{\gamma_1} p_1 q^T_1 \right) Z_{tst}
$$

$$
= -\theta_{tst} \beta^T_* A_{trn} \left( A^\dagger_{trn} - \frac{\theta^2_{trn} \|t\|^2}{\gamma_1} kk^T A^\dagger_{trn} - \frac{\theta_{trn} \xi}{\gamma_1} kh \right) uv^T_{tst}
$$

$$
= -\theta_{tst} \beta^T_* A_{trn} \left( kv^T_{tst} - \frac{\theta^2_{trn} \|t\|^2 \|k\|^2}{\gamma_1} kv^T_{tst} - \frac{\theta_{trn} \xi}{\gamma_1} khuv^T_{tst} \right)
$$

$$
= -\theta_{tst} \beta^T_* A_{trn} \left( 1 - \frac{\theta^2_{trn} \|t\|^2 \|k\|^2}{\gamma_1} - \frac{\theta_{trn} \xi}{\gamma_1} \frac{\xi - 1}{\theta_{trn}} \right) kv^T_{tst}
$$

$$
= -\theta_{tst} \frac{\xi}{\gamma_1} \beta^T_* A_{trn} A^\dagger_{trn} uv^T_{tst} = -\frac{\xi}{\gamma_1} \beta^T_* Z_{tst}.
$$

Hence, with $\mu = 0$ (see Corollary 1), if $c < 1$, the bias equals

$$\frac{1}{n_{tst}}\left\|\beta_*^T Z_{tst} - \beta_{spn}^T Z_{tst}\right\|_F^2 = \frac{1}{n_{tst}}\left\|\frac{\theta_{tst}\xi}{\theta_{trn}\gamma}\varepsilon_{trn}^T p_1 v_{tst}^T\right\|_F^2 \overset{E}{=} \frac{\theta_{tst}^2 \tau_{\varepsilon_{trn}}^2}{n_{tst}\left(\theta_{trn}^2 c + \tau_{A_{trn}}^2\right)}\left(\frac{c}{1-c}\right).$$

If $c > 1$, now we have dimension $d > n$ and need to further define $s = (I - A_{trn}A_{trn}^\dagger)u$, $\gamma_2 = \theta_{trn}^2 \|s\|^2 \|h\|^2 + \xi^2$, and

$$p_2 = -\frac{\theta_{trn}^2 \|s\|^2}{\xi}A_{trn}^\dagger h^T - \theta_{trn}k, \quad q_2^T = -\frac{\theta_{trn}\|h\|^2}{\xi}s^T - h.$$

We note $s^T u = \|s\|^2$. By Theorem 5 from Meyer (1973), the following pseudo-inverse holds:

$$(Z_{trn} + A_{trn})^\dagger = A_{trn}^\dagger + \frac{\theta_{trn}}{\xi}A_{trn}^\dagger h^T s^T - \frac{\xi}{\gamma_2}p_2 q_2^T.$$

With a similar simplification, we have

$$\beta_*^T Z_{tst} - \beta_{spn}^T Z_{tst} = \beta_*^T Z_{tst} - \beta_{so}^T Z_{tst} - \beta_*^T A_{trn}(Z_{trn} + A_{trn})^\dagger Z_{tst}$$

$$= \frac{\xi}{\gamma_2}\beta_*^T Z_{tst} + \frac{\theta_{tst}\xi}{\theta_{trn}\gamma_2}\varepsilon_{trn}^T p_2 v_{tst}^T - \beta_*^T A_{trn}(Z_{trn} + A_{trn})^\dagger Z_{tst}.$$

Furthermore, we recall expressions of the defined variables, and the third become can be simplified as:

$$\beta_*^T A_{trn}(Z_{trn} + A_{trn})^\dagger Z_{tst} = \theta_{tst}\beta_*^T A_{trn}\left(A_{trn}^\dagger u v_{tst}^T + \frac{\theta_{trn}}{\xi}A_{trn}^\dagger h^T s^T u v_{tst}^T - \frac{\xi}{\gamma_2}p_2 q_2^T u v_{tst}^T\right)$$

$$= \theta_{tst}\beta_*^T A_{trn}\left(k v_{tst}^T + \frac{\theta_{trn}\|s\|^2}{\xi}A_{trn}^\dagger h^T v_{tst}^T - \frac{\xi}{\gamma_2}p_2 q_2^T u v_{tst}^T\right)$$

$$= \theta_{tst}\beta_*^T A_{trn}\left(-\frac{1}{\theta_{trn}}p_2 v_{tst}^T - \frac{\xi}{\gamma_2}p_2\left(-\frac{\theta_{trn}\|h\|^2}{\xi}s^T - h\right)u v_{tst}^T\right)$$

$$= \theta_{tst}\beta_*^T A_{trn}\left(-\frac{1}{\theta_{trn}}p_2 v_{tst}^T + \frac{\xi}{\gamma_2}p_2\left(\frac{\theta_{trn}\|s\|^2\|h\|^2}{\xi} + \frac{\xi-1}{\theta_{trn}}\right)v_{tst}^T\right)$$

$$= \theta_{tst}\beta_*^T A_{trn}\left(-\frac{1}{\theta_{trn}}p_2 v_{tst}^T + \frac{\xi}{\gamma_2}p_2\left(\frac{\theta_{trn}^2\|s\|^2\|h\|^2 + \xi^2 - \xi}{\xi\theta_{trn}}\right)v_{tst}^T\right)$$

$$= \theta_{tst}\beta_*^T A_{trn}\left(-\frac{1}{\theta_{trn}}p_2 v_{tst}^T + \frac{\xi}{\gamma_2}p_2\left(\frac{\gamma_2 - \xi}{\xi\theta_{trn}}\right)v_{tst}^T\right)$$

$$= -\frac{\theta_{tst}\xi}{\theta_{trn}\gamma_2}\beta_*^T A_{trn}p_2 v_{tst}^T.$$

Hence, we have that

$$\left\|\beta_*^T Z_{tst} - \beta_{spn}^T Z_{tst}\right\|_F^2 = \left\|\frac{\xi}{\gamma_2}\beta_*^T Z_{tst} + \frac{\theta_{tst}\xi}{\theta_{trn}\gamma_2}(\beta_*^T A_{trn} + \varepsilon_{trn}^T)p_2 v_{tst}^T\right\|_F^2$$

$$\overset{E}{=} \left\|\frac{\xi}{\gamma_2}\beta_*^T Z_{tst}\right\|_F^2 + \left\|\frac{\theta_{tst}\xi}{\theta_{trn}\gamma_2}\varepsilon_{trn}^T p_2 v_{tst}^T\right\|_F^2 + \left\|\frac{\theta_{tst}\xi}{\theta_{trn}\gamma_2}\beta_*^T A_{trn}p_2 v_{tst}^T\right\|_F^2$$

$$+ \frac{2\theta_{tst}\xi^2}{\theta_{trn}\gamma_2^2}\beta_*^T A_{trn}p_2 v_{tst}^T Z_{tst}^T \beta_*.$$

The first two expectations are given in Corollary 1 (the bias). We compute expectations for the last two additional terms here. Similar to Lemma 17, we have $kA_{trn}^\dagger h^T \overset{E}{=} 0$ in this case. We recall the expression of $p_2$, $q_2$ and obtain

$$\left\|\frac{\theta_{tst}\xi}{\theta_{trn}\gamma_2}\beta_*^T A_{trn}p_2 v_{tst}^T\right\|_F^2 = \frac{\theta_{tst}^2\xi^2}{\theta_{trn}^2\gamma_2^2}\beta_*^T A_{trn}p_2 p_2^T A_{trn}^T \beta_*$$

$$\overset{E}{=} \frac{\theta_{tst}^2\xi^2}{\theta_{trn}^2\gamma_2^2}\beta_*^T A_{trn}\left(\frac{\theta_{trn}^4\|s\|^4}{\xi^2}A_{trn}^\dagger h^T h A_{trn}^{\dagger T} + \theta_{trn}^2 kk^T\right)A_{trn}^T \beta_*$$

$$\overset{E}{=} \frac{\theta_{tst}^2\theta_{trn}^2\|s\|^4}{\gamma_2^2}\beta_*^T A_{trn}A_{trn}^\dagger h^T h A_{trn}^{\dagger T}A_{trn}^T \beta_* + \frac{\theta_{tst}^2\xi^2}{\gamma_2^2}\beta_*^T A_{trn}kk^T A_{trn}^T \beta_*$$

where the cross terms have zero expectation. Taking general variances into account, from Lemma from Sonthalia & Nadakuditi (2023), we have that

$$\|s\|^2 \overset{E}{\cong} \frac{c-1}{c}, \ \xi \overset{E}{\cong} 1, \ \gamma_2 \overset{E}{\cong} \frac{\theta_{trn}^2 + \tau_{A_{trn}}^2}{\tau_{A_{trn}}^2}, \ \frac{1}{\lambda} \overset{E}{\cong} \frac{c}{\tau_{A_{trn}}^2 (c-1)}.$$

Furthermore, we turn to the SVD of the noise matrix ($A_{trn} = U\Sigma V^T$, $\Sigma \in \mathbb{R}^{d \times n_{trn}}$) to evaluate these expectations. We define $a = U^T u$, $b = U^T \beta_*$, $c = V^T v_{trn}$. Since $U$ and $V$ are uniformly random orthogonal matrices independent from each other, these vectors are centered and uniformly random. $a$, $b$ are unit vectors, and $c$ has length $\|\beta_*\|$. We expand the following two terms using SVD:

$$\beta_*^T A_{trn} A_{trn}^\dagger h^T h A_{trn}^{\dagger T} A_{trn}^T \beta_* = (\beta_*^T U \Sigma\Sigma^\dagger \Sigma^{\dagger T} V^T v_{trn})^2$$

$$= \left( \beta_*^T U \begin{bmatrix} I_{n_{trn}} & 0 \\ 0 & 0 \end{bmatrix} \Sigma^{\dagger T} V^T v_{trn} \right)^2$$

$$= \sum_{i=1}^{n_{trn}} a_i^2 b_i^2 \frac{1}{\sigma_i^2} + \text{ cross terms}$$

$$\overset{E}{\cong} n_{trn} \frac{\|\beta_*\|^2}{d} \frac{1}{n_{trn}} \frac{c}{\tau_{A_{trn}}^2 (c-1)} = \frac{\|\beta_*\|^2}{d} \frac{c}{\tau_{A_{trn}}^2 (c-1)}$$

$$\beta_*^T A_{trn} k k^T A_{trn}^T \beta_* = (\beta_*^T U \Sigma\Sigma^\dagger U^T u)^2 = \left( \beta_*^T U \begin{bmatrix} I_{n_{trn}} & 0 \\ 0 & 0 \end{bmatrix} U^T u \right)^2 \overset{E}{\cong} \frac{(\beta_*^T u)^2}{c}$$

In the first term, the cross terms have zero expectation due to the centered uniform vectors. In the second term, the terms in the middle do not change the alignment between $W$ and $u$, so we have $W^T u$. Putting everything together, we have

$$\left\| \frac{\theta_{tst} \xi}{\theta_{trn} \gamma_2} \beta_*^T A_{trn} p_2 v_{tst}^T \right\|_F^2 \overset{E}{\cong} \frac{\theta_{tst}^2 \tau_{A_{trn}}^4}{c(\theta_{trn}^2 + \tau_{A_{trn}}^2)^2} (\beta_*^T u)^2 + \frac{\theta_{tst}^2 \theta_{trn}^2 \tau_{A_{trn}}^2}{(\theta_{trn}^2 + \tau_{A_{trn}}^2)^2} \left( 1 - \frac{1}{c} \right) \frac{\|\beta_*\|^2}{d}.$$

With a similar approach, we now look at the last term:

$$\frac{2\theta_{tst} \xi^2}{\theta_{trn} \gamma_2^2} \beta_*^T A_{trn} p_2 v_{tst}^T Z_{tst}^T \beta_* = \frac{2\theta_{tst}^2 \xi^2}{\theta_{trn} \gamma_2^2} \beta_*^T A_{trn} \left( -\frac{\theta_{trn}^2 \|s\|^2}{\xi} A_{trn}^\dagger h^T - \theta_{trn} k \right) u^T \beta_*$$

$$\overset{E}{\cong} -\frac{2\theta_{tst}^2 \xi^2}{\gamma_2^2} \beta_*^T A_{trn} k u^T \beta_*.$$

Here the first term becomes 0 in expectation since

$$\beta_*^T A_{trn} A_{trn}^\dagger h^T u^T \beta_* = \beta_*^T \beta_*^T U \Sigma\Sigma^\dagger \Sigma^{\dagger T} V^T v_{trn} u^T \beta_*,$$

where $V^T v_{trn}^T$ is centered and uniformly random. Lastly,

$$\beta_*^T A_{trn} k u^T \beta_* = \beta_*^T U \Sigma\Sigma^\dagger U^T u u^T \beta_* \overset{E}{\cong} \frac{(\beta_*^T u)^2}{c},$$

$$\rightarrow \frac{2\theta_{tst} \xi^2}{\theta_{trn} \gamma_2^2} \beta_*^T A_{trn} p_2 v_{tst}^T Z_{tst}^T \beta_* \overset{E}{\cong} -\frac{2\theta_{tst}^2 \tau_{A_{trn}}^4}{c(\theta_{trn}^2 + \tau_{A_{trn}}^2)^2} (\beta_*^T u)^2.$$

Now we have all these terms. If $c > 1$, the bias equals

$$\frac{1}{n_{tst}} \left\| \beta_*^T Z_{tst} - \beta_{spn}^T Z_{tst} \right\|_F^2 \overset{E}{\cong}$$

$$\frac{\theta_{tst}^2}{n_{tst} (\theta_{trn}^2 + \tau_{A_{trn}}^2)^2} \left( \tau_{A_{trn}}^2 \left( 1 - \frac{1}{c} \right) \left( \tau_{A_{trn}}^2 (\beta_*^T u)^2 + \theta_{trn}^2 \frac{\|\beta_*\|^2}{d} \right) + \tau_{\varepsilon_{trn}}^2 \left( \frac{\theta_{trn}^2 c + \tau_{A_{trn}}^2}{c-1} \right) \right).$$

**Signal-plus-noise Variance:** By Lemma 4,

$$\left\|\beta_{spn}^T A_{tst}\right\|_F^2 \overset{\mathrm{E}}{=} \frac{\tau_{A_{tst}}^2 n_{tst}}{d} \|\beta_{spn}\|_F^2$$

$$= \frac{\tau_{A_{tst}}^2 n_{tst}}{d} (\beta_*^T (Z_{trn} + A_{trn}) + \varepsilon_{trn}^T)(Z_{trn} + A_{trn})^\dagger (Z_{trn} + A_{trn})^{\dagger T} (\beta_*^T (Z_{trn} + A_{trn}) + \varepsilon_{trn}^T)^T$$

$$\overset{\mathrm{E}}{=} \frac{\tau_{A_{tst}}^2 n_{tst}}{d} \beta_*^T (Z_{trn} + A_{trn})(Z_{trn} + A_{trn})^\dagger (Z_{trn} + A_{trn})^{\dagger T} (Z_{trn} + A_{trn})^T \beta_*$$

$$+ \frac{\tau_{A_{tst}}^2 n_{tst}}{d} \varepsilon_{trn}^T (Z_{trn} + A_{trn})^\dagger (Z_{trn} + A_{trn})^{\dagger T} \varepsilon_{trn}.$$

Again the cross term is 0 in expectation due to mean 0 entries of $\varepsilon_{trn}$. Proposition 2 gives the expectation of the second term (we make $\mu \to 0$). The first expectation is

$$\beta_*^T (Z_{trn} + A_{trn})(Z_{trn} + A_{trn})^\dagger (Z_{trn} + A_{trn})^{\dagger T}(Z_{trn} + A_{trn})^T \beta_* \overset{\mathrm{E}}{=} \min\left(1, \frac{1}{c}\right)\|\beta_*\|^2.$$

With $\mu = 0$ results (see Corollary 1), If c < 1, the variance equals

$$\frac{1}{n_{tst}}\left\|\beta_{spn}^T A_{tst}\right\|_F^2 \overset{\mathrm{E}}{=} \frac{\tau_{A_{tst}}^2 \tau_{\varepsilon_{trn}}^2 c}{\tau_{A_{trn}}^2(1-c)}\left(1 - \frac{\theta_{trn}^2 c}{d\left(\tau_{A_{trn}}^2 + \theta_{trn}^2 c\right)}\right) + \frac{\tau_{A_{tst}}^2}{d}\|\beta_*\|^2.$$

If c > 1, the variance equals

$$\frac{1}{n_{tst}}\left\|\beta_{spn}^T A_{tst}\right\|_F^2 \overset{\mathrm{E}}{=} \frac{\tau_{A_{tst}}^2 \tau_{\varepsilon_{trn}}^2}{\tau_{A_{trn}}^2(c-1)}\left(1 - \frac{\theta_{trn}^2 c}{d\left(\tau_{A_{trn}}^2 + \theta_{trn}^2\right)}\right) + \frac{1}{c}\frac{\tau_{A_{tst}}^2}{d}\|\beta_*\|^2.$$

**Further Adjustment:** By Lemma 4, we see that

$$\|\beta_*^T A_{tst}\|_F^2 \overset{\mathrm{E}}{=} \frac{\tau_{A_{tst}}^2 n_{tst}}{d}\|\beta_*\|^2, \text{ and}$$

$$\beta_*^T A_{tst} A_{tst}^T \beta_{spn} \overset{\mathrm{E}}{=} \beta_*^T A_{tst} A_{tst}^T (Z_{trn}+A_{trn})^{\dagger T}(Z_{trn}+A_{trn})^T \beta_* \overset{\mathrm{E}}{=} \min\left(1, \frac{1}{c}\right)\frac{\tau_{A_{tst}}^2 n_{tst}}{d}\|\beta_*\|^2,$$

If c < 1, the adjustment equals

$$\frac{1}{n_{tst}}\left(\left\|\beta_*^T A_{tst}\right\|_F^2 - 2\beta_*^T A_{tst} A_{tst}^T \beta_{spn}\right) \overset{\mathrm{E}}{=} -\frac{\tau_{A_{tst}}^2}{d}\|\beta_*\|^2.$$

If c > 1, the adjustment equals

$$\frac{1}{n_{tst}}\left(\left\|\beta_*^T A_{tst}\right\|_F^2 - 2\beta_*^T A_{tst} A_{tst}^T \beta_{spn}\right) \overset{\mathrm{E}}{=} \left(1 - \frac{2}{c}\right)\frac{\tau_{A_{tst}}^2}{d}\|\beta_*\|^2.$$

Putting things together from the three separate terms, we get the results.

$\square$

