# OpenReview forum: "Generalization for Least Squares Regression with Simple Spiked Covariances"
_ICLR.cc/2025/Conference — Submitted to ICLR 2025_

### Official Review · Reviewer_YeXr · 2024-10-21

**Soundness:** 3
**Presentation:** 4
**Contribution:** 3
**Rating:** 6
**Confidence:** 3

**Summary:**

Motivated by the problem of training the readout of a two-layer network after on large gradient step on the first layer, the authors consider the problem of linear regression on a spiked data model. They provide a characterization of the test error, for two linear target functions, respectively depending on only the spike part or the complete input. They discuss how for the latter, the spike does not asymptotically influence the test error, but does non-asymptotically.

**Strengths:**

The paper is well written, clearly motivating the study, and describing in simple terms the model considered. Sufficient intuition is provided at most steps of the discussion. The technical results are clearly exposed and sufficiently discussed. Although I did not go through the proof in detail, the technical results seem scientifically sound.

**Weaknesses:**

I have a number of concerns related to the technical discussions, and the relation to previous works, which I detail below, and in the question section. These concerns regard the discussion of the main theorems, and not the theorems themselves, although I have not carefully verified the proof. These concerns prevent me from giving a higher score to this submission. On the other hand, I would be very happy to increase my score, were those concerns to be addressed by the authors.

- The authors claim l.083 that Moniri et al. (2023) do not quantify the test error after one gradient step. To the best of my understanding, they do provide such a tight characterization (Theorem 4.5). Could the authors clarify their claim, and emphasize how their work is positioned with respect to Moniri et al. (2023)?

- I find the discussion l.332-355 somewhat confusing, as they discuss the specialization of Theorem 3 for $\theta=\tau\sqrt{n}$. Doesn't this directly contradict the assumption $\theta^2/\tau^2\ll n$ stated in the Theorem? Since this specialization leads to one of the main qualitative results of the paper (namely the spike only affects the test error in non-asymptotic cases), this point would gain to be clarified.  The same holds for l.452 with respect to Theorem 4.

- I believe more intuition about the different scaling considered would help solidify the intuition for the spn case, regarding when the spike matters. In particular, the authors could for instance recall the scaling of the terms $z_i^\top\beta_*$, $a_i^\top \beta_*$, and emphasize their respective strengths in the different scalings of $\theta, \tau$ considered. I am curious if the signal part is much smaller than the second term when the spike has no effect, or if the argument is more subtle.

**Questions:**

- In the discussion above 4.1, the assumption $\theta=\tau^2 n$ seems again to contradict the assumption $\theta^2/\tau^2\ll n$ in the statement of the Theorem. Furthermore, this seems to correspond to a strong spike regime, how can the authors recover the spikeless results of Hastie et al. (2022) in this case? This might be a misunderstanding on my side, but more discussion would be beneficial.

Minor:

- l.074, 347 : incomplete sentences
- l.203 I think $\ell_j$ is not defined
- In 3.3, more discussion in the main text about why only the unregularized case is considered for the spn case, while generic $\mu$ is considered for the signal-only model, would be helpful for intuition, whether it is for technical reasons or because it is not interesting.

---

> ### Author Response · Authors · 2024-11-14
>
> > The authors claim l.083 that Moniri et al. (2023) do not quantify the test error after one gradient step. To the best of my understanding, they do provide such a tight characterization (Theorem 4.5). Could the authors clarify their claim, and emphasize how their work is positioned with respect to Moniri et al. (2023)?
>
> We should have been more precise in our claim. While Theorem 4.5 in Moniri et al. 2023 shows that the difference between the test error using $F_1$ (features after 1 step) and $F_0$ (initial features) converges to a constant, there are important differences in our approach:
>
> 1. Moniri et al. 2023 have expressions requiring solutions to fixed point equations (see equations (5) in their paper). These only hold asymptotically with hard-to-quantify approximation rates.
>
> 2. In contrast, we provide:
>
>    - **Closed form expressions** for the risk itself (not just differences as is the case in Moniri et al. 2023)
>
>    - **Better control on approximation error rates**, enabling analysis of finite matrices
>
>    - The above two allow better control in understanding the relationship between the bulk and spike.
>
> > I find the discussion l.332-355 somewhat confusing
>
> We apologize for the unclear notation. We use the Vinogradov notation where $f \ll g$ means $f = O(g)$. Therefore, setting $\theta^2 = \tau^2 n$ is consistent with our assumptions.
>
> > I believe more intuition about the different scaling considered would help solidify the intuition for the spn case
>
> You raise an excellent point. Let's clarify the scaling relationships:
>
> 1. For the bulk term: $a_i^T \beta_* \approx \tau_A \|\beta_*\|$
>
> 2. For the spike term: $z_i^T \beta_* \approx \theta\|\beta_*\|$
>
> Using our scaling $\theta = \tau \sqrt{n}$, the signal part is always larger. However, when the bulk also grows (i.e., $\tau_A = \Theta(d)$), the spike's effect becomes invisible. Specifically, we need:
>
> - $a_i^T \beta_* = \Theta(1)$
>
> - $z_i^T \beta_* = \Theta(\sqrt{n})$
>
> (assuming $\|\beta_*\|= \Theta(1)$) for the spike to have a detectable effect.
>
> > In 3.3, more discussion in the main text about why only the unregularized case is considered for the spn case, while generic  is considered for the signal-only model, would be helpful for intuition, whether it is for technical reasons or because it is not interesting.
>
> This limitation is primarily technical. The regularized case for the signal-plus-noise model introduces many additional terms whose mean and variance would need to be bounded, significantly complicating the analysis.
>
> > typos
>
> Thank you for identifying these issues. We will correct all typos in the revision.

---

> ### Comment · Reviewer_YeXr · 2024-11-25
> **Acknowledgement of rebuttal**
>
> I thank the authors for taking the time to provide all the detailed clarifications and answers to my interrogations. I think the paper is scientifically sound, and thus increase slightly my score, although I have not checked the proofs.

---

> > ### Author Response · Authors · 2024-11-26
> >
> > We thank the reviewer for their valuable feedback and for increasing their score

---

### Official Review · Reviewer_xFzE · 2024-10-24

**Soundness:** 2
**Presentation:** 2
**Contribution:** 2
**Rating:** 5
**Confidence:** 3

**Summary:**

This paper analyses the generalization error of linear regression with spiked covariance. Previous literature has been using asymptotic limit of the empirical spectral density to analyse the generalization error of linear regression. At the limit, the effect of the spike vanishes. However, it is not the case for finite sample size. This paper fills the gap by showing there is a correction term for finite sample size $n$.

**Strengths:**

This paper provides a detailed proof of their main theorems with clearly stated definitions. It extends over previous results like [1,2].


[1]: Trevor Hastie, Andrea Montanari, Saharon Rosset, and Ryan J Tibshirani. Surprises in highdimensional
ridgeless least squares interpolation. Annals of statistics, 50(2):949, 2022.
[2]: Xinyue Li and Rishi Sonthalia. Least squares regression can exhibit under-parameterized double
descent. Advances in Neural Information Processing Systems, 2024.

**Weaknesses:**

However, this paper has some obvious weaknesses:

1. The paper is motivated by the spiked covariance from the one-step gradient feature learning in neural networks (Section 1). However, it did not show how the results can be applied to the feature learning scenario. I question the amount of contribution this paper provides.
2. The assumption in line 2221-222 and 253-255 is too strong. The analysis breaks down if there is dependency in the cross term. However, the paper did not show how big the difference the predicted result would be when there is dependence in the cross term. It is questionable if the result in this paper is applicable in realistic machine learning settings.
3. This paper has problems with the wordings, even in main theorems. This makes the reading difficult. For instance:

> Theorem 3 (line 313): ...Then, any for data $X\in\mathbb{R}^{n\times d}, y\in\mathbb{R}^n$ from the signal-plus-noise model that satisfy: $1\ll \tau\_{A_{trn}}^2,\tau_{A_{tst}}^2\ll d, \theta_{trn}^2/\tau_{A_{trn}}^2<<n, \theta_{tst}^2/\tau_{A_{tst}}^2 << n_{tst}$. Then for $c<1$,...

The first sentence needs to be rephrased and the symbol $\ll$ is not consistent. Also, there are some typos like:
> line 350: Hence the spike has does not have an effect...

> line 372, ... we see an affect that...

**Questions:**

Regarding the weaknesses mentioned above, I would like to ask:

1. What novel results could the authors conclude in the feature learning setting in neural networks using the main theorems 3,4?

2. How could the authors show the assumption on the dependence does not affect the result? Is there any experimental validation?

3. From Figure 4.1, we can see that the effect of the spike correction term is small when $n$ is large. Is the main theorem still useful to explain the phenomenon we see from feature learning?

---

> ### Author Response · Authors · 2024-11-14
>
> We thank the reviewer for the feedback.
>
> > The paper is motivated by the spiked covariance from the one-step gradient feature learning in neural networks (Section 1). However, it did not show how the results can be applied to the feature learning scenario. I question the amount of contribution this paper provides.
>
> Prior work (Moniri et al. 2023 and Ba et al. 2022) established the existence of spikes for specific target types $y$ (single index models). Our work focuses on understanding the spike's effect through several novel contributions:
>
> 1. We analyze various targets and alignments between spikes and targets, providing rigorous proofs of generalization bounds
>
> 2. We demonstrate how asymptotic results may not capture behavior in finite networks
>
> 3. We provide precise quantification of how spikes affect generalization in both finite and asymptotic regimes
>
> > The assumption in line 2221-222 and 253-255 is too strong. The analysis breaks down if there is dependency in the cross term. However, the paper did not show how big the difference the predicted result would be when there is dependence in the cross term. It is questionable if the result in this paper is applicable in realistic machine learning settings.
>
> We have now removed this assumption. When considering the dependence structure from Moniri et al. 2023, the analysis actually simplifies. A key term in our analysis is the projection of the spike direction on the bulk eigenvectors. Due to the dependence, this term becomes more tractable. Please see the general response.
>
> > What novel results could the authors conclude in the feature learning setting in neural networks using the main theorems 3,4?
>
> Our analysis reveals several important insights for feature learning:
>
> 1. Relationship between targets and spike size:
>
>    - For targets depending on the bulk (input data), large spikes are crucial
>
>    - The required spike size scales with input dimension and dataset size
>
> 2. Importance of spike-target alignment:
>
>    - The alignment between spike direction and targets significantly affects generalization
>
>    - This alignment term exhibits its own double descent behavior
>
>    - Small alignment improvements can yield large generalization gains
>
> 3. Double descent characteristics:
>
>    - Peak location depends on bulk variance and regularization strength
>
>    - Suggests weight decay regularization primarily affects the bulk, not learned features (spike)
>
> While some of these phenomena have been observed before, we provide simplified, quantitative connections between them.
>
> > How could the authors show the assumption on the dependence does not affect the result? Is there any experimental validation?
>
> We provide new theoretical results that explicitly handle the dependence structure. Please see the general response.
>
> > From Figure 4.1, we can see that the effect of the spike correction term is small when is large. Is the main theorem still useful to explain the phenomenon we see from feature learning?
>
> Yes - prior work shows that the spike represents the learned feature. Our results allow for larger spikes than previously considered in works like Hastie et al. 2022. However, this shows that for the spike to effect the generalization, we need even bigger spikes.
>
> > This paper has problems with the wordings, even in main theorems. This makes the reading difficult. For instance:
>
> Thank you for identifying these issues. We will fix all typos and improve clarity in the revised version.

---

> > ### Comment · Reviewer_xFzE · 2024-11-20
> >
> > Thank you for your detailed reply. I will raise my score accordingly.

---

> > > ### Author Response · Authors · 2024-11-21
> > >
> > > We thank the reviewer for the discussion and for increasing their score.
> > >
> > > If there are more aspects of the work that the reviewer would like to discuss, we would be delighted to continue the discussion.

---

### Official Review · Reviewer_LgJ3 · 2024-11-01

**Soundness:** 3
**Presentation:** 3
**Contribution:** 2
**Rating:** 5
**Confidence:** 4

**Summary:**

The paper considers the linear least squares regression for data with simple spiked covariance. They quantify the empirical risk of test data.

**Strengths:**

1. They construct two linear regression problems with spiked covariance.
2. They well explain the previous work of Moniri et al. (2023).
3. Precise quantification of the generalization errors are also provided for both model.

**Weaknesses:**

1. They reference the work of Moniri et al., but this work is unrelated to neural networks or gradient descent; it addresses a purely linear regression problem for data with simple spiked covariances.

2. They do not account for the generalization ability of neural networks after a single gradient step, as they bypass the gradient step entirely by assuming the W1 matrix directly, which does not reflect the full process of neural network training.

**Questions:**

1. Could you provide a reference for the statement, 'It has been shown that to understand the generalization...' on line 39?
2. Is your generalization analysis very different from the work  of Li & Sonthalia (2024)?

---

> ### Author Response · Authors · 2024-11-14
>
> We thank the reviewer for their comments.
>
> > They reference the work of Moniri et al., but this work is unrelated to neural networks or gradient descent; it addresses a purely linear regression problem for data with simple spiked covariances.
>
> We respectfully disagree. Our work is directly motivated by and connected to neural networks through the following chain of reasoning:
> 1. Ba et al. (2022) and Moniri et al. (2023) show that after one gradient step, the feature matrix $F_1$ can be written as $ F_0 + P$, where $P$ is a rank-$ell$ matrix.
> 2. This creates a spiked covariance structure in $F_1^TF_1$.
> 3. To understand the generalization error of such networks, we need to analyze least squares regression with $F_1$ as the feature matrix.
> 4. Our work studies this exact setting, though in a simplified form, to make the analysis tractable.
> 5. In the general rebuttal, we removed many of our simplifications. This further strengthens the connection
>
> > They do not account for the generalization ability of neural networks after a single gradient step, as they bypass the gradient step entirely by assuming the W1 matrix directly, which does not reflect the full process of neural network training.
>
> We agree with the reviewer. Building on the results from Ba et al. 2022 and Moniri et al. 2023, our new results take us towards understanding the generalization error for two-layer networks.
>
> Our analysis provides valuable insights:
> - It shows how spikes affect generalization in finite vs asymptotic regimes
> - It demonstrates the importance of the alignment between the spike direction and the target function
>
> We never meant our paper to claim that we understood the generalization error for two-layer networks, as reflected in our title's focus on least squares regression.
>
> > Could you provide a reference for the statement, 'It has been shown that to understand the generalization...' on line 39?
>
> In addition to the RMt paper cited in the paper, see [1] for an empirical result on more realistic networks.
>
> [1] Martin and Mahoney, "Implicit Self-Regularization in Deep Neural Networks: Evidence from Random Matrix Theory and Implications for Learning", JMLR 2021.
>
> > Is your generalization analysis very different from the work of Li & Sonthalia (2024)?
>
> Compared to Li and Sonthalia 2024. Only one of their two models allows for an eigenvalue to diverge. This model is closely related to the Signal only model in this paper. However, we have output noise $\varepsilon$. This creates many new dependencies requiring novel analysis techniques.
>
> 1. Consider the proof sketch on page 10. Line 491 shows that for the signal-only problem, our solution is the solution from the Li and Sonthalia 2024 paper plus an extra term. The $\hat{\varepsilon}$ is not isotropic in the ridge regularied version. Hence, we do not immediately have free independence between $\hat{Z}+\hat{A}$ and $\hat{\varepsilon}$. Hence, we need to be very careful about the alignment between the two.
>
> 2. Hence, we get terms that are cubic and quartic in eigenvalues (vs quadratic in prior work). These required the development of new concentration bounds for these higher-order terms. See Lemmas 17 through 22.
>
> The Li and Sonthalia 2024 paper does not consider the signal plus noise case, which introduces further terms that need bounding. Finally, the Li and Sonthalia 2024 only consider the case when $\tau_A = 1$ or more broadly $\tau_A = \Theta(1)$, we allow $\tau_A = \Theta(\sqrt{d})$. **This is also significant**.

---

> > ### Comment · Reviewer_LgJ3 · 2024-11-24
> >
> > Thanks so much for all your careful reply and the new updated version draft! I'll raise my score a little bit!

---

> > > ### Author Response · Authors · 2024-11-25
> > >
> > > We thank the reviewer for increasing their score and valuable contributions

---

### Official Review · Reviewer_qVHv · 2024-11-01

**Soundness:** 2
**Presentation:** 1
**Contribution:** 1
**Rating:** 3
**Confidence:** 4

**Summary:**

Motivated by a recent work studying two-layer neural networks (Moniri et al., 2023), the paper studies linear regression under a data model with a spiked covariance (Couillet & Liao, 2022). The spiked covariance consists of a spike component (signal) and a bulk component (noise). Thus, the authors characterize the risk (a.k.a generalization error) with a specific focus on the effect of the spike. They find that the spike does not impact the risk in the underparameterized case. In contrast, the spike introduces an additional term (called "correction term") in the risk for the overparameterized case. However, they mention that the correction term is of order $O(1/d)$, which vanishes in the asymptotic case. Thus, the spike does not affect the risk in the asymptotic case but does in the finite case. Then, the authors focus on a case where the targets $y$ only depend on the signal (spike) component of inputs $\mathbf{x}$ in order to highlight the effect of the spike on the risk. In this case, the correction term depends on the alignment between the spiked eigenvector $\mathbf{u}$ corresponding to the spike and the target function $\boldsymbol{\beta}$. Furthermore, the paper illustrates how the generalization error for this setting exhibits the so-called double-descent phenomenon with a formula for the peak location (a.k.a interpolation threshold).

**Strengths:**

* The motivation for this paper is good since the recent line of work studying two-layer neural networks after one gradient step (Ba et al., 2022; Moniri et al., 2023) has received significant attention.
* The authors precisely characterize generalization errors (risk) for two linear regression problems with spiked covariance data, while the problems differ regarding the target function.
   + They provide bias and variance decomposition of the risk.
   + They illustrate the "double-descent phenomenon" and provide a formula for the peak location (a.k.a interpolation threshold) of the double-descent phenomenon, which is beneficial for understanding the phenomenon.
   + The authors specifically focus on the impact of the spike (in the data model) on the risk for different cases. Thus, they show when and how the spike affects the generalization error.

**Weaknesses:**

* The presentation in this paper is not good
  + Although the paper is motivated by Moniri et al. (2023), there are significant discrepancies between the setting of this paper and that of Moniri et al. (2023), as the authors mention in Section 5. While Moniri et al. (2023) considered two-layer neural networks after one gradient step under isotropic data assumption, this work considers linear regression under spiked covariance data assumption. There exists a relationship between these two, but they are not exactly the same. For example, there is a difference between the target $y$ generation of the two settings. Furthermore, $\mathbf{A}$ (noise component) and $\mathbf{Z}$ (spike component) are dependent in the case of Moniri et al. (2023), while the dependence is ignored here (see lines 251-255).
  + Some notations are used without definition (e.g, $\delta_{\lambda_i}(\lambda)$ in Line 126, or $\Sigma(d_k)$ in Line 147).
  + There are significant typos in equations. For example, $y$ should be a scaler in Line 76, but it is written as a vector, which makes the equation wrong. Another example is that $l_j$ in Theorem 2 (Line 204) is not defined, and I think the authors meant $l$ instead of $l_j$. A third example is that the function $R_{spn}(c;\tau,\theta)$ defined in Line 301 and its usage $R_{spn}(c,0,\tau)$ in Theorem 3 (Line 317-321) are different in terms of parameters.

* Limited contribution/novelty
  + Most of the results in this paper are trivial extensions of the results by Hastie et al. (2022) and Li & Sonthalia (2024), which significantly limits the novelty and originality of the paper. Note that Hastie et al. (2022) studied linear regression under a generic covariance assumption with bounded eigenvalues. Here, some eigenvalues can diverge as dimensions go to infinity, but this case is also covered by Li & Sonthalia (2024).
  + There exists a related work (Cui et al., 2024) that is not mentioned in this paper. Cui et al. (2024) characterized the generalization error (risk) for two-layer neural networks after one gradient step under isotropic data (same setting as that of Moniri et al. (2023)). Although there exist methodological differences between (Cui et al., 2024) and this paper, the motivations are the same, and their settings are similar.
  + During the review period of this paper, a related work (Dandi et al., 2024) that can be considered as follow-up of (Cui et al., 2024) was appeared on arXiv. While Cui et al. (2024) used (non-rigorous) replica method from statistical physics for their analysis, Dandi et al. (2024) studied the same setting with random matrix theory, which is also the main tool in this paper. Therefore, this paper and (Dandi et al. 2024) studied similar settings with similar methodologies. Note that since (Dandi et al., 2024) appeared after the submission of this paper, I am only mentioning it for the sake of completeness.

Overall, I think this paper should be rewritten with more focus on the impacts of the spike covariance on the generalization error of linear regression, and the new presentation should clearly differentiate the current work from the work by Hastie et al. (2022), Li & Sonthalia (2024), Cui et al. (2024), and Dandi et al. (2024).

Cui et al. (2024): Asymptotics of feature learning in two-layer networks after one gradient-step. (ICML 2024)

Dandi et al. (2024): A Random Matrix Theory Perspective on the Spectrum of Learned Features and Asymptotic Generalization Capabilities.

**Questions:**

1. In Line 178, $F_1$ denotes the case with a single spike (as shown by Moniri et al., 2023). However, Moniri et al., 2023 showed that $F_1$ can include multiple spikes, and the number of spikes depends on the step size of the gradient step. Where is the discussion about the effect of step size in this paper? Similarly, where is the discussion on the impact of $o(\sqrt{n})$ term for $F_1$?

2. What is $l_j$ in Theorem 2 (Line 204)? Do the authors mean $l$?

3. In footnote 3 (Line 266), the authors say "... the limiting e.s.d for $F_0$ is not necessarily Marchenko-Pastur distribution ... This difference is not too important, as instead of using the Stieltjes transform for the Marchenko-Pastur distribution in our paper, we could use the result from Péché (2019); Piccolo & Schröder (2021) instead." Why wouldn't the authors directly use the mentioned result directly?

4. Why is there no regularization for the signal-plus-noise problem when there is regularization for the signal-only problem (Line 278-285)?

5. Typo in Line 285: "We consider on the instance-specific risk.". Typo in Line 313: " Then, any for data ...".

6. Undefined symbols in Theorem 3 (Line 312 - 324): $\asymp$ and $<<$.

7. How do the authors arrive at "Hence, we see that if the target vector y has a smaller dependence on the noise (bulk) component A, then we see that the spike affects the generalization error." in Line 380? Its connection to the previous part seems to be missing.

8. How do the authors come up with the equation for the peak point of double descent in Line 477? Is it an empirical observation or a theoretical result?

---

> ### Author Response · Authors · 2024-11-14
> **Part 1**
>
> We thank the reviewer for their detailed feedback. Let us address the key points:
>
> > Limited contribution/novelty... Most of the results in this paper are trivial extensions of the results by Hastie et al. (2022) and Li & Sonthalia (2024)
>
> We respectfully disagree. Our contributions extend beyond prior work in several important ways:
>
> 1. Finite vs Asymptotic Analysis: Prior work focused on asymptotic results. We provide finite-sample corrections that reveal how spikes affect generalization. We show when these corrections matter (small bulk variance) and when they don't (large bulk variance)
>
> 2. Technical Novelty:
>    - As the reviewers point out, we allow one eigenvalue to diverge compared to Hastie et al. 2022. This is a significant difference.
>
>    - Compared to Li and Sonthalia 2024. Only one of their two models allows for an eigenvalue to diverge. This model is closely related to the Signal only model in this paper. However, we have output noise $\varepsilon$. This creates many new dependencies requiring novel analysis techniques
>        - Consider the proof sketch on page 10. Line 491 shows that for the signal-only problem, our solution is the solution from the Li and Sonthalia 2024 paper plus an extra term. The $\hat{\varepsilon}$ is not isotropic in the ridge regularied version. Hence, we do not immediately have free independence between $\hat{Z}+\hat{A}$ and $\hat{\varepsilon}$. Hence, we need to be very careful about the alignment between the two.
>        - Hence, we get terms that are cubic and quartic in eigenvalues (vs quadratic in prior work). These required the development of new concentration bounds for these higher-order terms. See Lemmas 17 through 22.
>
>       The Li and Sonthalia 2024 paper does not consider the signal plus noise case, which introduces further terms that need bounding. Finally, the Li and Sonthalia 2024 only consider the case when $\tau_A = 1$ or more broadly $\tau_A = \Theta(1)$, we allow $\tau_A = \Theta(\sqrt{d})$. **This is also significant**.
>
> > Although the paper is motivated by Moniri et al. (2023), there are significant discrepancies.
>
> We agree. Please see the general response on how we can handle the dependency.
>
> Additionally, we think of the difference in the targets as a strength of the paper, as we can show things as the following.
>
> 1. Relationship between targets and spike size:
>    - For targets depending on the bulk (input data), large spikes are crucial
>    - The required spike size scales with input dimension and dataset size
>
> 2. Importance of spike-target alignment:
>    - The alignment between spike direction and targets significantly affects generalization
>    - This alignment term exhibits its own double descent behavior
>    - Small alignment improvements can yield large generalization gains
>
> 3. Double descent characteristics:
>    - Peak location depends on bulk variance and regularization strength
>    - Suggests weight decay regularization primarily affects the bulk, not learned features (spike)
>
> > There exists a related work (Cui et al., 2024)... Dandi et al. (2024)
>
> Thank you for bringing these to our attention. The second is quite new and, as the reviewer points out, was only posted well after the submission deadline. Hence, we believe that it is concurrent work and should **not** affect the review of our paper. We shall nonetheless discuss the two papers in the revision.
>
> There are many differences to Cui et al. 2024
>
> 1. We work in a more restricted setting but provide rigorous proofs.
>
> 2. We simplify expressions. For example, $\zeta$ in equation 17 in Cui et al. 2024  is exactly $\xi - 1$ in our paper (see Lemma 13 in the appendix for a definition). The expression in Cui et al. 2024 is left in terms of $\zeta$. **This is because the results are a product of dependent terms. Hence, simplification is not easy**. However we
>     a. Compute the expectations and variances of each of the terms
>     b. Compute the expectations of the products.
>     c. Greatly simplify expressions
>   These simplifications are a major strength of our work, as the expressions are interpretable without numerical computations.
>
> > In Line 178,  denotes the case with a single spike (as shown by Moniri et al., 2023)...
>
> We only consider the single spike case. The analysis here is similar to Sonthalia and Nadakuditi 2023. Kaushik et al. 2024 extend Sonthalia and Nadakuditi 2023 to the higher rank version. We would need to do the same to extend to multiple spikes. As mentioned before, the difficulty in the analysis was
>     a. Compute the expectations and variances of each of the terms
>     b. Compute the expectations of the products.
>     c. Greatly simplify expressions
>
> These are currently scalar expressions, so we can use commutativity. For multiple spikes, we have matrix expressions, so we no longer have commutativity. The analysis is possible, but it is just quite tedious.
>
> We ignore the $o(\sqrt{n})$ we shall highlight this as another discrepensacy

---

> > ### Author Response · Authors · 2024-11-14
> > **Part 2**
> >
> > > In footnote 3 (Line 266), the authors say "...
> >
> > If we use these results, then similar to Eqautuons C.23 in Ba et al. 2022 and Equation (5) in Moniri et al. 2023, we would have that the value of Stieljtes transform is given to us as the unique solution to a set of consistency equations. Hence, we would replace the **explicit** values in Lemmas 7 and 8 with these **implicit** values. However, we did not do so since we wanted explicit closed-form expressions. However, please see the general response, showing how this can be achieved.
> >
> > > Why is there no regularization for the signal-plus-noise problem when there is regularization for the signal-only problem (Line 278-285)?
> >
> > This limitation is primarily technical. The regularized case for the signal-plus-noise model introduces many additional terms whose mean and variance would need to be bounded, significantly complicating the analysis.
> >
> > > How do the authors arrive at "Hence, we see that if the target vector y has a smaller dependence on the noise (bulk) component A, then we see that the spike affects the generalization error." in Line 380? Its connection to the previous part seems to be missing.
> >
> > Here we have that $ y_i = \beta_*^Ta_i + \beta_*^T z_i + \varepsilon_i$. We see that for the bulk term: $a_i^T \beta_* \approx \tau_A \|\beta_*\|$. For the spike term: $z_i^T \beta_* \approx \theta\|\beta_*\|$
> >
> > Using our scaling $\theta = \tau \sqrt{n}$, the signal part is always larger. However, when the bulk also grows (i.e., $\tau_A = \Theta(d)$), the spike's effect becomes invisible. Specifically, we need:
> > - $a_i^T \beta_* = \Theta(1)$
> > - $z_i^T \beta_* = \Theta(\sqrt{n})$
> > (assuming $\|\beta_*\|= \Theta(1)$) for the spike to have a detectable effect.
> >
> > > Undefined symbols
> >
> > We apologize. $f \ll g$ means $f = O(g)$ and $f \asymp g$ means $f \ll g$ and $g \ll f$
> >
> > > How do the authors come up with the equation for the peak point of double descent in Line 477? Is it an empirical observation or a theoretical result?
> >
> > This is currently empirical.
> >
> > It can be proved by computing the expression's derivative (and second derivative) and evaluating it at the derivative. The derivative expressions are quite involved; even symbolic programs such as SciPy struggled. **Hence, this leads us back to our contribution to simplified expression**. While our expressions are further simplified compared to Cui et al. 2024, we believe further simplification is only a positive.
> >
> > > Typos
> >
> > The reviewer is correct in identifying typos. These shall be fixed.

---

> > > ### Comment · Reviewer_qVHv · 2024-11-23
> > >
> > > Thank you for the detailed responses. I appreciate that the authors introduced a dependency between bulk and spike during the rebuttal to address discrepancies with the motivating work by Moniri et al. (2023). However, I still believe the paper requires significant revision, particularly in its presentation and writing. Furthermore, in their responses, the authors argued that their novelty lies in three main points: (1) finite matrix effects, (2) exploration of different target functions, and (3) simplified expressions compared to Cui et al. (2024). While these might be interesting contributions, they should have been more thoroughly motivated in the paper. Specifically, what are the additional benefits of these aspects?
> > >
> > > Overall, after considering the authors' responses and the other reviews, I have decided to maintain my current rating.

---

> > > > ### Author Response · Authors · 2024-11-25
> > > >
> > > > We thank the reviewer for their valuable feedback. We hope that our new results help expand on contribution (2).

---

### Official Review · Reviewer_weJN · 2024-11-07

**Soundness:** 2
**Presentation:** 3
**Contribution:** 1
**Rating:** 5
**Confidence:** 3

**Summary:**

The authors analyze the generalization properties of spiked covariate models. The theoretical analysis is motivated by recent works on two-layer networks trained with a single gradient step that showed how the feature matrix possesses different spikes associated with the learning rate scaling used in the optimization step. The proof scheme uses tools coming from random matrix theory that enables the asymptotic computation of the generalization error. The theoretical claims are accompanied by coherent numerical illustrations.

**Strengths:**

The paper is nicely written. The mathematical claims are correctly phrased and the numerical illustrations are coherent with the main text. The research problem is relevant in the theoretical machine learning community.

**Weaknesses:**

My main concern with the present submission is the lack of clear elements of novelty. The paper heavily relies on results coming from related works and it restricts their setting in many ways (as fairly reported by the authors at the end of the manuscript). More details are provided below.

**Questions:**

As hinted above my main concern on this manuscript is the close relationship with previous works, namely (Ba et al., 2022; Moniri et al., 2024).

Could the authors comment on the link between their results and (Ba et al. 2022) in the context of Gaussian Universality (see e.g. [1]) ? From my understanding of their paper, i.e. a single spike in the feature matrix, they show that in the learning rate regime considered in this paper Gaussian Universality should hold. There is indeed an extensive regime of learning rates after the BBP transition that still falls under the umbrella of Gaussian models, resulting in effectively "linear" generalization properties.

One additional weakness of this submission is the related works coverage. The authors do a great job in covering the random matrix theory literature, while many manuscripts that analyze learned representations with gradient descent with different tools are not properly mentioned, see e.g. [2,3,4]. Although in these works the authors do not focus on the exact asymptotic calculation of the test error, many insights should translate to the present setting. On the other hand, [5] precisely characterize the generalization error using non-rigorous methods; what is the relationship with the present work?

The results in the present submission should relate directly to the ones in Section 4 of (Moniri et al. 2024), albeit the differences correctly reported by the authors in the two settings. Could the author elaborate on this?

What is the bottleneck for the present thereotcial tools to analyze multiple spikes (corresponding to higher learning rate scaling in Moniri et al. 2024)?

Closely related to the above, [5] worked along the lines of (Moniri et al. 2024) to provide the equivalent description in the regime where the spikes recombine with the bulk (maximal scaling regime). Do the authors see a possible extension of their analysis to this scaling?


- [1] Hu & Lu 2022, Universality laws for high-dimensional learning with random features.
- [2] Damian et al. 2022, Neural networks can learn representations with gradient descent.
- [3] Dandi et al. 2023, How two-layer neural networks learn, one (giant) step at a time.
- [4] Ba et al. 2023, Learning in the presence of low-dimensional structure: a spiked random matrix perspective.
- [5] Cui et al. 2024, Asymptotics of feature learning in two-layer networks after one gradient-step.

**Details Of Ethics Concerns:**

N/A.

---

> ### Author Response · Authors · 2024-11-14
>
> We thank the reviewer for the feedback and comments. Key differences between our work and important prior research are that we (1) provide finite matrix correction terms and (2) offer simplified closed-form expressions.
>
> > Could the authors comment on the link between their results and (Ba et al. 2022) in the context of Gaussian Universality (see e.g. [1]) ?
>
> Yes, the problem we and prior work are interested in understanding is the generalization error for the following. First, we solve a regression problem
> $$ \beta\_{LS} = argmin ||y - \beta^T F||\_F^2 + \lambda ||\beta||\_2^2 $$
> Then, we are interested in the generalization performance of $\beta\_{LS}$. Let's call this risk $R(F)$ to highlight the dependence on $F$. The difference between setups lies in the $F$ term. There are three different $F$'s considered.
>
> 1. $F_{CK} := \sigma(WX)$. For Gaussian $X$ and $W$, after taking a step of GD. This is from Ba et al. 2022
> 2. $F_{CE} := \theta_1 WX + \theta_2 A$ where $A$ has IID standard Gaussian entries independent of $W,X$ and $W$ is after taking gradient step of GD
> 3. $F_{SP} := A + \theta uv^T$, where $A$ has IID standard Gaussian entries.
>
> The Ba et al. 2022 paper shows that in the small learning rate regime, $R(F_{CK}) = R(F_{CE})$ **asymptotically**.
>
> However, we do things differently:
>
> 1. We allow spikes from the large learning rate limit. Hence, the result from Ba et al. 2022 does not apply. Equation 3.1 in Ba et al. shows that for small learning rate, the size of the spike is $\Theta(1)$, where as for large learning rates, it is $\Theta(\sqrt{d})$ (note for the rank one spike the Frobenius norm is equal to the spectral norm). We are interested in the case when the spike is large. The idea behind the large step size is that we are in a regime in **which the Gaussian Equivalence Property is no longer true**.
>
> 2. We provide more precise correction terms for finite matrices. While the prior work is purely asymptotical.
>
> 3. In our rebuttal, we also generalize to models closer to that from Moniri et al.
>
> > One additional weakness of this submission is the related works coverage.
>
> We thank the reviewer for pointing us to these works. We shall add these references.
>
> [5] characterizes the risk for the setting from Moniri et al. 2023 using, as the reviewer and the paper says, using the non-rigorous replica symmetry method. The differences are three-fold:
>
> 1. Our results in the paper are for a restricted setting; however, we provide proof.
> 2. We simplify expressions. For example, $\zeta$ in equation 17 in Cui et al. 2024 is exactly $\xi - 1$ in our paper (see Lemma 13 in the appendix for a definition). The expression in Cui et al. 2024 is left in terms of $\zeta$. **This is because the results are a product of dependent terms. Hence, simplification is not easy**. However we
>     a. Compute the expectations and variances of each of the terms
>     b. Compute the expectations of the products.
>     c. Greatly simplify expressions
>
> We believe these are the main challenges we overcome in our proof.
>
> > Could the authors elaborate on the connection with Moniri et al. 2024?
>
> While Theorem 4.5 in Moniri et al. 2023 shows that the difference between the test error using $F_1$ (features after 1 step) and $F_0$ (initial features) converges to a constant, there are important differences in our approach:
>
> 1. Moniri et al. 2023 has expressions requiring solutions to fixed point equations (see equations (5) in their paper). These only hold asymptotically with hard-to-quantify approximation rates.
>
> 2. In contrast, we provide:
>
>    - **Closed form expressions** for the risk itself (not just differences as is the case in Moniri et al. 2023)
>
>    - **Better control on approximation error rates**, enabling analysis of finite matrices
>
>    - The above two allow better control on understanding the relationship between the bulk and spike.
>
> > What is the bottleneck for analyzing multiple spikes?
>
> The analysis here is similar to Sonthalia and Nadakuditi 2023. Kaushik et al. 2024 extend Sonthalia and Nadakuditi 2023 to the higher rank version. We would need to do the same to extend to multiple spikes. As mentioned before, the difficulty in the analysis was bounding variances. These are currently scalar expressions hence we can use commutativity. For multiple spikes we have matrix expressions. Hence no longer have commutativity. The analysis is possible, but it is just quite tedious.
>
> > maximal scaling regime
>
> We don't know this regime. Does the reviewer mean when the step size is too small and we do not see a spike or the regime from [1] where the spectrum becomes heavy-tailed? In the heavy-tailed situation analysis similar to [2] can be used.
>
> [1] Martin and Mahoney JMLR 2021 - Implicit Self-Regularization in Deep Neural Networks: Evidence from Random Matrix Theory and Implications for Learning
>
> [2] Wang et al. 2024 AISTATS - Near-interpolators: Rapid norm growth and the trade-off between interpolation and generalization

---

> ### Comment · Reviewer_weJN · 2024-11-26
>
> I warmly thank the authors for addressing my concerns. After reading carefully the other reviewers' comments, I believe that the paper still heavily relies on previously published works, and I would like therefore to keep my original score. On the writing side,I believe the authors should expand the discussion about the connection with previous works not belonging only to the Random Matrix Theory literature (e.g. see the suggested references [1,5]).

---

> > ### Author Response · Authors · 2024-11-28
> >
> > We thank the reviewer for the feedback and help in improving the paper.

---

### Author Response · Authors · 2024-11-14
**Introducing Dependency Between Bulk and Spike**

A common criticism among reviewers was our abstraction of the dependency between bulk and spike components. Here we demonstrate how our proof framework extends to handle the dependent case from Moniri et al. 2023.

Recall that Moniri et al.'s spike structure is:
$$ \sigma(W_0\tilde{X}^T) + c (\tilde{X}\beta_{sp}) \zeta^T $$
where $\tilde{X}$ is Gaussian data, $W_0$ is inner layer weights, and $\zeta$ are outer layer weights.

We modeled this as:
$$ A + \theta vu^T $$
where $A$ is Gaussian. Below we show how our analysis extends to Moniri et al.'s setting.

## Introducing Dependence

First, consider the intermediate structure:
$$ X = A + \theta (A\beta_{sp})u^T $$

To analyze this, we only need to modify two parts of our proof:

1. In Lemma 10, the norm of $v^T A^\dag$  scaling changes:
   - Original: $\mathbb{E}_{\lambda}\left[\frac{\lambda}{(\lambda + \mu^2)^2}\right]$
   - New: $v = A\beta\_{sp}$, and norm becomes $\|\beta\_{sp}\|$

   where expectations are over the Marchenko-Pastur distribution.

2. The variable $t = (I-AA^\dag) v$ is no longer zero.

For the Signal-Only case, this gives bias:
$$ \frac{\theta_{tst}^2}{n_{tst}}\left[(\beta_*^T u)^2 + \tau_{\varepsilon}^2\left(\frac{(1+c)}{2T_1} + \frac{\mu^2 c - T_1}{2\tau_{A_{trn}}^2 T_1} \right)\right] $$
where $T_1$ is unchanged. We do not present the whole formula for brevity. To extract insights, we consider the same simplifications for the paper.

Under simplifications ($\mu = 0$, $\tau_{A_{trn}} = \tau_{A_{tst}}$, $\theta = \tau \sqrt{n}$), for $c > 1$ we get:
$$ \tau_{A}^2 (\beta_*^Tu)^2\left(1+\frac{\tau_A^2}{c}\|\beta_{sp}\|^2\right) + \tau^2_{\varepsilon}\frac{c}{c-1}\left(2 + \frac{\tau_A^2}{c}\|\beta_{sp}\|^2\right) $$

Note: Due to the extra $A$ factor, this only holds for $\tau_A = \Theta(1)$ versus our original $\tau_A = O(\sqrt{d})$.

Note: this is the Signal only version so $y = \theta (A \beta\_{sp}) u^T\beta\_*$.

## Full Moniri et al. Structure

Now consider:
$$ X = \sigma(W_0A^T) + c (A\beta_{sp}) \zeta^T $$

The risk becomes a random variable dependent on $W_0$, $\zeta$, and $\beta_{sp}$. Using standard assumptions (isotropic with unit expected norm for the $\zeta$, and $\beta_{sp}$ and the rows of $W_0$), we analyze the expected risk. Importantly, since $W_0$ and $(\beta_{sp}, \zeta)$ are independent, the bulk remains independent of the spike. This is because functions of independent random variables are independent. Hence, our assumptions are reasonable.

To get the generalization error for this model, we need to replace Lemmas 7-9. As an example, the new Lemma 7 resembles equations from Moniri et al. (Eq. 5) and Ba et al. (Eq. C.23):

**Lemma 7:** For $W_0$ ($m \times d$), $X$ ($n \times d$), $m < n$, with $d/n \to \phi$, $d/m \to \psi$, $m/n \to c$:

1. $\mathbb{E}\left[\frac{1}{\lambda+\mu^2}\right] = \frac{c}{\tau_A^2}m_c\left(-c\frac{\mu^2}{\tau_A^2}\right)$
2. $\mathbb{E}\left[\frac{1}{(\lambda+\mu^2)^2}\right] = \frac{c^2}{\tau_A^4}m_c'\left(-c\frac{\mu^2}{\tau_A^2}\right)$
3. $\mathbb{E}\left[\frac{1}{(\lambda+\mu^2)^2}\right] = \frac{c^3}{2\tau_A^6}m_c''\left(-c\frac{\mu^2}{\tau_A^2}\right)$

where $m_c(z)$ satisfies:
$$\frac{\psi}{z} H(z) - \frac{\psi - 1}{\psi} = m_c(z)$$
$$H(z) = 1 + \frac{H^\phi(z)H^\psi(z)(c_1 - c_2)}{\psi z} + \frac{H^\phi(z) H^\psi(z)c_2}{\psi z - H^\phi(z)H^\psi(z)c_2}$$
with $H^\kappa(z) = 1 - \kappa + \kappa H(z)$

---------
Advantages of our approach
---------

This approach avoids using the Gaussian Equivalence property, providing finer control over finite matrix approximation errors. While we must restrict $\tau_A$ and $\theta$ magnitudes and take expectations over $W_0$ and $\zeta$, this allows us to:
1. Better understand finite matrix effects
2. Explore different target functions than Ba et al. and Moniri et al.

We are happy to represent the corresponding results for the Signal Plus Noise case and the missing details. We presented a shortened version for brevity.

----------

We are currently still working on the revision and will post it shortly.

---

### Author Response · Authors · 2024-11-14

We thank the reviewers for their comments and help in improving the paper and hope that our responses with the new results have improved their opinions. If there are further questions that we can answer, we would be happy to continue the discussion.

---

### Author Response · Authors · 2024-11-24
**Interpolation between signal-plus-noise and signal-only models**

One other implicit criticism seems to be the lack of connection between the two models we are studying. We would like to point out that there can be a way to interpolate the signal-plus-noise and signal-only models.

Intuitively, we introduce extra parameters $\alpha_{trn}$ and $\alpha_{tst}$ that define the magnitude of dependence of y on the noise matrices $A_{trn}$ and $A_{tst}$. If they are both 0, then we should recover the signal-only case. To be specific, here we train with:

$$\beta_{spn}^T = argmin \| (Z_{trn} + \alpha_{trn}A_{trn})\beta_* + \varepsilon_{trn}^T -(Z_{trn} + A_{trn})\beta \|_F^2$$

And evaluate the following error:

$$\frac{1}{n_{tst}} \mathbb{E} \left[\| (Z_{tst} + \alpha_{tst}A_{tst})\beta_* - (Z_{tst} + A_{tst})\beta_{spn} \|_F^2 \right]$$

Then using the same decomposition as before and approaches from random matrix theory, we can arrive that a similar error formula that involves extra terms of $\alpha$. We present the formula for $c < 1$:
$$Bias = \frac{\theta_{tst}^2\tau_{\varepsilon_{trn}}^2}{n_{tst}(\theta_{trn}^2c + \tau_{A_{trn}}^2)}\left( \frac{c}{1-c}\right) + \frac{(1 - \alpha_{trn}^2)\tau_{A_{trn}}^4\theta_{tst}^2}{n_{tst}(\theta_{trn}^2c + \tau_{A_{trn}}^2)^2}(\beta_*^Tu)^2$$

$$\mathbf{Variance + Others} = $$
$$\frac{\tau_{A_{tst}}^2\tau_{\varepsilon_{trn}}^2c}{\tau_{A_{trn}}^2(1-c)} + \frac{\tau_{A_{tst}}^2}{d}\left[(\alpha_{trn} - \alpha_{tst})^2\|\beta_*\|^2 + 2(1 - \alpha_{trn})(\alpha_{trn} - \alpha_{tst}) \frac{\theta_{trn}^2(\beta_*^Tu)^2}{\theta_{trn}^2c} \right] + $$
$$\frac{\tau_{A_{tst}}^2}{d}\left[(\alpha_{trn} - 1)^2(\beta_*^Tu)^2 \frac{\theta_{trn}^2}{(\theta_{trn}^2c + \tau_{A_{trn}}^2)^2}(\theta_{trn}^2 + \tau_{A_{trn}}^2)\frac{c^2}{1-c} - \frac{\tau_{\varepsilon_{trn}}^2}{\tau_{A_{trn}}^2}\frac{\theta_{trn}^2}{(\theta_{trn}^2c + \tau_{A_{trn}}^2)^2}\frac{c^2}{1-c}\right]$$
We note here when we set $\alpha_{trn} = \alpha_{tst} = 1$, we have the current signal-plus-noise formula in the paper. When we set both to 0, we have the current signal-only formula.

Consider similar simplifications. We set $\tau_{A_{trn}}^2 = \tau_{A_{tst}}^2 = d$, $\theta_{trn}^2 = \tau^2n$, $\theta_{tst}^2 = \tau^2n_{tst}$. We obtain the following formula:
$$\tau_{\varepsilon_{trn}}^2\frac{c}{1-c} + (\alpha_{trn} - \alpha_{tst})^2\|\beta_*\|^2 + 2(1-\alpha_{trn})(\alpha_{trn} - \alpha_{tst})(\beta_*^Tu)^2 + (\alpha_{trn}-1)^2(\beta_*^Tu)^2\frac{1}{1-c}$$

Again when we set $\alpha$'s to be 1, we have some cancellations, making our current signal-plus-noise formula a special case. We hope this connection can provide more intuition and help reviewers understand the work better.

---

### Author Response · Authors · 2024-11-24
**Revised Version**

We have posted the revised version, where we fixed the typos and ambiguities pointed out by the reviewers.

We would like to thank the reviewers for their invaluable feedback and their time to help improve our work. If there is anything else we can answer, please let us know, and we would be more than happy to do so.

---

### Meta-Review · Area_Chair_aazW · 2024-12-20

**Metareview:**

Summary of Scientific Claims and Findings:
The paper investigates the generalization properties of least squares regression with spiked covariance matrices, motivated by neural network training dynamics after one gradient step. The authors provide asymptotic analyses and derive corrections for finite-sample generalization errors, introducing a novel parameter that interpolates between "signal-only" and "signal-plus-noise" regimes. The findings highlight the impact of spike-target alignment on generalization and demonstrate phenomena like double descent.

Strengths:
Well-Explained Setup: The authors offer a clear motivation by connecting spiked covariance to neural network dynamics post-training.
Finite-Sample Corrections: The inclusion of finite-sample effects extends beyond prior asymptotic-focused works.
Novel Interpolation Parameter: The introduction of a parameter to interpolate between different models is a thoughtful addition.
Mathematical Rigor: The proofs are detailed and provide bias-variance decompositions and characterizations of risk.


Weaknesses:
Limited Novelty: The paper heavily relies on extensions of prior works (e.g., Moniri et al. and Li & Sonthalia) without sufficiently differentiating its contributions.
Weak Connection to Neural Networks: While motivated by neural networks, the setting deviates significantly, and the connection to practical feature learning scenarios is tenuous.
Clarity and Presentation Issues: Ambiguities in notation, numerous typos, and inconsistencies reduce readability and undermine clarity.
Assumptions: The paper relies on strong assumptions, such as independence of cross terms, which may limit applicability to realistic machine learning settings.
Impact on the Field: Despite deriving finite-sample corrections, the theoretical advancements and insights remain incremental.

Decision:
Given the identified strengths and weaknesses, the paper is marginally below the acceptance threshold. While it offers an interesting exploration of spiked covariance effects on generalization, it lacks sufficient novelty and clarity to merit acceptance. However, I encourage the authors to address these concerns in future submissions, particularly improving clarity and rigorously connecting the results to neural networks and practical scenarios.

**Additional Comments On Reviewer Discussion:**

Reviewer Concerns:
Novelty: Reviewers questioned the originality of the findings, noting overlap with prior works.
Presentation: Typos and unclear notation were significant points of contention.
Connection to Neural Networks: Reviewers felt the connection to practical settings, such as neural networks after one gradient step, was not sufficiently developed.
Assumptions: The independence assumption for cross terms was viewed as overly restrictive.
Author Rebuttal:
The authors responded by:

Introducing new results to address dependency between spike and bulk components.
Expanding proofs to include finite-sample effects.
Adding a novel interpolation parameter to broaden the scope of the analysis.
These changes led to marginal improvements in some reviewer scores, reflecting better alignment with reviewer expectations but not fully resolving core concerns about novelty and clarity.

Final Assessment:
While the authors’ responses added valuable clarifications and incremental extensions, the paper still suffers from limited novelty and an unclear narrative. The reviewers’ concerns about applicability and presentation remain valid. These factors weighed significantly in the final decision to reject.

---

### Decision · Program_Chairs · 2025-01-22

Reject